# mTORC2–NDRG1–CDC42 axis couples fasting to mitochondrial fission

Nuria Martinez-Lopez[1,2,3,4], Pamela Mattar[1,3], Miriam Toledo [3,11],
Henrietta Bains[5], Manu Kalyani[3], Marie Louise Aoun[3], Mridul Sharma[1,3],
Laura Beth J. McIntire[6], Leslie Gunther-Cummins[7], Frank P. Macaluso[7],
Jennifer T. Aguilan[8], Simone Sidoli[8], Mathieu Bourdenx[9,10] &
Rajat Singh [1,2,3,5,4] ✉

Fasting triggers diverse physiological adaptations including increases in circulating fatty acids and mitochondrial respiration to facilitate organismal survival. The mechanisms driving mitochondrial adaptations and respiratory sufficiency during fasting remain incompletely understood. Here we show that fasting or lipid availability stimulates mTORC2 activity. Activation of mTORC2 and phosphorylation of its downstream target NDRG1 at serine 336 sustains mitochondrial fission and respiratory sufficiency. Time-lapse imaging shows that NDRG1, but not the phosphorylation-deficient NDRG1^Ser336Ala mutant, engages with mitochondria to facilitate fission in control cells, as well as in those lacking DRP1. Using proteomics, a small interfering RNA screen, and epistasis experiments, we show that mTORC2-phosphorylated NDRG1 cooperates with small GTPase CDC42 and effectors and regulators of CDC42 to orchestrate fission. Accordingly, *Rictor*^KO, NDRG1^Ser336Ala mutants and *Cdc42*-deficient cells each display mitochondrial phenotypes reminiscent of fission failure. During nutrient surplus, mTOR complexes perform anabolic functions; however, paradoxical reactivation of mTORC2 during fasting unexpectedly drives mitochondrial fission and respiration.

Dynamic mitochondrial networks are essential for mitochondrial function and organismal wellbeing[1]. Mitochondrial networking is in turn controlled by coordinated fission and fusion events regulated by proteins localized at endoplasmic reticulum (ER)–mitochondria contacts (mitochondria-associated membranes, MAMs)[2]. Indeed, blocking fission by deleting dynamin-related protein 1 (DRP1) (refs. 3,4) or mitochondrial fission factor (MFF)[4] or inhibiting fusion by silencing optic atrophy 1 (OPA1) (ref. 5) and mitofusin (MFN) proteins[6] alters mitochondrial morphology and function. Although, recent work has identified a role of AMPK in mitochondrial fission[7], we do not completely understand how dietary stressors such as fasting influence mitochondrial dynamics in intact organisms. Since nutrient signalling is coupled to healthspan, it remains critical to understand how impairment in these processes lead to age-related diseases.

In this Article, we show that nutrient-responsive mTORC2 is paradoxically reactivated by fasting to stimulate mitochondrial fission.

[1]Department of Medicine, University of California Los Angeles, Los Angeles, CA, USA. [2]Vatche and Tamar Manoukian Division of Digestive Diseases, University of California Los Angeles, Los Angeles, CA, USA. [3]Department of Medicine, Albert Einstein College of Medicine, Bronx, NY, USA. [4]Liver Basic Research Center at University of California Los Angeles, Los Angeles, CA, USA. [5]Department of Developmental and Molecular Biology, Albert Einstein College of Medicine, Bronx, NY, USA. [6]Department of Radiology, Weill Cornell Medicine, New York, NY, USA. [7]Department of Anatomy and Structural Biology, Albert Einstein College of Medicine, Bronx, NY, USA. [8]Department of Biochemistry, Albert Einstein College of Medicine, Bronx, NY, USA. [9]UK Dementia Research Institute, London, UK. [10]UCL Queen Square Institute of Neurology, London, UK. [11]Present address: Neuronal Control of Metabolism Laboratory, Institut d'Investigacions Biomediques August Pi i Sunyer (IDIBAPS), Barcelona, Spain. ✉e-mail: rajatsingh@mednet.ucla.edu

We show that the mTORC2–SGK1 cascade phosphorylates a known target NDRG1 (ref. [8]) at Ser336, which then engages with mitochondria to drive fission. NDRG1, but not the phosphorylation-deficient NDRG1$^{Ser336Ala}$ mutant, interacts with CDC42 (ref. [9]), a cytokinetic protein with intrinsic GTP hydrolysis activity[10], to drive fission. mTORC2, NDRG1 and CDC42 each localize to MAMs, and silencing *Rictor*, *Ndrg1* or *Cdc42* or identified CDC42 effectors blocks fission. Thus, paradoxical reactivation of an mTORC2–NDRG1$^{Ser336}$–CDC42 axis drives mitochondrial fission during fasting.

## Results

### Fasting or lipid availability activates mTORC1/2 signalling

Fasting increases circulating free fatty acids (FFAs), which undergo mitochondrial oxidation to support organismal sustenance[11]. To understand the mechanisms driving metabolic adaptations during fasting or fatty acid availability, we sought to identify the signalling cascades that are activated under these conditions. To this purpose, we performed unbiased quantitative phosphoproteomics in livers of mice that were (1) basal fed; (2) overnight (14–16 h) fasted; or fasted overnight and then gavaged with (3) dietary triglycerides as corn oil; or (4) BODIPY FL C$_{16}$/palmitic acid; or (5) refed a high-fat diet (Fig. 1a). Corn oil or BODIPY FL C$_{16}$ groups served as models for exogenous lipid availability, while refeeding served as a control to simulate physiological feeding. Corn oil is absorbed as FFA and repackaged and secreted by enterocytes as lipoproteins and subsequently delivered to liver as FFA. Delivery of BODIPY FL C$_{16}$ to livers was confirmed by direct fluorescence of liver slices (Extended Data Fig. 1a). Phosphoproteomics in the five groups identified 2,160 phosphosites across 942 phosphoproteins, of which 863 phosphosites (39.95%) were significantly modulated. Unsupervised hierarchical clustering analyses grouped basal and refed cohorts into one cluster, while lipid-exposed groups (that is, fasted, corn oil and BODIPY FL C$_{16}$) clustered into the second group (Extended Data Fig. 1b and Supplementary Table 1). A second clustering analysis to determine phosphoproteins that are coordinately modulated revealed a major 'green cluster' encompassing 86.9% of significantly modulated phosphosites (Fig. 1b and Supplementary Table 2). The average normalized abundance of phosphosites belonging to the green cluster (to better appreciate group-to-group modulation rather than phosphoprotein expression differences) was significantly higher in lipid-exposed groups, when compared with basal and refed groups (Fig. 1c). Interestingly, despite strong reduction in phosphorylations in the refed group, the green cluster-normalized abundance was relatively higher in the refed cohort compared with the basal fed cohort, indicating qualitative differences in phosphopeptides between basal fed and refed groups (Fig. 1c).

To predict the kinases putatively modulating the phosphosites in the green cluster, we used GPS algorithm with the interaction filter, or in vivo GPS (iGPS)[12], which revealed that these phosphosites are targets of cyclin-dependent kinases (CDKs), Ca$^{2+}$/calmodulin-dependent kinases, mitogen-activated protein kinases (MAPKs) and Ser/threonine

(Thr) cAMP-dependent, cGMP-dependent and protein kinase C (AGC) kinases (Fig. 1d, Extended Data Fig. 1c and Supplementary Table 2). Pairwise comparisons showed upregulation of CDK2/CDK8 and MAPK1 during fasting when compared with basal group (Fig. 1e). Corn oil, which is devoid of proteins, and BODIPY FL C$_{16}$, each perturbed a number of kinase groups albeit to a lesser extent when compared with fasting. Interestingly, iGPS revealed enrichment of downstream substrates of mTORC1/2 signalling, that is, RPS6KA2/B2 (ref. [13]), AKT1-3 (ref. [14]), PRKCB/D (refs. [15,16]) and SGK1-3 (ref. [17]) in livers of corn oil-gavaged mice, and SGK1-3 (ref. [17]) in livers of BODIPY FL C$_{16}$-treated mice, indicating lipid-driven mTORC1/2 activation (Fig. 1f,g). Refeeding reduced the overall kinase network, but expectedly activated nutrient-sensitive kinases, for example, AKT1-3 and SGK1-3, and suppressed those induced by fasting, for example, CDK2 (Extended Data Fig. 1d).

Confirming that dietary triglycerides stimulate mTORC1/2 signalling, immunoblotting showed higher levels of P-P70$^{Thr389}$ and P-S6$^{Ser235/236}$ (mTORC1 markers) and P-AKT$^{Ser473}$ (mTORC2 marker) in livers in response to corn oil without perturbing other nutrient-sensitive kinases, for example, AMPK[18] (Extended Data Fig. 1e,f). Since insulin activates mTOR, we asked if insulin drives mTOR activation in response to corn oil. Interestingly, low-dose streptozotocin (STZ), which depletes insulin, causing hyperglycaemia due to β-cell destruction (Extended Data Fig. 2a–c), failed to block lipid-driven mTORC1 (P-S6$^{Ser235/236}$), mTORC2 (P-AKT$^{Ser473}$) and AKT (P-AKT$^{Thr308}$) activation (Extended Data Fig. 2d–g). Furthermore, corn oil per se did not elicit insulin secretion (Extended Data Fig. 2b) excluding a role for insulin in lipid-driven mTOR activation. Similarly, equivalent serum levels of adipokine leptin or IGF-1 in corn-oil-treated and untreated mice excluded their role in lipid-driven mTOR activation (Extended Data Fig. 2h,i).

Since fasting redistributes FFA from adipose tissue to liver[11,19], we asked whether fasting-induced increases in FFA (Extended Data Fig. 1g) reactivate mTORC1/2 signalling in liver, as noted with corn oil exposure. Interestingly, fasting for 14 h reactivates mTORC1/2 signalling in liver, indicated by phosphorylation of their respective targets[20], P70$^{Thr389}$ (mTORC1) and AKT$^{Ser473}$, SGK1$^{Thr256}$ and NDRG1$^{Thr346}$ (mTORC2) (Fig. 1h). By contrast, fasting did not affect phosphorylated levels of PKA$^{Thr197}$, PKCα/βII$^{Thr638/641}$, PKCδ$^{Thr505}$, PKCδ/θ$^{Ser643/676}$ and PKCζ/λ$^{Thr410/403}$ in liver (Extended Data Fig. 2j,k), suggesting that fasting specifically reactivates the AKT and SGK1, but not PKA/PKC, arms of mTORC2 signalling. Hence, exogenous lipids or endogenous FFA availability during fasting activates mTORC1/2 signalling.

### mTORC2 supports mitochondrial respiration during fasting

To determine the physiological roles of mTOR reactivation during fasting, we inactivated mTORC1 or mTORC2 or hyperactivated mTORC1 by knocking out *Raptor*[21], *Rictor*[22] or *Tsc1* (ref. [23]), respectively, using liver-restricted AAV8-TBG-iCre (Fig. 2a,b and Extended Data Fig. 3a,b). Loss of *Rictor*/mTORC2 activity in liver, and not fat or muscle, was confirmed by reduced AKT$^{Ser473}$ phosphorylation (Extended Data Fig. 3c). Since fasted livers accumulate triglyceride, we examined the effect of

**Fig. 1 | Phosphoproteomics reveal kinases that respond to fasting or lipids.** **a**, Phosphoproteomics in livers as per plan in cartoon. **b**, Heat map and hierarchical clustering of phosphosites across groups indicated in **a**. **c**, Phosphoproteome-wide comparisons via *z* score normalization of phosphosites in green cluster. Grey dots represent individual phosphosites. Blue diamonds represent group means. ***$P < 0.001$, non-parametric ANOVA (Kruskal–Wallis statistic 797.3, $P < 0.0001$) followed by Dunn's multiple comparisons test. **d**, iGPS prediction of upstream individual kinases that respond to lipids across groups indicated and identified in the green cluster in **b**. **e**–**g**, Pairwise comparisons between indicated groups (fasted versus basal (**e**), corn oil versus fasted (**f**), and BODIPY FL C$_{16}$ versus fasted (**g**)), showing upregulated or downregulated kinase networks. For **a**–**g**, $n = 4$ mice. For **d**, darker colour intensity reflects higher kinase score. **h**, Immunoblots (IB) and quantification for indicated proteins in livers of 2–10-month-old male and female mice that were fed or fasted for indicated

durations. *N* values for number of mice analysed at each timepoint for individual phosphoproteins are indicated in parentheses. P-P70$^{Thr389}$/P70: 0 h ($n = 27$), 3 h ($n = 21$), 8 h ($n = 5$), 14 h ($n = 16$) and 20 h ($n = 14$); P-S6$^{Ser235/236}$/S6: 0 h ($n = 26$), 3 h ($n = 20$), 8 h ($n = 5$), 14 h ($n = 16$) and 20 h ($n = 14$); P-AKT$^{Ser473}$/AKT: 0 h ($n = 26$), 3 h ($n = 21$), 8 h ($n = 5$), 14 h ($n = 16$) and 20 h ($n = 15$); P-SGK1$^{Thr256}$/SGK1: 0 h ($n = 11$), 3 h ($n = 9$), 8 h ($n = 4$), 14 h ($n = 9$) and 20 h ($n = 12$); P-NDRG1$^{Thr346}$/NDRG1: 0 h ($n = 17$), 3 h ($n = 14$), 8 h ($n = 5$), 14 h ($n = 12$) and 20 h ($n = 13$); and P-AKT$^{Thr308}$/AKT: 0 h ($n = 27$), 3 h ($n = 20$), 8 h ($n = 5$), 14 h ($n = 16$) and 20 h ($n = 15$). Ponceau is loading control. Individual replicates and means are shown. *$P < 0.05$ and **$P < 0.01$, one-way ANOVA followed by Tukey's multiple comparisons test (**h**). Please refer to Supplementary Table 10 statistical summary, and Supplementary Tables 1 and 2. Source numerical data are available in Source Data Extended Data Table 1, and unprocessed blots are available in the Source Data for this figure.

loss of each gene on fasting-induced increases in liver triglycerides. While control and mTORC1 inactivated (*Raptor*[KO]) livers showed equivalent liver triglycerides during fasting, hyperactivation of mTORC1

(*Tsc1*[KO]) lowered liver triglycerides (Fig. 2c) consistent with the role of mTORC1 in VLDL secretion[24]. Surprisingly, in contrast to the established triglyceride lowering effect of *Rictor* loss in fed/obesogenic states[25],

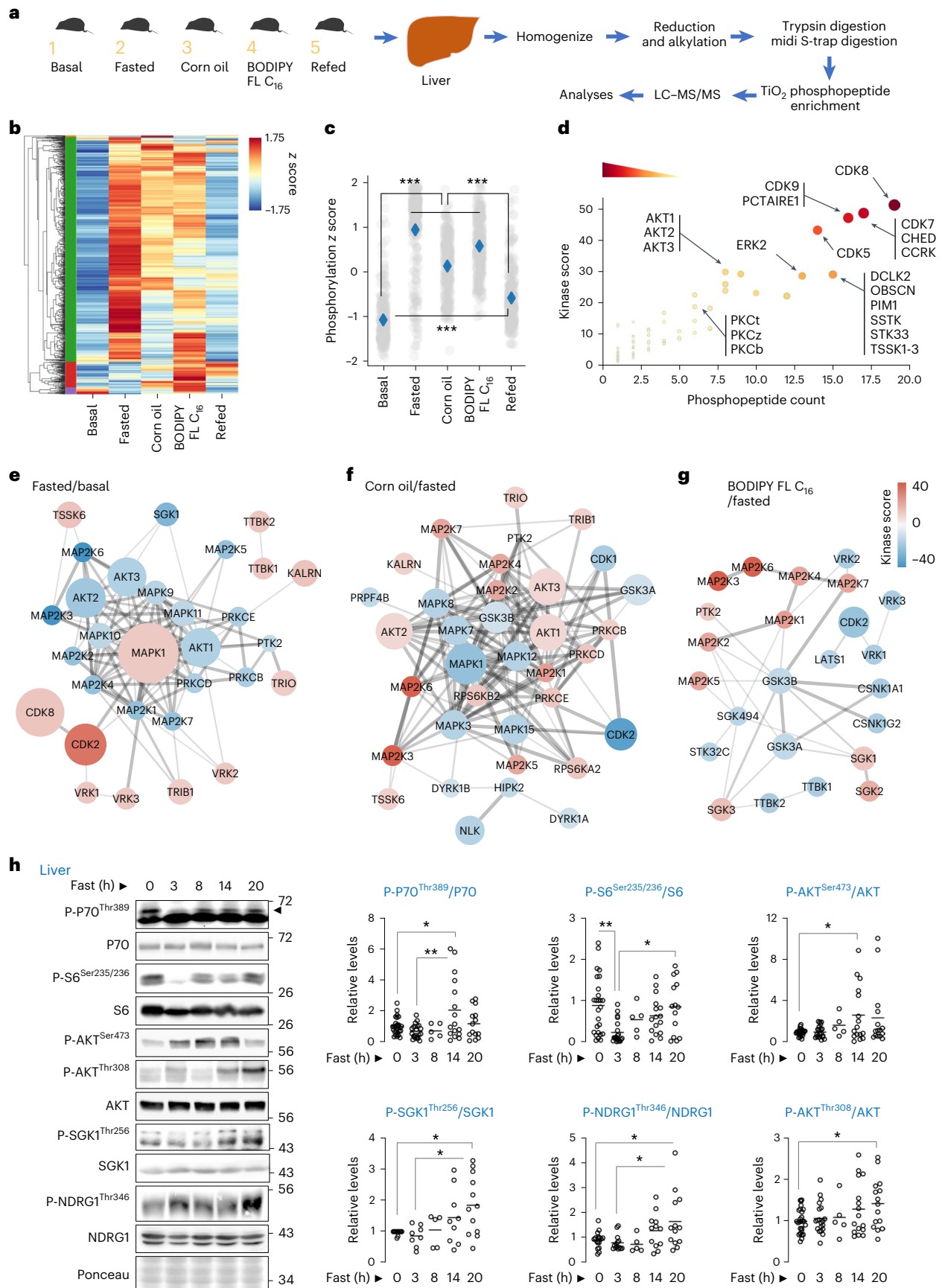

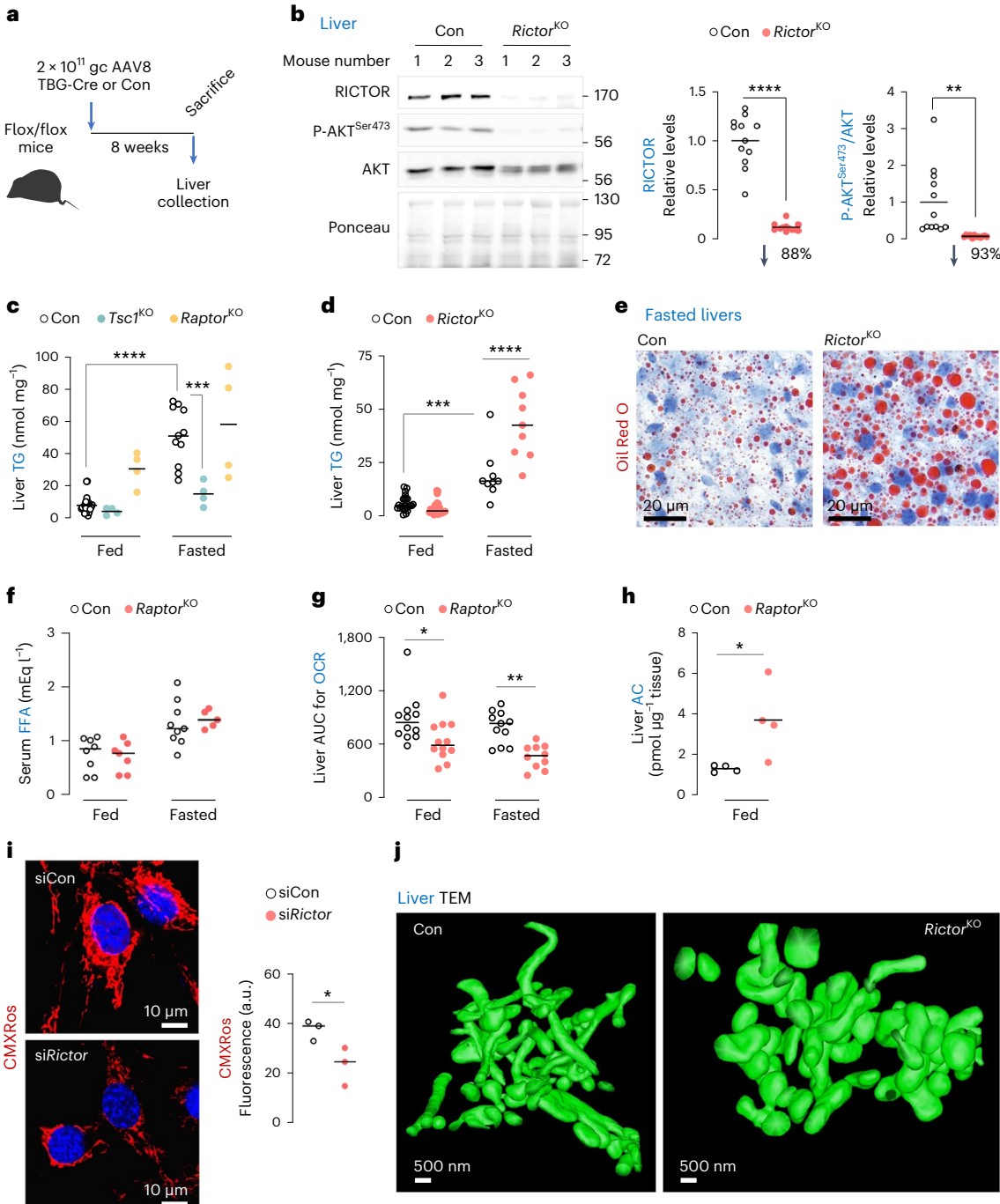

**Fig. 2 | mTORC2 supports mitochondrial respiration and triglyceride disposal during fasting. a**, Generation of liver-specific knockout (KO) of *Rictor*, *Tsc1* or *Raptor*. **b**, Representative immunoblots to validate deletion of *Rictor* gene in livers of 4–6-month-old *Rictor*[KO] male and female mice. Quantifications and percentage reduction of protein levels for RICTOR and P-AKT[Ser473]/AKT in *Rictor*[KO] livers are shown (*n* = 12 mice). **c,d**, Liver triglyceride (TGs) in 3–7-month-old Con, *Tsc1*[KO] and *Raptor*[KO] (**c**) or 4–5-month-old Con and *Rictor*[KO] (**d**) male and female mice that were fed or fasted for 14–16 h. *N* values for number of mice per group are indicated in parentheses. For **c**, fed Con (*n* = 27), fasted Con (*n* = 11) and fed or fasted *Tsc1*[KO] or *Raptor*[KO] mice (*n* = 4). For **d**, fed Con (*n* = 24), fasted Con (*n* = 13), fed *Rictor*[KO] (*n* = 23) and fasted *Rictor*[KO] (*n* = 12). **e**, Representative Oil Red O stains in livers of 6-month-old Con or *Rictor*[KO] male mice fasted for 14–16 h (*n* = 5 mice). **f**, Serum FFA levels in fed or 14–16 h fasted 3–6-month-old Con or *Rictor*[KO] male mice. Fed Con (*n* = 8), fasted Con (*n* = 9), fed *Rictor*[KO] (*n* = 7) and fasted *Rictor*[KO] (*n* = 5) mice. **g**, Area under curve (AUC) for OCRs in Con and *Rictor*[KO] livers from

fed or 14–16 h fasted mice. Fed Con (*n* = 12), fasted Con (*n* = 11), fed *Rictor*[KO] (*n* = 12) and fasted *Rictor*[KO] (*n* = 10) mice. **h**, Acylcarnitine (AC) content in livers of 4–5-month-old Con or *Rictor*[KO] male mice (*n* = 4 mice). **i**, MitoTracker CMXRos fluorescence in serum-deprived and OA-treated siControl (siCon) or si*Rictor* NIH3T3 cells (siCon 59 cells and si*Rictor* 59 cells from *n* = 3 independent experiments). **j**, Representative 3D-TEM images of mitochondria in livers of 4–5-month-old Con or *Rictor*[KO] male mice (*n* = 3 mice). Please refer to Supplementary Video 1 (Con) and Supplementary Video 2 (*Rictor*[KO]). Ponceau is loading control. Individual replicates and means are shown. **P* < 0.05, ***P* < 0.01, ****P* < 0.001 and *****P* < 0.0001, two-tailed unpaired Student's *t*-test (**b**, **h** and **i**); two-way ANOVA followed by Tukey's multiple comparisons test (**c**, **d** and **g**). Please refer to Supplementary Table 10 statistical summary. Source numerical data are available in Source Data Extended Data Table 1, and unprocessed blots are available in the Source Data for this figure.

inactivating mTORC2 (*Rictor*[KO]) markedly increased liver triglycerides and lipid droplet content during fasting (Fig. 2d,e) without affecting circulating FFA (Fig. 2f). Interestingly, *Rictor*[KO] livers displayed lower oxygen consumption rates (OCRs) (Fig. 2g) and accumulation of substrates for mitochondrial respiration, acyl carnitines (Fig. 2h), which in conjunction with reduced mitochondrial membrane potential in si*Rictor* cells (Fig. 2i) indicate mitochondrial insufficiency. Decreased OCR was not due to impaired expression of FFA oxidation or electron transport genes. In fact, fasted *Rictor*[KO] livers displayed increased expression of genes involved in fat oxidation (*Ppara*, *Cpt1a*, *Cpt1b*, *Cact* and *Cpt2*), electron transport (*Cox4*, *Nd1* and *Cytb*) and mitochondrial biogenesis (*Ppargc1a*) (Extended Data Fig. 3d–s). Levels of mitochondrial fatty acid uptake proteins (CPT1A/CPT2/CACT) and electron transport chain components, NDUFB8 (complex I), SDHB (II), UQCRC2 (III), MT-CO1 (IV) and ATP5A (V) were also comparable in control and *Rictor*[KO] livers (Extended Data Fig. 3t,u). Interestingly, loss of *Rictor* led to increased expression of mitochondrial fusion genes *Mfn2* and *Opa1* during fasting without affecting fission genes *Mff* and *Dnm1l* (DRP1) (Extended Data Fig. 3v–z). Consistent with mitochondrial insufficiency, 3D transmission electron microscopy (TEM) revealed mitochondrial distention and blunted networking (Fig. 2j and Supplementary Videos 1 and 2) suggesting that reactivation of mTORC2 during fasting supports mitochondrial dynamics.

### Loss of mTORC2 blocks fasting-induced mitochondrial fission

To determine whether mTORC2 regulates mitochondrial dynamics during fasting, we first determined how fasting impacts mitochondrial dynamics. TEM of fasted livers revealed increased mitochondrial number and higher frequency of mitochondria with reduced area, perimeter and length (Extended Data Fig. 4a–c), reflecting increased fission. Increased mitochondrial number during fasting occurred independently of changes in mitochondrial mass indicated by similar VDAC1 and CYT *c* levels in fed or fasted livers (Extended Data Fig. 4d). By contrast, *Rictor*[KO] livers and livers silenced for *Mff*[26] or *Dnm1l*[27], each failed to increase their mitochondrial number during fasting (Fig. 3a) and displayed increased mitochondrial area and perimeter (Extended Data Fig. 5a–d) indicating fission failure. Consistently, TEM of fasted *Rictor*[KO] livers showed reduced mitochondrial–ER contacts (MAMs) (Fig. 3b), which are contact sites regulating mitochondrial fission[4].

Since perturbations in membrane lipids or ER stress could alter mitochondrial dynamics, we tested if these changes associate with impaired fission in our model. Loss of *Rictor* mildly affected lipid composition of MAMs (Supplementary Table 3) without inducing ER stress or proteostasis failure (Extended Data Fig. 5e), excluding their contribution to impaired fission. Because mTORC2 localizes to MAMs to regulate Ca[2+] homeostasis and apoptosis[28], we envisioned that

mTORC2 at MAMs also regulates fission. Indeed, MAMs from 14–16 h fasted control livers revealed the presence of RICTOR, fission proteins[4] MFF and DRP1, and fusion proteins OPA1 (ref. 29), MFN1 and MFN2 (Fig. 3c). By contrast, MAMs from fasted *Rictor*[KO] livers showed markedly reduced MFF levels without affecting P-DRP1[Ser616], P-DRP1[Ser637], DRP1, MFN1, MFN2 or OPA1 levels (Fig. 3c) supporting that loss of mTORC2 impairs mitochondrial fission.

Confirming the role of mTORC2 in fission, time-lapse microscopy revealed markedly reduced fission rates in si*Rictor* cells that were comparable to fission failure in si*Dnm1l* cells (Fig. 3d,e and Supplementary Videos 3–5). Fusion rates were identical in siCon and si*Rictor* cells, excluding the contribution of excessive fusion to the mitochondrial phenotype in si*Rictor* cells. Furthermore, AML12 and HepG2 hepatocytes silenced for *Rictor/RICTOR*, and *Rictor*[−/−] mouse fibroblasts (Fig. 3f and Extended Data Fig. 5f–i), each showed decreased mitochondrial number and circularity, and markedly increased mitochondrial area and perimeter, demonstrating that mTORC2 drives fission in diverse cell types. Since mTORC1 stimulates mitochondrial fission via the fission protein MTFP1 (ref. 30), we examined whether impaired fission in *Rictor*[KO] livers is due to altered mTORC1 signalling. Loss of *Rictor* did not affect levels of P-P70[Thr389] (mean ± s.e.m.: Con (*n* = 6 mice) versus *Rictor*[KO] (*n* = 10 mice): 0.77 ± 0.17 versus 0.81 ± 0.13; *P* = 0.85) or MTFP1 (Con (*n* = 7 mice) versus *Rictor*[KO] (*n* = 10 mice): 1.00 ± 0.20 versus 1.06 ± 0.12; *P* = 0.80) or affect levels of DRP1 or DRP1[Ser616] and DRP1[Ser637] phosphorylation (Extended Data Fig. 5j), which regulate mitochondrial dynamics[31]. Hence, reactivation of mTORC2 during fasting drives fission, independent of mTORC1 signalling.

### mTORC2–SGK1 phosphorylates NDRG1 at Ser336

To determine whether phosphorylated targets of mTORC2 (ref. 32) support fission, we used quantitative nano liquid chromatography coupled online with tandem mass spectrometry (nLC–MS/MS) in control and *Rictor*[KO] livers (Fig. 4a), which identified 4,553 phosphosites from 1,712 phosphoproteins. Of these, 309 phosphosites (145 upregulated and 164 downregulated) (6.79%) on 212 phosphoproteins (12.38%) were significantly modulated in *Rictor*[KO] livers (Extended Data Fig. 6a and Supplementary Table 4). Gene Ontology and enrichment map network analysis[33] revealed that the hypophosphorylated clusters in *Rictor*[KO] livers were related to cytoskeleton and cellular architecture, mRNA processing and splicing, protein targeting and regulation of cellular catabolic processes (Fig. 4b and Supplementary Tables 2 and 5). The denser cluster populated by both upregulated and downregulated phosphoproteins contained the term 'regulation of metabolism'. Since protein function is modulated by site-specific phosphorylation or cumulative phosphorylation of multiple phosphosites[34], we measured the overall phosphorylation status (ΔPs) of

**Fig. 3 | Loss of mTORC2 blocks fasting-induced mitochondrial fission.**
**a**, Conventional TEM in 14–16 h fasted Con, *Rictor*[KO], si*Mff* or si*Dnm1l* livers of 4–9-month-old male mice. *N* values for number of mice per group are indicated in parentheses. Con or *Rictor*[KO] mice (*n* = 6), and si*Mff* or si*Dnm1l* mice (*n* = 4). Quantification for mitochondrial number is shown. **b**, TEM in fed or 14–16 h fasted Con and *Rictor*[KO] livers of 4–9-month-old male mice. Fed Con (*n* = 7), fasted Con (*n* = 9) and fed or fasted *Rictor*[KO] mice (*n* = 5). Quantification for percentage of mitochondria–ER contacts is shown. Red arrowheads depict contact sites. **c**, Immunoblots and quantification of indicated proteins in homogenates (Hom), pure mitochondria (Mp), MAMs, cytosol (Cyt) and ER fractions from 14–16 h fasted Con (*n* = 3) and *Rictor*[KO] livers (*n* = 5). Two livers were pooled to generate one sample. **d**, Live cell imaging and quantification for fission and fusion rates in siCon, si*Rictor* or si*Dnm1l* NIH3T3 cells cultured in serum-free medium for 30 min in presence of MitoTracker green to stain for mitochondria (siCon 12 cells, si*Rictor* 11 cells (fission rate) and *n* = 12 cells (fusion rate), and si*Dnm1l* 11 cells from *n* = 8 independent experiments; each cell was tracked on an independent plate). White arrowheads depict mitochondrial constriction sites. Yellow arrowheads depict daughter mitochondria arising from fission at

a mitochondrial constriction. Scale bar, 2 μm. Please refer to Supplementary Videos 3 (siCon cells), 4 (si*Rictor* cells) and 5 (si*Dnm1l* cells). **e**, Representative IB for indicated proteins in siCon, si*Dnm1l* and si*Rictor* NIH3T3 cells. Blots are representative of *n* = 8 (DRP1) and *n* = 4 (RICTOR) independent experiments obtaining similar results. **f**, Representative confocal images of (top) AML12 and (bottom) HepG2 cells knocked down for *Rictor*, and corresponding controls in serum-free medium for 30 min in presence of MitoTracker green to stain for mitochondria. Magnified insets are shown. Quantifications for mitochondrial number and mitochondrial size/shape descriptors (area, perimeter and circularity) are shown (AML12 45 siCon or si*Rictor* cells; HepG2 35 siCon and 38 si*RICTOR* cells from *n* = 3 independent experiments each). Ponceau is loading control. Individual replicates and means are shown. *\*P* < 0.05, *\*\*P* < 0.01 and *\*\*\*\*P* < 0.0001, one-way ANOVA followed by Tukey's multiple comparisons test (**a** and **d**); two-way ANOVA followed by Tukey's multiple comparisons test (**b**); two-tailed unpaired Student's *t*-test (**c** and **f**). Please refer to Supplementary Table 10 statistical summary. Source numerical data are available in Source Data Extended Data Table 1, and unprocessed blots are available in the Source Data for this figure.

phosphoproteins in our dataset, which revealed hyperphosphorylation ($\Delta$Ps > $2\sigma$) in 37 phosphoproteins and hypophosphorylation ($\Delta$Ps < $-2\sigma$) in 49 phosphoproteins (Fig. 4c). Interestingly, phosphorylation of

mitophagy receptor BNIP3 at Ser79 and Ser88 was significantly reduced in $Rictor^{KO}$ livers (Fig. 4d and Extended Data Fig. 6b) with no known roles assigned to BNIP3$^{Ser79/Ser88}$. We also focused on NDRG1 (Fig. 4e), which

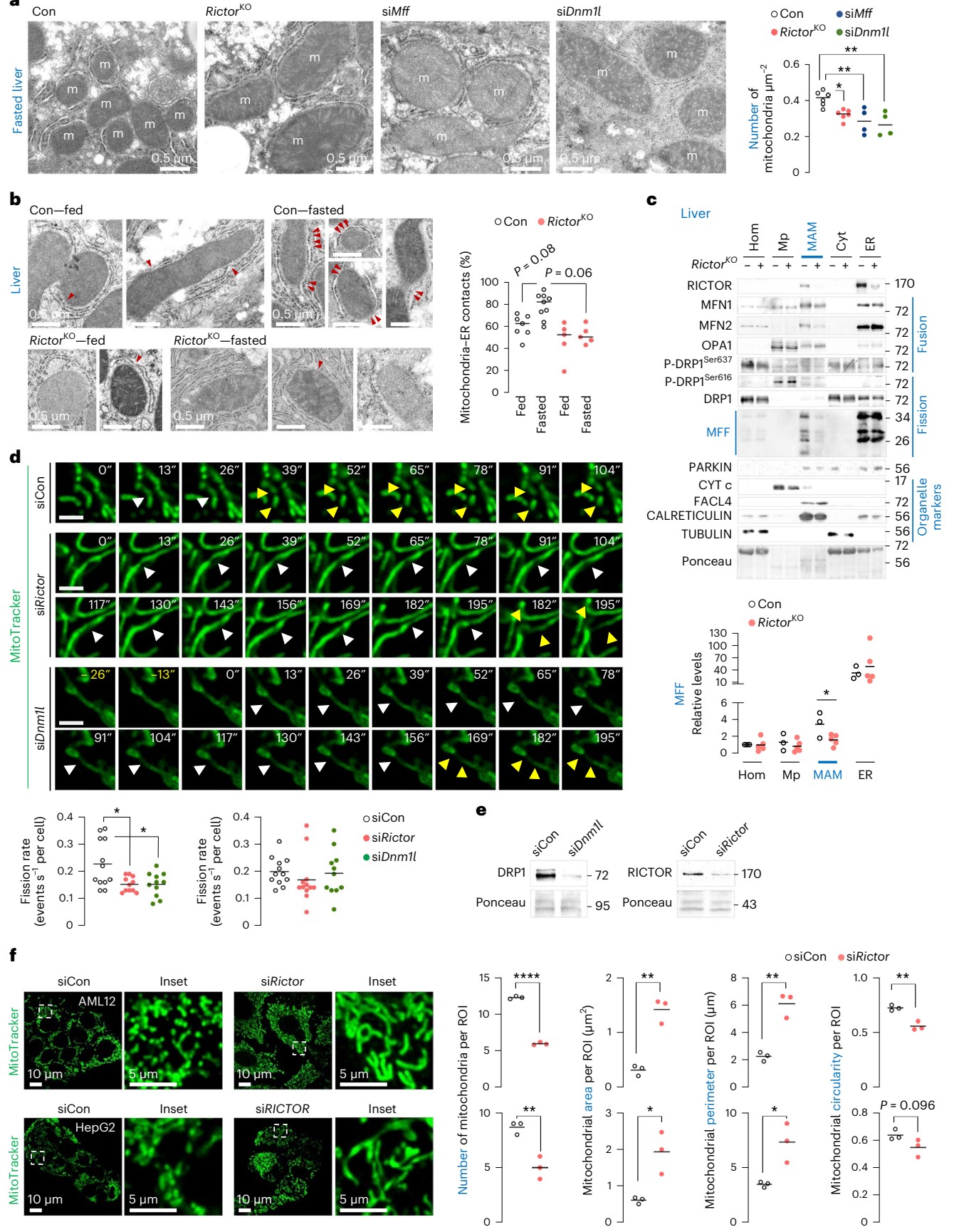

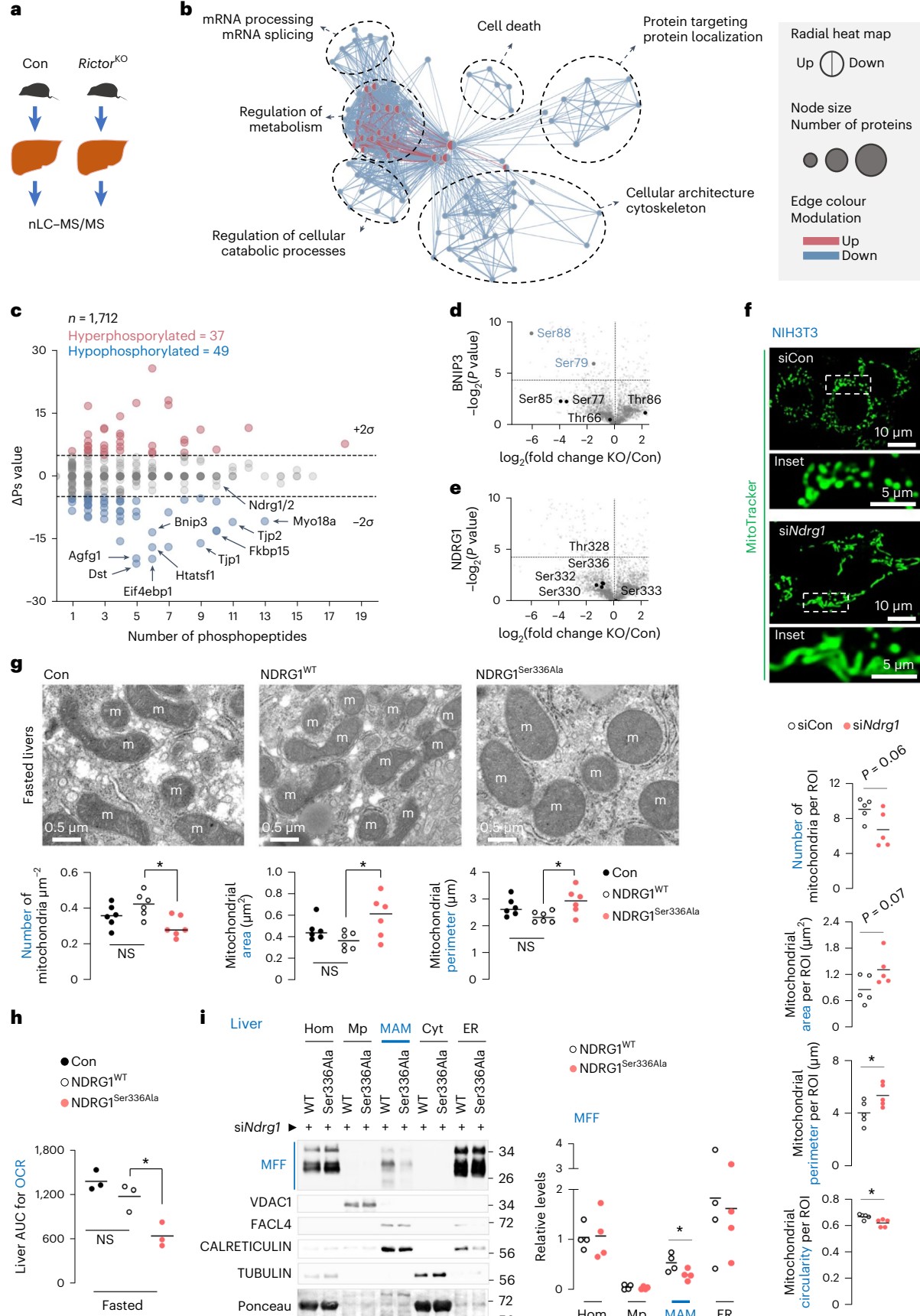

is phosphorylated on its C-terminus by the mTORC2 target SGK1 (refs. 8,35), and regulates lipid droplet content[36]. Although NDRG1 showed trends towards hypophosphorylation in *Rictor*[KO] total homogenates

(Fig. 4e and Extended Data Fig. 6c), phosphoproteomics from fasted livers showed marked NDRG1[Ser336] hypophosphorylation in *Rictor*[KO] MAMs when compared with controls (Extended Data Fig. 6d–f and

**Fig. 4 | mTORC2 signalling drives mitochondrial fission via NDRG1[Ser336] phosphorylation. a**, Experimental plan for **b**–**e**. **b**, Enrichment map-based network visualization of Gene Ontology enrichment for differentially modulated phosphosites. Blue edges show similarity between decreased phosphosites, and red nodes show similarity between increased phosphosites. Node size indicates the number of proteins per node; major clusters are circled. Associated name represents the major functional association. **c**, Global ΔPs analyses of phosphoproteins. Hyperphosphorylated and hypophosphorylated peptides in each comparison are shown. Labels indicate the genes encoding the proteins. Dotted lines: ΔPs = ±2σ. **d**,**e**, Volcano plot for BNIP3 (**d**) or NDRG1 (**e**) phosphorylation in *Rictor*[KO] versus Con livers fasted for 14–16 h. For **a**–**e**, *n* = 3 mice. **f**, Representative MitoTracker green fluorescence in serum-deprived siCon and si*Ndrg1* NIH3T3 cells (siCon 84 cells and si*Ndrg1* 91 cells

from *n* = 5 independent experiments). Quantifications for mitochondrial number and mitochondrial size/shape descriptors are shown. **g**–**i**, TEM with mitochondrial quantifications (*n* = 6 mice) (**g**), AUC for liver OCR (*n* = 3 mice) (**h**) and immunoblots and quantification (**i**) for indicated proteins in indicated fractions from livers of 3–4-month-old fasted (14–16 h) male mice expressing NDRG1[WT] or NDRG1[Ser336Ala] after silencing endogenous *Ndrg1* by siRNAs (*n* = 4 mice). Ponceau is loading control. Individual replicates and means are shown. *\*P* < 0.05, two-tailed unpaired Student's *t*-test (**c**–**f** and **i**); one-way ANOVA followed by Tukey's comparisons test (**g** and **h**). NS, not significant. Please refer to Supplementary Table 10 statistical summary, and Supplementary Tables 4 and 5. Source numerical data are available in Source Data for Extended Data Table 1, and unprocessed blots are available in the Source Data for this figure.

Supplementary Table 6). Since NDRG1 is present in MAMs (Extended Data Fig. 6g), we sought to confirm that mTORC2–SGK1 indeed phosphorylates NDRG1 at Ser336. Accordingly, phosphoproteomics and relative quantification of extracted ion chromatogram of peptide SRTASGSSVTS(p)LEGTRSR, corresponding to Flag–NDRG1, from siCon or si*Rictor* cells (Extended Data Fig. 6h–j and Supplementary Table 7) revealed reduced enrichment in si*Rictor* cells compared with siCon cells, confirming that mTORC2 phosphorylates NDRG1 at Ser336.

### Phosphorylated NDRG1[Ser336] drives mitochondrial fission

To determine whether mTORC2 drives fission by phosphorylating BNIP3 at Ser79 or Ser88 or NDRG1 at Thr328, Ser332 or Ser336, we used in vitro Seahorse-based mito-stress screens. We expressed Flag-tagged phosphorylation-deficient or phosphomimetic BNIP3 or NDRG1 mutants by switching these Ser or Thr residues to Ala or Asp, respectively (Extended Data Fig. 7a,b). Serum-deprived and oleic acid (OA)-treated cells (to emulate fasting) expressing BNIP3[WT] (wild type, WT) or BNIP3[Ser79Ala] or BNIP3[Ser88Ala] showed equivalent mitochondrial respiration (Extended Data Fig. 7c), eliminating that P-BNIP3[Ser79/Ser88] regulates mitochondrial function. Expressing NDRG1[Thr328Ala], NDRG1[Ser332Ala], NDRG1[Thr328Asp] or NDRG1[Ser332Asp] mutants also failed to impact respiration; however, blocking NDRG1[Ser336] phosphorylation reduced basal and maximal respiration and ATP production (Extended Data Fig. 7d–g), while phosphomimetic NDRG1[Ser336Asp] stimulated respiration compared with NDRG1[WT] (Extended Data Fig. 7f,h). Furthermore, silencing *Ndrg1* (Extended Data Fig. 7i,j) or expressing each phosphorylation-deficient NDRG1 mutant (Extended Data Fig. 7k) substantially lowered mitochondrial membrane potential, suggesting that mTORC2 supports mitochondrial function via NDRG1[Ser336] phosphorylation. Indeed, the mTORC2–SGK1 axis mediates fission via NDRG1[Ser336] phosphorylation, since silencing *Sgk1-3* or *Ndrg1*, but not *Akt1/2*, significantly reduced mitochondrial number and increased

mitochondrial area, perimeter and elongation (Fig. 4f and Extended Data Fig. 8a) as observed in si*Dnm1l*[37] or si*Mff*[3] cells. Consistently, silencing *Sgk1* and *Ndrg1*, but not *Akt1/2*, reduced cellular respiration in vivo (Extended Data Fig. 8b,c).

### NDRG1[Ser336Ala] mutant livers exhibit fission failure

To determine whether NDRG1[Ser336Ala] mutant livers recapitulate the mitochondrial phenotype of *Rictor*[KO] livers, we expressed Flag–NDRG1[WT] or Flag–NDRG1[Ser336Ala] in livers silenced for endogenous *Ndrg1* and confirmed equivalent Flag expression by immunohistochemistry (Extended Data Fig. 8d). Consistent with our observations in *Rictor*[KO] livers, fasted NDRG1[Ser336Ala] livers showed enlarged mitochondria with reduced mitochondrial number, and increased area and perimeter when compared with NDRG1[WT] and untransfected livers (Con) (Fig. 4g), reflecting impaired fission. As observed in *Rictor*[KO] livers, fasted NDRG1[Ser336Ala] livers showed reduced cellular respiration (Fig. 4h). Furthermore, when compared with corresponding controls, MAMs from *Rictor*[KO] (Fig. 3c) and NDRG1[Ser336Ala] livers (Fig. 4i), each showed lower levels of MFF without affecting levels of total and phosphorylated DRP1[Ser616] and DRP1[Ser637], which modulate dynamics[31] (Fig. 3c). Hence, our data support a role for the mTORC2–NDRG1[Ser336] axis in driving mitochondrial fission.

### NDRG1 requires MFF, but not DRP1, for mitochondrial fission

To determine how NDRG1 facilitates fission, we used time-lapse microscopy to test if NDRG1[WT] interacts with mitochondria. Interestingly, NDRG1[WT] frequently co-localized with a constricted region of mitochondria, culminating in fission (Fig. 5a–c and Supplementary Video 6). Quantifications revealed that, while NDRG1[WT]–mitochondrial interactions caused fission within ~90.8 ± 11.1 s of contact (Fig. 5a–c and Supplementary Video 6), NDRG1[Ser336Ala] maintained its co-localization with ER (Extended Data Fig. 8e–g and Supplementary Videos 7 and

**Fig. 5 | Phosphorylated NDRG1[Ser336] requires MFF, but not DRP1, for mitochondrial fission. a**, Live cell imaging of mCherry–NDRG1[WT] or mCherry–NDRG1[Ser336Ala] and MitoTracker green in siCon NIH3T3 cells or live cell imaging of mCherry–NDRG1[WT] and MitoTracker in si*Rictor* or si*Dnm1l* NIH3T3 cells cultured in serum-free medium for 30 min. Orange arrowheads: NDRG1 (mCherry). White arrowheads: mCherry/MitoTracker contact reflecting NDRG1/mitochondria contact before fission. Yellow arrowheads: divided mitochondria after scission by NDRG1 (mCherry). Magnified insets are shown. Please refer to Supplementary Video 6 (mCherry–NDRG1[WT]/MitoTracker), Supplementary Video 9 (mCherry–NDRG1[Ser336Ala]/MitoTracker), Supplementary Video 10 (si*Rictor*; mCherry–NDRG1[WT]/MitoTracker) and Supplementary Video 11 (si*Dnm1l*; mCherry–NDRG1[WT]/MitoTracker). **b**, Quantification for duration/fate (fission versus no fission) of interaction between NDRG1 (mCherry) and mitochondria (MitoTracker). Quantifications are also shown for duration of NDRG1 (mCherry)-mitochondrial (MitoTracker) interaction events and whether each interaction led to fission (useful) or not (futile) as recorded via live cell imaging. The *X* axis represents time in seconds–reflecting duration of contact of NDRG1 (mCherry)

with mitochondria (MitoTracker). Each individual-coloured bar on the *Y* axis represents one interaction per individual cell. The length of each coloured bar represents the time from the initiation of interaction of NDRG1 (mCherry) with mitochondria (MitoTracker) till end of interaction. **c**, Quantification for mean duration of mCherry–NDRG1/mitochondria (MitoTracker) interaction is shown. For **b** and **c** (NDRG1[WT] 11 cells, NDRG1[Ser336Ala] 15 cells, si*Rictor*/NDRG1[WT] 10 cells and si*Dnm1l*/NDRG1[WT] 9 cells from *n* = 4 independent experiments; each tracked cell was monitored on an independent plate). **d**,**e**, Representative images of si*Dnm1l* (*n* = 4 independent experiments) (**d**) or si*Mff* (*n* = 3 independent experiments) (**e**) NIH3T3 cells expressing mCherry–NDRG1[WT] or not and cultured in serum-free medium for 30 min in presence of MitoTracker green. Magnified insets are shown. Quantifications for mitochondrial number and mitochondrial size/shape descriptors are shown. Individual replicates and means are shown. *\*P* < 0.05, *\*\*P* < 0.01, *\*\*\*P* < 0.001 and *\*\*\*\*P* < 0.0001, one-way ANOVA followed by Tukey's multiple comparisons test. Please refer to Supplementary Table 10 statistical summary. Source numerical data are available in Source Data to Extended Data Table 1.

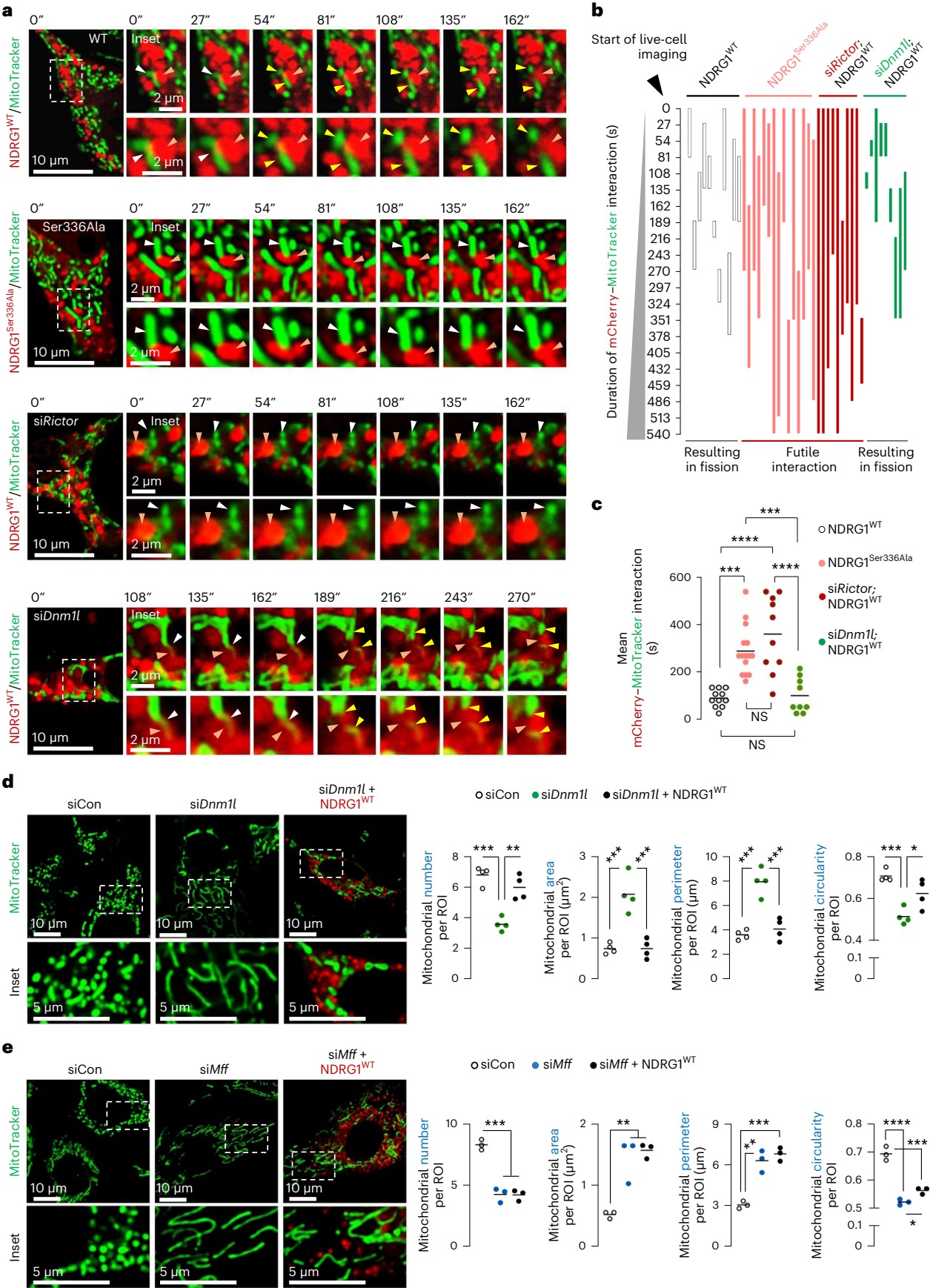

8) but exhibited extended futile interactions (~289.8 ± 26.7 s) with mitochondria that did not lead to fission (Fig. 5a–c and Supplementary Video 9). Consistently, silencing *Rictor* led to extended futile interactions of NDRG1$^{WT}$ with mitochondria (409.5 ± 63.7 s) and blocked the ability of NDRG1$^{WT}$ to divide mitochondria (Fig. 5a–c and Supplementary Video 10), linking mTORC2-driven NDRG1$^{Ser336}$ phosphorylation to mitochondrial fission. To determine whether NDRG1-mediated fission requires DRP1, we attempted to KO *Dnm1l* using CRISPR, but failed to generate viable healthy cells, and therefore this limit in interpretation remains. However, upon using small interfering RNAs (siRNAs) to deplete *Dnm1l*, mitochondrial division via NDRG1$^{WT}$ remained intact in si*Dnm1l* cells (Fig. 5a–c and Supplementary Video 11). Indeed, despite >90% loss of *Dnm1l*, NDRG1$^{WT}$ continued to engage with mitochondria resulting in fission in ~102 ± 24.6 s. In fact, expressing NDRG1$^{WT}$ completely restored the altered mitochondrial number, area, perimeter and circularity in si*Dnm1l* cells (Fig. 5d and Extended Data Fig. 9a). By contrast, NDRG1$^{WT}$ failed to restore the alterations in mitochondrial number, area, perimeter and circularity in si*Mff* cells (Fig. 5e and Extended Data Fig. 9b) suggesting that, although DRP1 is a key regulator of fission, it appears to not influence mitochondrial fission via the mTORC2–NDRG1 axis. By contrast, the mTORC2–NDRG1 axis requires MFF for fission, supported by data showing reduced MFF enrichment in MAMs from *Rictor*$^{KO}$ (Fig. 3c) and NDRG1$^{Ser336Ala}$ livers (Fig. 4i) and that si*Mff* cells resist NDRG1$^{WT}$-mediated fission (Fig. 5e).

### Phosphorylated NDRG1$^{Ser336}$ binds CDC42 to drive fission

Since NDRG1 does not exhibit the intrinsic GTPase activity required for membrane scission[38,39], we asked whether P-NDRG1$^{Ser336}$ engages with proteins with intrinsic GTPase activity to facilitate fission. Accordingly, proteomics to identify proteins bound to Flag-tagged NDRG1$^{WT}$, but not NDRG1$^{Ser336Ala}$, revealed interaction with CDC42, a RHO GTPase that regulates actin cytoskeleton[40] and cytokinesis[9] (Fig. 6a and Supplementary Table 8). Indeed, when compared with NDRG1$^{WT}$, NDRG1$^{Ser336Ala}$ displayed reduced, albeit modest, binding to CDC42, ARHGEF10 (RHO GEF that activates RHO GTPases by stimulating GDP/GTP exchange)[41], and ARHGAP35 (RHO GAP that facilitates GTP hydrolysis to inactivate RHO GTPases)[41]. Consequently, we hypothesized that GTPase CDC42 mediates the effects of mTORC2–NDRG1$^{Ser336}$ on mitochondrial fission. Supporting this hypothesis, co-immunoprecipitation (co-IP) confirmed that exogenously expressed GFP–CDC42 interacts with Flag–NDRG1$^{WT}$ but fails to interact with mutant NDRG1$^{Ser336Ala}$ (Fig. 6b). Furthermore, Flag–NDRG1$^{WT}$ interacts with both mCherry–CDC42$^{WT}$ and mutant CDC42$^{Thr17Asn}$, which is expected to be predominantly GDP-bound (Extended Data Fig. 9c), indicating that CDC42–GTP loading is not required for CDC42–NDRG1 interaction. However, NDRG1$^{Ser336}$ phosphorylation and CDC42–GTP loading are each critical

for mitochondrial fission. Indeed, silencing *Cdc42* (Extended Data Fig. 9d) led to prolonged and futile engagement of NDRG1$^{WT}$ with mitochondria and blocked fission (Fig. 6c–e and Supplementary Videos 12 and 13). Furthermore, while CDC42$^{WT}$ engaged with, and divided, mitochondria in ~157 ± 26 s, mutant CDC42$^{Thr17Asn}$ exhibited prolonged (~339 ± 58 s) and futile interactions with mitochondria (Extended Data Fig. 9e–g and Supplementary Videos 14 and 15). Supporting that the mTORC2–NDRG1–CDC42 axis drives fission, knocking down *Rictor* or *Ndrg1* or *Cdc42* each reduced mitochondrial number and circularity, and increased mitochondrial area, perimeter and elongation (Fig. 6f), recapitulating the fission failure phenotype of si*Mff* or si*Dnm1l* cells (Fig. 6f). Consistently, silencing *Cdc42* reduced mitochondrial membrane potential (Fig. 6g), suggesting that CDC42 cooperates with P-NDRG1$^{Ser336}$ to support mitochondrial division.

### CDC42 regulators and effectors modulate fission

Since CDC42 activity is tightly orchestrated by regulators or effectors, we used proteomics to identify CDC42 interactors that may potentially regulate fission (Fig. 7a and Supplementary Table 9). Using NIH3T3 cells expressing GFP–CDC42 or GFP–empty vector as negative control and analysing fold-change interaction (using cut-off *P* value of 4.32), we short-listed 11 proteins that were significantly enriched in GFP–CDC42 pulldowns when compared with empty vector (Fig. 7b and Extended Data Fig. 10a). Of note, the five enriched targets were CDC42 effectors (CDC42EP4/BORG4, CDC42EP1/BORG5 and CDC42EP2/BORG21), RHO GTPase inhibitor RHO GDP Dissociation Inhibitor (GDI) alpha (ARHGDIA/RHOGDI1) and IQ motif-containing GTPase-activating protein 1 (IQGAP1), which is a downstream effector and upstream scaffold protein for CDC42 (ref. 42) (Fig. 7b and Extended Data Fig. 10a). We also observed enrichment (albeit insignificant) in GFP–CDC42 pulldowns of a known driver of CDC42 in muscle cells, bridging integrator 3 (BIN3), the atypical RHO GTPase/RHO-Related BTB Domain Containing 1 (RHOBTB1) and RHO GTPase inhibitor beta (ARHGDIB/RHOGDI2) (Fig. 7b and Extended Data Fig. 10a). In addition, we included to screen for two identified NDRG1 binding partners, ARHGEF10 and ARHGAP35, which serve as GEF and GAP[41], respectively (Fig. 6a).

To examine if the identified candidates regulate mitochondrial fission, we transfected NIH3T3 cells with siRNAs against each target (Extended Data Fig. 10b), except *Cdc42ep2* since silencing it severely reduced viability. Our results indicate that CDC42 and its family of effectors/regulators control mitochondrial fission, since deleting *Cdc42* or CDC42 activators, *Arhgef10* and *Bin3*, or CDC42 downstream effectors, *Cdc42ep4*/BORG4 or *Cdc42ep1*/BORG5, each resulted in increased mitochondrial area, perimeter and elongation, reflecting impaired fission (Fig. 7c). RHOGDI proteins can act as negative regulators of RHO GTPases by retaining RHO GTPases in the cytosol, inhibiting

**Fig. 6 | Phosphorylated NDRG1$^{Ser336}$ interacts with CDC42 to drive mitochondrial fission. a**, The log$_2$-transformed fold change (FC) of interaction of Flag–NDRG1$^{WT}$ or Flag–NDRG1$^{Ser336Ala}$ with CDC42, ARHGAP35 and ARHGEF10 (*n* = 3 independent experiments). **b**, Pulldowns of Flag (using Flag M2 agarose) and immunoblots and quantification for GFP and Flag levels in OA-treated (2.5 h) NIH3T3 cells co-expressing Flag–NDRG1$^{WT}$ or Flag–NDRG1$^{Ser336Ala}$ and GFP–CDC42$^{WT}$ (*n* = 3 independent experiments). Flag-tagged empty vector is the negative control. Ponceau is loading control. Quantification for relative enrichment of GFP in Flag pulldowns was calculated by normalizing the densitometric value of GFP to the densitometric value of Flag. **c**, Representative live-cell imaging of mCherry–NDRG1$^{WT}$ and MitoTracker green in siCon or si*Cdc42* NIH3T3 cells. Magnified insets are shown. Orange arrowhead: NDRG1 (mCherry) mediating fission. White arrowhead: mCherry/MitoTracker reflecting NDRG1/mitochondrial co-localization before fission. Yellow arrowheads: divided mitochondria after fission. Please refer to Supplementary Video 12 (siCon cells; mCherry–NDRG1$^{WT}$) and Supplementary Video 13 (si*Cdc42* cells; mCherry–NDRG1$^{WT}$). **d**, Graphical representation for duration of interaction between mCherry–NDRG1 and mitochondria (MitoTracker) in siCon or si*Cdc42* cells, and whether interactions lead to division or are futile. **e**, Quantification for

mean duration of mCherry–NDRG1$^{WT}$/mitochondria (MitoTracker) interaction is shown (siCon 6 cells and si*Cdc42* 11 cells from *n* = 3 independent experiments; each tracked cell was monitored on an independent plate). **f**, Representative confocal images of NIH3T3 cells transfected with indicated siRNAs and cultured in serum-free medium with MitoTracker green for 30 min. Magnified insets are shown. Quantifications for mitochondria number and mitochondrial size/shape descriptors are shown (siCon 142 cells, si*Rictor* 105 cells, si*Ndrg1* 91 cells, si*Cdc42* 64 cells, si*Mff* 106 cells, si*Dnm1l* 128 cells, si*Opa1* 86 cells and si*Mfn1* 83 cells from *n* = 8 (siCon), *n* = 6 (si*Rictor* and si*Mff*), *n* = 5 (si*Ndrg1*, si*Opa1* and si*Mfn1*), *n* = 4 (si*Cdc42*) and *n* = 7 (si*Dnm1l*) independent experiments. Grey areas indicate mitochondria fission-deficient models. **g**, MitoTracker CMXRos fluorescence in siCon and si*Cdc42* cells cultured in serum-free medium in presence of OA for 5 h (siCon 102 cells and si*Cdc42* 116 cells from *n* = 3 independent experiments). Individual replicates and means are shown. *\*P* < 0.05, *\*\*P* < 0.01, *\*\*\*P* < 0.001 and *\*\*\*\*P* < 0.0001, two-tailed unpaired Student's *t*-test (**a**, **b**, **e** and **g**); one-way ANOVA and Dunnett's multiple comparisons test (**f**). Please refer to Supplementary Table 10 statistical summary, and Supplementary Table 8. Source numerical data are available in Source Data Extended Data Table 1, and unprocessed blots are available in the Source Data for this figure.

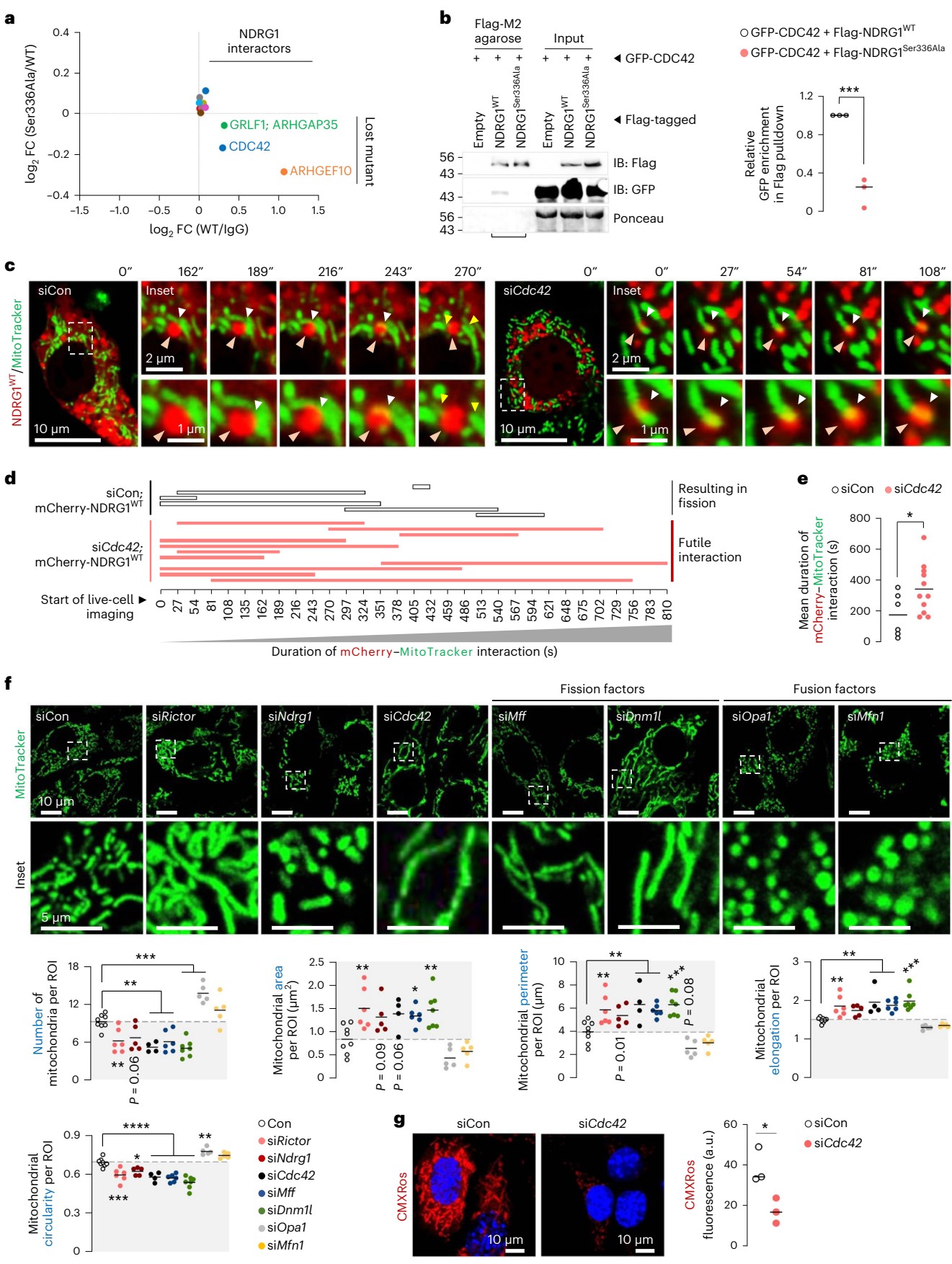

their GTPase activity, and preventing their interaction with GEFs, GAPs and effectors[43,44]. Accordingly, we suspect that silencing *Arhgdia*/RHOGDI1 releases CDC42 from the inhibitory effect of ARHGDIA/RHOGDI1, leading to fission. Consistently, silencing *Arhgdia*/RHOGDI1 (but not *Arhgdib*/RHOGDI2) increased mitochondrial number (Fig. 7c), reflecting increased fission. Not all RHO GTPases impact mitochondrial dynamics, since depleting the RHO GTPase *Rhobtb1* gene had no effect on mitochondrial morphology, suggesting specificity of RHO GTPase CDC42 towards fission. We also found that silencing IQGAP1, which regulates CDC42 as an upstream scaffold and as a downstream effector of CDC42 (ref. 42), increased mitochondrial number (Fig. 7c). Indeed, as a scaffold protein, IQGAP1 provides a molecular link between Ca²⁺/calmodulin and CDC42-mediated processes[45], while as a downstream effector, CDC42 enhances the F-actin-cross-linking activity of IQGAP1 during actin reorganization[46]. Since ARHGAP35 inactivates GTPases, we anticipated that depleting *Arhgap35* would stimulate CDC42, leading to fission; however, knocking down *Arhgap35* decreased mitochondrial number, reflecting fission failure (Fig. 7c). This probably reflects the complex regulation of CDC42 requiring subsequent inactivation to complete its function[47,48], as well as specificity among the different effectors and regulators in stimulating fission. Alternatively, ARHGAP35 is perhaps not active towards CDC42 and affects, instead, an antagonistic GTPase. Consistent with these findings, in addition to CDC42, we detected the presence in MAMs of CDC42 effector, CDC42EP1/BORG5, and ARHGAP35 and ARHGDIA/RHOGDI1 (Extended Data Fig. 10c,d). Interestingly, levels of ARHGAP35, CDC42EP1/BORG5 and ARHGDIA/RHOGDI1 in MAMs from *Rictor*^KO (Extended Data Fig. 10c) and NDRG1^Ser336Ala expressing livers (Extended Data Fig. 10d) were comparable to those in corresponding controls, indicating that fission is probably regulated at the level of recruitment of CDC42 to MAMs.

Since CDC42 action is regulated by membrane binding[49] and GTP loading[50], we suspect that mTORC2-driven NDRG1^Ser336 phosphorylation is a signal to recruit CDC42 to MAMs for its activation to drive fission. Indeed, CDC42 and RHOA[51] (Fig. 7d), but not dynamins, were abundantly present in MAMs from fasted livers. By contrast, MAMs from fasted *Rictor*^KO (Fig. 7d) and NDRG1^Ser336Ala livers (Fig. 7e) each showed markedly reduced CDC42 levels without affecting RHOA levels, suggesting that the mTORC2–NDRG1 axis recruits CDC42 to MAMs. Given the enrichment of CDC42 in MAMs in an mTORC2- and NDRG1-sensitive manner, it is likely that CDC42 governs local downstream mechanisms that control fission. Since dynamic cycling of actin through populations of mitochondria controls fission[52,53], we asked whether CDC42 mediates the effect of mTORC2–NDRG1 on mitochondrial fission by remodelling actin cytoskeleton. Indeed, in control cells, actin assembled around mitochondria to generate ring-like structures consistent with maintained fission[53] (Fig. 7f). By contrast, silencing *Cdc42* decreased co-localization of actin with mitochondria, which

correlated with elongated mitochondria (Fig. 7f), suggesting that CDC42 facilitates the organization of actin around mitochondria to enable fission.

## Discussion

In sum, we show that the typically nutrient-responsive mTORC2 is paradoxically reactivated by fasting to regulate NDRG1^Ser336 phosphorylation, which serves to recruit CDC42 to MAMs to drive mitochondrial fission. In support of this model (Fig. 7g), NDRG1 engages with mitochondrial constrictions to facilitate fission, and that fission events are blocked in cells expressing phosphorylation-deficient NDRG1^Ser336Ala or in cells lacking *Rictor* or *Cdc42* or the identified CDC42 effector/regulators (Fig. 7g), thus revealing an mTORC2–NDRG1–CDC42 axis facilitating mitochondrial fission during fasting.

Fasting and feeding are hormonally distinct physiological states[19]. While nutrient deprivation in cultured cells blocks mitochondrial fission to preserve ATP synthesis and cell viability[54,55], cultured cells do not completely recapitulate the complex physiology of intact organisms. In fact, we show that the highly integrated liver exhibits marked increases in fission during an acute fast. Indeed, switching between nutrient availability and deprivation modulates mitochondrial cristae and ER contacts, which per se impact mitochondrial dynamics[56,57]. In keeping with this, we suspect that fasting-induced increases in adipose lipolysis and increased availability of lipids reactivate mTOR during fasting. Indeed, cholesterol[58] and phosphatidic acid[59] activate mTORC1 in vitro, and we show here that exposure to dietary corn oil or fasting each activates mTORC2 in liver, as has been shown for mTORC1 in starved cultured cells[60]. Although no function has been assigned to fasting-induced reactivation of mTOR, we demonstrate that paradoxical reactivation of mTORC2 during fasting is required for mitochondrial remodelling to possibly support the increased energetic demands of fasting. In fact, enzymatic reactions, for example, those part of the Krebs cycle, appear to be sensitive to changes in mitochondrial shape, volume and connectivity[61]. Consistently, not only does loss of *Rictor* impact mitochondrial fission, we also noted marked accumulation of acylcarnitines, a metabolic signature consisted with dampened mitochondrial respiration.

Mitochondrial division is tightly orchestrated by recruitment of dynamin-related GTPase DRP1 from the cytosol to MAMs by the mitochondrial outer membrane receptor MFF[4]. DRP1 oligomerization and interaction with actin filaments promote scission in a GTP hydrolysis-dependent manner. Indeed, overexpression of MFF fails to restore fission in cells co-expressing the assembly-defective DRP1 mutant, indicating that MFF acts via DRP1 (ref. 4). Yet, our data show that DRP1 is dispensable for mTORC2–NDRG1-mediated mitochondrial fission, since overexpressing NDRG1^WT restores mitochondrial fission in DRP1-deficient cells, and perhaps most surprisingly, NDRG1^WT fails to restore fission when MFF is depleted. These data suggest that

---

**Fig. 7 | CDC42 regulators and effectors modulate fission. a**, Experimental plan to pull down GFP-tagged CDC42 to identify interactors that regulate mitochondrial fission. **b**, Cartoon representing significantly ($P < 0.05$) enriched interacting partners of CDC42 (in bold) identified via proteomics, some of which belong to the RHO family of GTPases ($n = 4$ independent experiments). **c**, Representative images of NIH3T3 cells knocked-down for indicated CDC42-binding partners and cultured in serum-free medium for 30 min in presence of MitoTracker green. Magnified insets are shown. Quantifications for mitochondrial number and mitochondrial size/shape descriptors are shown (siCon, si*Arhgef10*, si*Iqgap*, si*Arhgdib*, si*Rhobtb1* and si*Arhgap35* 39 cells, si*Bin3* and si*Cdc42ep4* 40 cells, si*Cdc42ep1* 43 cells and si*Arhgdia* 37 cells from $n = 3$ independent experiments). Grey areas indicate mitochondria fission-deficient models. **d,e**, Immunoblots and quantification for indicated proteins in indicated fractions from livers of 14–16 h-fasted 5–6-month-old Con ($n = 3$) and *Rictor*^KO ($n = 5$) mice (**d**), and 3–4-month-old mice expressing NDRG1^WT or NDRG1^Ser336Ala plasmids after silencing endogenous *Ndrg1* with siRNAs (**e**).

*N* values for number of mice per fraction are indicated in parentheses. CDC42: all fractions from NDRG1^WT and NDRG1^Ser336Ala mice are each $n = 5$ except for NDRG1^Ser336Ala MAMs where $n = 4$; RHOA: $n = 6$ mice for all groups. Ponceau is loading control. **f**, Representative confocal images of siCon or si*Cdc42* NIH3T3 cells expressing mCherry–Lifeact-7 and cultured in serum-free medium for 30 min in presence of MitoTracker green. Magnified insets are shown. Quantification for percentage co-localization of mCherry–Lifeact-7 with mitochondria is shown (siCon 33 cells and si*Cdc42* 30 cells from $n = 5$ (siCon) or $n = 4$ (si*Cdc42*) independent experiments). **g**, Reactivation of mTORC2 during fasting phosphorylates NDRG1 at Ser336, which engages with mitochondria and recruits CDC42 to mitochondria–ER contact sites wherein CDC42 and its effector proteins orchestrate fission. Individual replicates and means are shown. *$P < 0.05$ and **$P < 0.01$, two-tailed unpaired Student's *t*-test. NS, not significant. Please refer to Supplementary Table 10 statistical summary, and Supplementary Table 9. Source numerical data are available in Source Data Extended Data Table 1, and unprocessed blots are available in the Source Data for this figure.

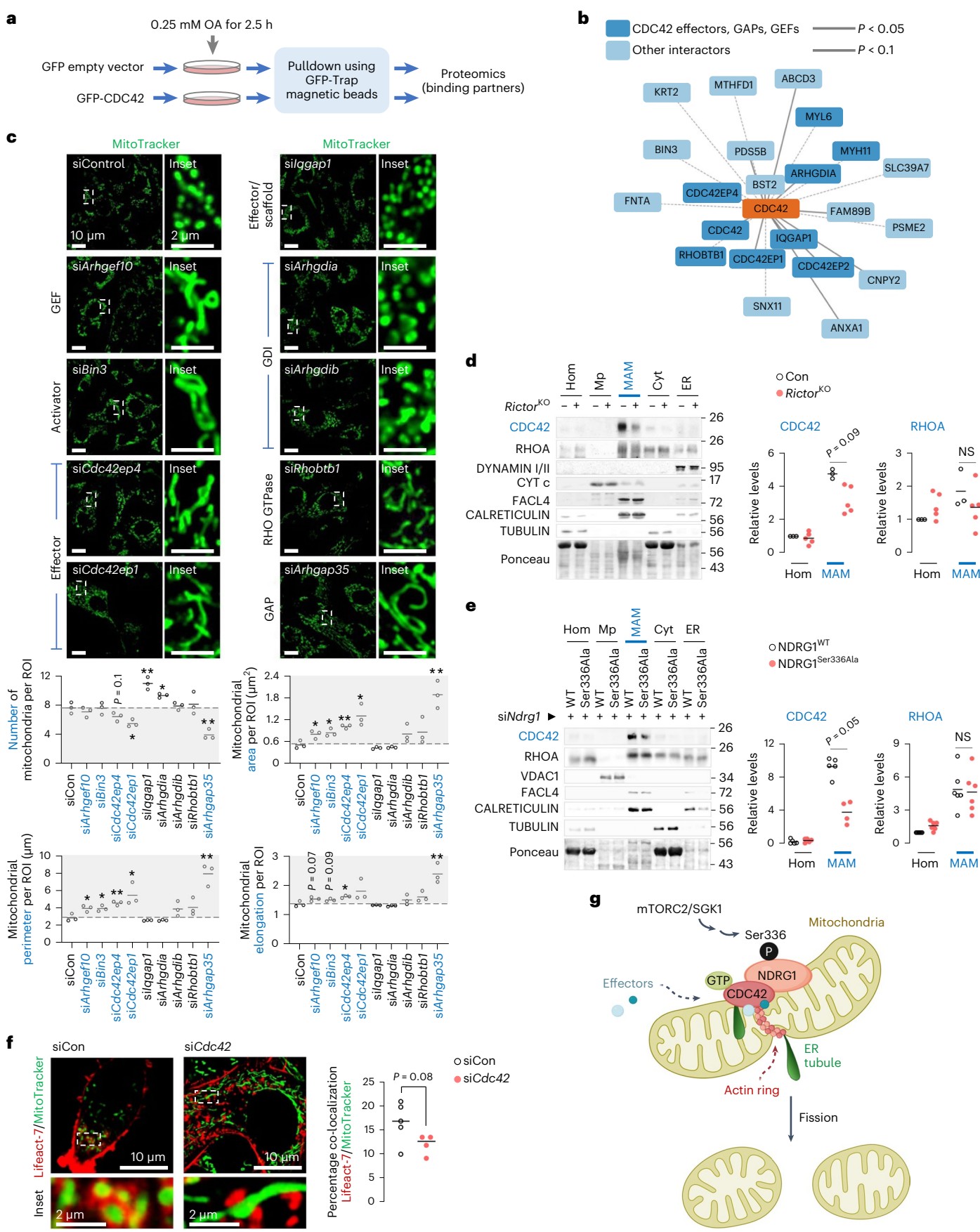

mTORC2–NDRG1-mediated fission is dependent on MFF but appears to not require DRP1. Supporting this possibility, loss of *Rictor* or expressing the NDRG1[Ser336Ala] mutant each markedly reduced MFF levels in MAMs without affecting DRP1 levels. These findings suggest that roles for MFF in fission are complex and not restricted to merely serving as a receptor for DRP1 recruitment[4].

How then does NDRG1 drive mitochondrial fission? Since NDRG1 lacks GTP hydrolysis activity, a requirement for fission[38,39], we examined whether NDRG1 engages with additional GTPases to facilitate fission. Here we identify the small GTPase CDC42 as a binding partner of NDRG1 that fails to interact with the phosphorylation-deficient NDRG1[Ser336Ala] mutant. Indeed, time-lapse imaging revealed that NDRG1 and CDC42 both engage at mitochondrial constrictions to facilitate fission, and that in absence of *Cdc42* or presence of inactive GDP-bound CDC42[Thr17Asn] mutant, NDRG1 fails to cut mitochondria. Furthermore, we detected the presence of CDC42 in MAMs, the enrichment of which appears to depend on an intact functional mTORC2–NDRG1 axis since loss of *Rictor* or expressing the NDRG1[Ser336Ala] mutant each markedly reduced CDC42 levels in MAMs. Given that CDC42 modulates the actin cytoskeleton[40], it is tempting to speculate that recruitment of CDC42 to MAMs by the mTORC2–NDRG1 axis orchestrates a local interplay between actin tubules and ER in driving scission, although careful future assessments are needed to conclusively demonstrate the same. Since *Rictor* insufficiency shortens lifespan[62,63], and given the age-related impairment in mitochondrial function, it is also tempting to speculate that stimulation of mTORC2 to sustain mitochondrial fission could potentially delay age-related diseases in which defective mitochondrial dynamics play a part.

## Online content

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

## Methods

This research complies with all relevant ethical regulations including animal protocol approval from the IACUC of Albert Einstein College of Medicine (protocol number 00001051).

### Animal models

C57BL/6 (000664), *Rictor*$^{flox/flox}$ (020649), *Raptor*$^{flox/flox}$ (013188) and *Tsc1*$^{flox/flox}$ (005680) mice were from Jackson Laboratory. Studies were performed in 2–10-month-old male and female mice fed regular chow (5058; Lab Diet) and maintained in barrier facility at 22–23 °C under 40–60% humidity and a 12 h:12 h light/dark cycle. Liver-specific *Rictor*$^{KO}$, *Raptor*$^{KO}$ or *Tsc1*$^{KO}$ mice were generated via retro-orbital injections of $2 \times 10^{11}$ genome copies per mouse of AAV8-TBG-iCre adenovirus (Vector Biolabs, VB1724) and mice were humanely killed after 8 weeks[64]. AAV8-TBG-eGFP (Vector Biolabs, VB1743)-injected mice were controls. Mice were fasted with free access to water and were compared with ad libitum mice. Fasted mice were treated with: oral gavage of (1) corn oil (400 µl; Sigma-Aldrich, 8267), (2) BODIPY FL C$_{16}$ (10 mg kg$^{-1}$; Invitrogen, D3821) or (3) refed high-fat diet (60% kcal in fat; Research Diets, D12492) for 30 min. Mice were mock-gavaged for 5 days before experiment, and control animals were gavaged with vehicle (10% dimethyl sulfoxide–saline solution) to match for volume and distension. STZ (40 mg kg$^{-1}$; Sigma-Aldrich, S0130) was injected intraperitoneally once a day for 5 consecutive days. The protocol was repeated after 8 weeks, and tissues were collected 2 weeks after the last injection.

### Corn oil

Corn oil (Sigma-Aldrich, 8267) is protein free, and provides fatty acids and monoacylglycerols, which are absorbed by the gut and delivered systemically. The composition of corn oil is detailed in Supplementary Table 11.

### Cell culture

NIH3T3 (ATCC, CRL-1658) and HepG2 (ATCC, HB-8065) cells were cultured in high-glucose Dulbecco's modified Eagle medium (DMEM) (Gibco, 11965118) supplemented with 10% (v/v) foetal bovine serum (FBS) (Sigma-Aldrich, 12106C) and 1% (v/v) penicillin–streptomycin (Gibco, 15140). AML12 cells (ATCC, CRL-2254) were cultured in DMEM/F-12 medium (Gibco, 11320033) supplemented with 10% FBS, 1% insulin–transferrin–selenium (Gibco, 41400-045), 40 ng ml$^{-1}$ dexamethasone (Sigma-Aldrich, D4902) and 1% penicillin–streptomycin. Cells were maintained at 37 °C in 5% CO$_2$. Wherever indicated, NIH3T3 cells were washed once with PBS and incubated in serum-free DMEM/P/S in presence of 0.25 mM OA (Sigma-Aldrich, O3008) for indicated durations.

Primary mouse embryonic fibroblasts were isolated from *Rictor*$^{flox/flox}$ mice as described[65]. *Rictor*$^{flox/flox}$ cells were plated at ~80% confluency and infected with 50 multiplicity of infection of adenoviral-null Ad(RGD)-fLuc (Vector Biolabs, 9999) or Ad(RGD)-CMV-iCre (Vector Biolabs, 1769) in serum-free medium for 24 h. The virus-containing medium was replaced by 10% FBS medium, and 72 h post-infection cells were used for experiments.

### Plasmid DNAs

Mouse NDRG1_OMu19504D, BNIP3_OMu13517D and CDC42_OMu16203C_cDNA expression plasmids were synthesized by GenScript USA. NDRG1_OMu19504D and BNIP3_OMu13517D were each cloned into pcDNA3.1+/C-(K)-DYK vector. CDC42_OMu16203C was cloned into pcDNA3.1(+)-N-eGFP vector. cDNA encoding NDRG1$^{Thr328Ala}$, NDRG1$^{Thr328Asp}$, NDRG1$^{Ser332Ala}$, NDRG1$^{Ser332Asp}$, NDRG1$^{Ser336Ala}$, NDRG1$^{Ser336Asp}$, BNIP3$^{Ser79Ala}$ or BNIP3$^{Ser88Ala}$_pcDNA3.1+/C-(K)-DYK mutants was generated by site-directed mutagenesis. pcDNA3.1+/C-(K)-DYK or pcDNA3.1(+)-N-eGFP vectors were negative controls. For live-cell imaging, mouse NDRG1_OMu19504D WT, mutant NDRG1$^{Ser336Ala}$ and CDC42_OMu16203C WT plasmids were each cloned into a pcDNA3.1(+)-mCherry vector. cDNA encoding CDC42$^{Thr17Asn}$_pcDNA3.1(+)-mCherry mutant was generated by site-directed mutagenesis. mCherry–Lifeact-7 was a gift from M. Davidson (Addgene, 54491).

### In vitro transfections of nucleic acids

In vitro transfections were performed using Lipofectamine 3000 (Invitrogen, L3000). For expression of DNA plasmids, 120,000 NIH3T3 cells ml$^{-1}$ of growth medium were transfected with 1 µg of DNA and plated in 12-well plate dishes for 48 h. For gene silencing, 120,000 cells ml$^{-1}$ of growth medium were transfected with siRNA for 48 h (Supplementary Table 12). Scrambled RNA was used as negative control (siCon). Silencing efficiency was confirmed by western blotting or qPCR.

### In vivo delivery of nucleic acids

In vivo delivery of plasmid DNAs was performed via in vivo-jetPEI (Polyplus-transfection SA, 201-50G) as per the manufacturer's instructions. Briefly, 100 µg of NDRG1$^{WT}$ or NDRG1$^{Ser336Ala}$_pcDNA3.1+/C-(K)-DYK was diluted in glucose solution and combined with 7 µl of in vivo-jetPEI for 15 min at room temperature. Then 200 µl of transfection mix was administered retro-orbitally to C57BL/6 mice in a single injection 24 h before tissue collection. Transfection efficiency was determined by immunohistochemistry. Livers from non-transfected mice were used as negative controls. In vivo siRNA delivery was performed using Invivofectamine 3.0 Reagent (Invitrogen, IVF3005) as per the manufacturer's instructions. Briefly, 50 µg of siRNAs was mixed with complexation buffer, added to Invivofectamine 3.0 Reagent (1:1 ratio) and incubated for 30 min at 50 °C. The mix was diluted in PBS (pH 7.4), and 200 µl of siRNA mix was administered retro-orbitally to C57BL/6 mice every 24 h for 3 consecutive days before tissue collection.

### RNA isolation and real-time PCR

mRNA expression was performed as described[66] using M-MLV Reverse Transcriptase (Invitrogen, 28025). The primers are detailed in Supplementary Table 13.

### Western blotting

Total cell lysates from cells in culture were prepared using lysis buffer (20 mM Tris pH 7.5, 50 mM NaCl, 0.5%, 1 mM EDTA, 1 mM EGTA and 1% Triton X-100) supplemented with complete EDTA-free protease inhibitor (Roche, 11873580001) and phosphatase inhibitor cocktails 2 and 3 (Sigma-Aldrich, P5726 and P0044). Total protein from liver or epididymal adipose tissue was isolated in RIPA buffer (50 mM Tris pH 8.0, 150 mM NaCl, 0.5% sodium deoxycholate, 1% SDS and 1% NP-40) supplemented with protease/phosphatase inhibitors. Total protein from soleus muscle was isolated as described[67]. Lysates were centrifuged at 17,000$g$ for 30 min at 4 °C, and supernatants were immunoblotted by denaturing 20–30 µg of protein at 95 °C for 5 min in 3× Laemmli buffer. For analysis of OXPHOS, samples were boiled at 50 °C for 5 min and resolved by SDS–PAGE as described[68]. Protein bands were normalized to Ponceau S and quantified by ImageJ (National Institutes of Health, NIH). Antibodies are detailed in Supplementary Table 14.

### Subcellular fractionation

Fresh livers were fractionated for isolation of MAMs, pure mitochondria, cytosol and ER fractions as described[69]. Cytochrome *c* (CYT *c*) and Voltage Dependent Anion Channel 1 (VDAC1) were used as enrichment/purity markers for mitochondria, long-chain fatty acid coenzyme A ligase 4 (FACL4) as marker for MAM, calreticulin as marker for MAMs and ER, and tubulin as marker for cytoplasm.

### Co-IP

For Fig. 6b, lysates (1,000 µg) from NIH3T3 cells co-expressing Flag–NDRG1$^{WT}$ or Flag–NDRG1$^{Ser336Ala}$ with GFP–CDC42 were incubated with 30 µl of Anti-Flag M2 Affinity Gel (Sigma-Aldrich, A2220) and eluted

with 3× Flag peptide (Sigma-Aldrich, F4799) by incubating for 2 h at 4 °C in rotation. Co-IP eluents were immunoblotted. Cells expressing Flag empty vector were negative controls. For Extended Data Fig. 9c, lysates (1,000 µg) from NIH3T3 cells co-expressing mCherry–CDC42$^{WT}$ or mCherry–CDC42$^{Thr17Asn}$ with Flag–NDRG1$^{WT}$ were subjected to Flag pulldowns as above. For identification of Flag–NDRG1$^{WT}$ or Ser336Ala-interacting partners in Fig. 6a, lysates (700 µg) from NIH3T3 cells expressing Flag–NDRG1$^{WT}$ or Flag–NDRG1$^{Ser336Ala}$ were subjected to Flag pulldowns, and eluents were subjected to mass spectrometry using S-trap columns and nLC–MS/MS as below. Non-transfected cells were negative controls. For identification of CDC42-interacting partners in Fig. 7b, lysates (700 µg) from NIH3T3 cells expressing GFP–CDC42 or GFP–empty vector were incubated with 25 µl of GFP-Trap Magnetic Agarose (Chromotek, gtma) for 2 h at 4 °C in rotation after which beads were washed and subjected to on-beads digestion for mass spectrometry using S-trap columns and nLC–MS/MS as below.

## Sample preparation for phosphoproteomics

Liver (500 µg), liver MAM fractions (700 µg) or co-IP eluents from Flag pulldowns performed in total cell lysates (700 µg) of siCon or si*Rictor* NIH3T3 cells co-expressing Flag–NDRG1$^{WT}$ were homogenized in 2% SDS/5 mM dithiothreitol (with protease/phosphatase inhibitors) to retrieve proteins in solution and incubated for 1 h at room temperature for disulfide bond reduction. Proteins were alkylated using 20 mM iodoacetamide for 30 min in the dark. Protein digestion was performed utilizing S-trap mini cartridges (ProtiFi) as per the manufacturer's instructions. Phosphorylated peptides were enriched from the S-trap eluate using titanium dioxide beads (TiO$_2$, GL Sciences) as described[70]. Following TiO$_2$ enrichment, peptides were concentrated with a speed vac, desalted in HLB resin (Waters) and concentrated in a speed vac once more before analysing peptides by nLC–MS/MS.

## nLC–MS/MS acquisition

Samples were resuspended in 10 µl of water/0.1% trifluoroacetic acid and loaded onto a Dionex RSLC Ultimate 300 (Thermo Scientific) coupled online with an Orbitrap Fusion Lumos (Thermo Scientific). The two-column chromatographic separation system consisted of a C18 trap cartridge (300 µm internal diameter (ID), 5 mm length) and a picofrit analytical column (75 µm ID, 30 cm length) packed in-house with reversed-phase Repro-Sil Pur C18-AQ 3 µm resin. Peptides were separated using a 180 min gradient from 2% to 28% buffer B (buffer A: 0.1% formic acid; buffer B: 80% acetonitrile/0.1% formic acid) at a flow rate of 300 nl min$^{-1}$. The mass spectrometer acquired spectra in a data-dependent acquisition mode. Briefly, the full MS scan was set to 300–1,200 $m/z$ in the Orbitrap with a resolution of 120,000 (at 200 $m/z$) and an AGC target of $5 \times 10^5$. MS/MS was performed in the ion trap using top speed mode (2 s), an AGC target of $10 \times 10^4$ and higher collisional dissociation (HCD) collision energy of 30. Two additional targeted scans were added in each instrument duty cycle to detect the low-abundance NDRG1$^{Ser336}$ peptide: a selected ion monitoring scan for the intact mass quantification and a targeted MS/MS scan for identification of the peptide.

Phosphoproteomics data analysis was conducted using Proteome Discoverer v2.4 (Thermo Scientific) at standard settings for tolerances, modifications and filters, and phosphorylation on Ser/Thr/tyrosine as dynamic modifications. SwissProt mouse proteome database was used (downloaded August 2019). Peptide abundance was obtained using the intensity of the extracted ion chromatogram; values were log$_2$ transformed and normalized, and missing values were imputed as described[71]. Comparisons between groups were performed in a binary manner; each sample type was compared with the fasted condition utilizing a two-tails heteroscedastic $t$-test (significant, if $P$ value < 0.05). The data distribution was assumed to be normal.

Significantly modified proteins were selected by Benjamini–Hochberg correction ($P < 0.05$). When false discovery rate correction led to no hit, inspection of uncorrected $P$ value distribution was performed: if an anti-conservative distribution was observed, we applied an alternative method of false discovery rate control by combining threshold for significance ($P < 0.05$) with fold-change cut-off (fold-change >1.5) as suggested[72]. Phosphorylation state change ($\Delta$Ps) for individual proteins was calculated as described[34], as the sum of log$_2$(fold change) value of all phosphopeptides with statistically significant changes ($P < 0.05$) compared to control. If none of the phosphopeptide $P$ values is below 0.05, the $\Delta$Ps value will be zero. We applied a stringent cut-off for $\Delta$Ps value at two standard deviations ($2\sigma$) to represent the concept of cumulative phosphorylation. Gene Ontology was performed using BINGO or Enrichr[73]. In the enrichment map-based network visualization of Gene Ontology enrichment of differentially modulated phosphosites, blue edges show similarity between decreased phosphosites while red nodes show similarity between increased phosphosites; node size indicates the number of proteins per node; major clusters are circled, and the associated names represent major functional associations. The enrichment map was generated in Cytoscape (3.8.1) using Enrichment map plugin (3.3.0) (ref. 33) using $P$ value <0.05, false discovery rate <0.001. Data handling and statistical analyses were performed using Python (Python software foundation; v.3.7.4 available at https://www.python.org/) and Scientific Python Stack: SciPy (v.1.3.1) (ref. 74), NumPy (v.1.17.2) (ref. 75) and Matplotlib (v.3.1.1). Phosphosites showing significant regulation between groups were used to predict the kinase responsible for their catalysis using the iGPS software[12]. Significantly regulated phosphorylation events were used to predict the kinases responsible for their catalysis using iGPS[12]. Positive kinase scores represent most confident and frequent predictions for upregulated phosphosites, with blue representing downregulated phosphosites. The higher the cumulative score retrieved from iGPS, the more intense the colour coding of the bubbles in the network. Upregulated and downregulated refers to numerator and denominator as defined in the header of each panel. The bubble size is scaled on the basis of the number of phosphorylation events predicted to be catalysed by the given kinase. The connector lines represent previously associated genetic interactions between listed proteins retrieved from the database STRING v.11 (ref. 76). The network was displayed using Cytoscape[77].

## Biochemical analyses

Blood glucose was measured using Ascensia Contour glucometer (Bayer). Serum insulin (ALPCO, 80-INSNS-E01), leptin (R&D Systems, DY49805), IGF-1 (R&D Systems, DY791), FFAs (FUJIFILM, NEFA-HR (2)), serum triglyceride (Sigma-Aldrich, T2449, F6428) and liver triglycerides (BioVision, K622) were evaluated as per the manufacturer's instructions.

Seahorse XF Cell Mito Stress Test was performed as per the manufacturer's instructions (Agilent Technologies). Briefly, 12,000 NIH3T3 cells per 200 µl of medium were transfected with DNAs and/or siRNAs and seeded onto a Seahorse XF96 Cell Culture Microplate (Agilent Technologies, 101085-004) for 32 h. After 16 h of stress in low-glucose medium (1 g l$^{-1}$; Agilent Technologies, 103577), cells were washed with PBS, cultured in 165 µl of XF Base Medium (Agilent Technologies, 103335) supplemented with 1 g l$^{-1}$ D-glucose, 2 mM sodium pyruvate (Gibco, 11360) and 4 mM L-glutamine (Gibco, 25030) and incubated at 37 °C without CO$_2$ for 1 h. OA (0.25 mM) was added, and the microplate was loaded into XF Analyzer. Basal OCR measurements were recorded four times (mix 3 min, wait 2 min, measure 3 min) after sequential injections of oligomycin (1 µM), carbonyl cyanide-4 (trifluoromethoxy) phenylhydrazone (FCCP; 20 µM) and rotenone/antimycin (1 µM) with four readings (mix 3 min, wait 2 min, measure 3 min) after each injection. OCR was normalized to cell number estimated with CyQUANT Cell Proliferation Assay (Invitrogen, C7026) as per the manufacturer's instructions. Mitochondrial parameters were calculated as per the manufacturer's instructions. OCRs of liver explants were performed as described[66].

## Histological analyses

Flag was detected using a Mouse on Mouse (MOM) ImmPRESS HRP (Peroxidase) Polymer Kit (Vector Laboratories, MP-2400). Paraffin-embedded livers were cut into 5 µm sections, deparaffinated in xylene and rehydrated in a series of graded alcohols and water. For antigen unmasking, sections were incubated in citrate-based antigen unmasking solution (pH 6.0; Vector Laboratories, H-3300) at high temperature for 20 min. After blocking in BLOXALL Endogenous Blocking Solution (Vector Laboratories, SP-6000) for 10 min and a 1 h incubation in MOM Mouse IgG Blocking Reagent, sections were stained with mouse monoclonal anti-DYKDDDDK Tag antibody (1:100; Cell Signaling Technology, 8146) in 2.5% normal horse serum MOM solution overnight at 4 °C. DYKDDDDK signal was revealed by incubation with MOM ImmPRESS Reagent for 10 min and enhanced with ImmPACT DAB EqV Peroxidase (HRP) Substrate (Vector laboratories, SK-4103) for 1 min. Sections were counterstained with haematoxylin, dehydrated, mounted with Permount mounting medium (Fisher, SP15) and imaged in a Zeiss Axiolab 5 microscope/Axiocam 305 colour camera (Carl Zeiss Microscopy). Quantification of Flag percentage area was performed as described[78].

## Oil Red O staining

Oil Red O staining was performed as described[79].

## Confocal microscopy

Was performed as described[68]. For Flag detection, DYKDDDDK Tag Rabbit antibody was used at 1:100 dilution (Cell Signaling Technology, 14793). Where indicated, 30 min before fixation with 4% paraformaldehyde, cells were incubated with 100 nM MitoTracker Red CMXRos (Invitrogen, M7512) to assess mitochondrial membrane potential. Mounted coverslips were imaged on a Leica TCS SP8 Confocal Laser Scanning Microscope (Leica Microsystems) with ×63 objective and 1.4 numerical aperture. Quantification of MitoTracker Red CMXRos fluorescence intensity per cell was performed using ImageJ (NIH) and expressed as mean integrated density. For detection of BODIPY FL $C_{16}$ in vivo, sections from freshly isolated livers were mounted with Fluoromount-G medium (SouthernBiotech, 0100) and imaged on Leica TCS SP8 Confocal Laser Scanning Microscope with ×10 objective and 1.4 numerical aperture.

## Live cell imaging

Cells were transfected with siRNA and/or DNA as above and seeded onto a glass-bottom 35 mm culture dish (MatTek Corporation, P35G-1.5-14-C) for 48 h. After PBS washing, cells were incubated in serum-free DMEM in presence of MitoTracker Green FM (500 nM; Invitrogen, M7514) or ER-Tracker Green (500 nM; Invitrogen, E34251) for 30 min to stain mitochondria and ER, respectively. Cells were washed once with PBS and incubated in red phenol-free DMEM (Gibco, 31053) with 4 mM L-glutamine and 12 mM HEPES (pH 7.4), and imaged using a Leica TCS SP8 confocal laser scanning microscope (Leica Microsystems) and single planes were acquired with ×63 objective and 1.4 numerical aperture. For time-lapse imaging, cells were tracked at a rate of one frame per 13 or 27 s (for single or simultaneous dual-channel acquisition, respectively) over 10 min.

Image analysis was done with ImageJ (NIH). Individual frames were denoised by applying Gaussian filter and a region of interest (ROI) of 35 µm$^2$ was selected across the different experimental conditions. After image auto-thresholding, quantification of mitochondrial number and morphology parameters was performed using the 'analyze particles' macro as described[80]. Mitochondrial elongation was calculated as the inverse of circularity[81]. Mitochondrial fission and fusion frequency were calculated as described[82] and expressed as number of events per cell per second. Percentage co-localization was calculated using the JACoP plugin as described[68].

## TEM

Freshly isolated livers were fixed with 2% paraformaldehyde and 2.5% glutaraldehyde in 0.1 M sodium cacodylate, post-fixed with 2% osmium tetroxide, 1.5% potassium ferrocyanide, 0.15 M sodium cacodylate, 2 mM CaCl$_2$, followed by 1% thiocarbohydrazide, and then 2% osmium tetroxide, en bloc stained with 1% uranyl acetate and further stained with lead aspartate. Samples were dehydrated through graded series of ethanol and embedded in LX112 resin (LADD Research Industries). Ultrathin (55 nm) sections were cut on a Leica ARTOS 3D ultramicrotome and collected onto silicon wafers. Sections were examined on Zeiss Supra 40 Field Emission Scanning Electron Microscope (Carl Zeiss Microscopy, LLC North America) in backscatter mode using an accelerating voltage of 8.0 kV. The number of mitochondria was counted manually in an ROI of 71.2 µm$^2$. Quantification of mitochondrial shape descriptors was performed by manual tracing of individual mitochondria using freehand tool. Contact sites between mitochondria and ER (defined to be at 10–30 nm distance[83]) were quantified, normalized to total number of mitochondria and expressed as percentage. For 3D reconstruction, regions of interest were collected using ATLAS 5.0, with a pixel size of 6.0 and dwell time of 6 µs. Stacks were aligned, and segmentation was done using IMOD[84]. Tomographic reconstruction was performed as described[85].

## Lipidomic analyses

Lipid extracts from liver homogenates, MAM, pure mitochondria and ER fractions were prepared using modified Bligh and Dyer method, spiked with appropriate internal standards, and analysed on an Agilent 1260 Infinity HPLC integrated to Agilent 6490 A QQQ mass spectrometer controlled by Masshunter v 7.0 (Agilent Technologies). Glycerophospholipids and sphingolipids were separated with normal-phase HPLC as described[86], with a few modifications. An Agilent Zorbax Rx-Sil column (2.1 × 100 mm, 1.8 µm) at 25 °C was used under the following conditions: mobile phase A (chloroform:methanol:ammonium hydroxide, 89.9:10:0.1, v/v) and mobile phase B (chloroform:methanol:water:ammonium hydroxide, 55:39:5.9:0.1, v/v); 95% A for 2 min, decreased linearly to 30% A over 18 min and further decreased to 25% A over 3 min, before returning to 95% over 2 min and held for 6 min. Separation of sterols and glycerolipids was carried out on a reverse phase Agilent Zorbax Eclipse XDB-C18 column (4.6 × 100 mm, 3.5 µm) using an isocratic mobile phase, chloroform:methanol:0.1 M ammonium acetate (25:25:1) at a flow rate of 300 µl min$^{-1}$.

Quantification of lipid species was accomplished using multiple reaction monitoring transitions[86,87] under positive and negative ionization modes and using internal standards: phosphatidic acid (PA) 14:0/14:0, phosphatidylcholine (PC) 14:0/14:0, phosphatidylethanolamine (PE) 14:0/14:0, phosphatidylgylcerol (PG) 15:0/15:0, phosphatidylinositol (PI) 17:0/20:4, phosphatidylserine (PS) 14:0/14:0, bis[monoacylglycero]phosphate (BMP) 14:0/14:0, acylphosphatidyl glycerol (APG) 14:0/14:0, lysophosphatidylcholine (LPC) 17:0, lysophosphatidylethanolamine (LPE) 14:0, lysophosphatidylinositol (LPI) 13:0, ceramide (Cer) d18:1/17:0, sphingomyelin (SM) d18:1/12:0, dihydrosphingomyelin (dhSM) d18:0/12:0, galactosylceramide (GalCer) d18:1/12:0, glucosylceramide (GluCer) d18:1/12:0, lactosylceramide (LacCer) d18:1/12:0, D7-cholesterol, cholesterol ester (CE) 17:0, monoglyceride (MG) 17:0, 4ME 16:0 diether DG, D5-TG 16:0/18:0/16:0 (Avanti Polar Lipids). Lipids per sample were calculated by summing total moles of all lipid species measured by all three LC–MS methodologies, and normalizing to mol % (Supplementary Table 3).

## Data collection and analysis softwares

The following devices and softwares were used: (1) Zeiss Axiolab 5 microscope with Axiocam 305 colour camera for immunohistochemistry (Zeiss ZEN v3.7), (2) Leica TCS SP8 confocal laser scanning microscope (LAS X v.5.7.23225), (3) Zeiss Supra 40 Field Emission Scanning

Electron Microscope to acquire transmission electron microscope (Zeiss SmartSEM v6.0) and 3D Modeling (3DMOD v4.9.10), (4) XF96 and X24 Seahorse analyzers (Agilent Technologies) to collect OCRs (WAVE Pro v10.0.1.84; v2.6.1.56, respectively), (5) Synergy HTX (BioTek) multi-mode plate reader (Gen5 v3.12), (6) StepOnePlus Real-Time PCR System (Applied Biosystems) for mRNA expression (StepOne v2.3), (7) Microsoft Excel v16.48, Microsoft Word v16.48, Microsoft PowerPoint v16.47, (8) Prism v8.4.3, (9) Endnote X9.3.3 and (10) ImageJ v2.0.0-rc-69/1.52p. Enrichment map was generated in Cytoscape v3.8.1, Enrichment map plugin v3.3.0. Handling and analyses of proteomics data were performed via Python v.3.7.4 and Scientific Python Stack: SciPy v.1.3.1, NumPy v.1.17.2 and Matplotlib v.3.1.1. Phosphosites showing significant regulation between groups were used to predict the kinase responsible for their catalysis using the iGPS software.

### Illustration

The proposed model in Fig. 7g was created with BioRender (BioRender. com).

### Statistics

All data are mean of a minimum of three independent experiments unless otherwise stated. Statistical significance was assessed by two-tailed unpaired Student's *t*-test, one-way or two-way analyses of variance (ANOVAs) followed by Tukey's, Šídák's or Dunnett's multiple-comparisons test. *n* numbers indicate biological replicates. Statistical summary is presented in Supplementary Table 10. Raw source data are presented in Source Data Extended Data Table 1.

### Reporting summary

Further information on research design is available in the Nature Portfolio Reporting Summary linked to this article.

## Data availability

The mass spectrometry proteomics data have been deposited to the ProteomeXchange Consortium via the PRIDE[88] partner repository, and data are available via ProteomeXchange with identifier PXD041696. Data from this study are available at https://doi.org/10.6084/m9.figshare.22670575. Source data have been provided with this paper. All other data supporting the findings of this study are available from the corresponding author on reasonable request.

## Code availability

All the Python codes used in proteomic analyses are fully available on GitHub (https://github.com/MathieuBo/mTorc2_mito_fission).

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

## Acknowledgements

This work was supported by RF1AG043517, R01DK123327, R01AG065985 and P01AG031782 (R.S.); NIH P30 CA013330 47 (S.S. and J.T.A.); R56AG072794 and R56AG062271 (L.B.J.M); P30CA013330 and SIG #1S10OD016214-01A1 (Einstein Analytical Imaging Facility). This study was supported by the Biomarkers Core Laboratory at the Irving Institute for Clinical and Translational Research, home to Columbia University's Clinical and Translational Science Award. We thank K. Kulej for assistance with phosphoproteomic analyses. We thank S. Kaushik for suggestions for time-lapse imaging.

## Author contributions

Conceptualization, R.S. and N.M.-L.; methodology, R.S. and N.M.-L.; investigation, N.M.-L., P.M., M.T., H.B., M.K., M.L.A. and M.S.; writing—original draft, R.S.; revised draft, R.S. and N.M.-L.; data analyses, R.S., N.M.-L., S.S. and M.B.; funding acquisition, R.S.; resources, L.G.-C., F.P.M., L.B.J.M, J.T.A. and S.S.; supervision, R.S.

## Competing interests

The authors declare no competing interests.

## Additional information

**Extended data** is available for this paper at https://doi.org/10.1038/s41556-023-01163-3.

**Correspondence and requests for materials** should be addressed to Rajat Singh.

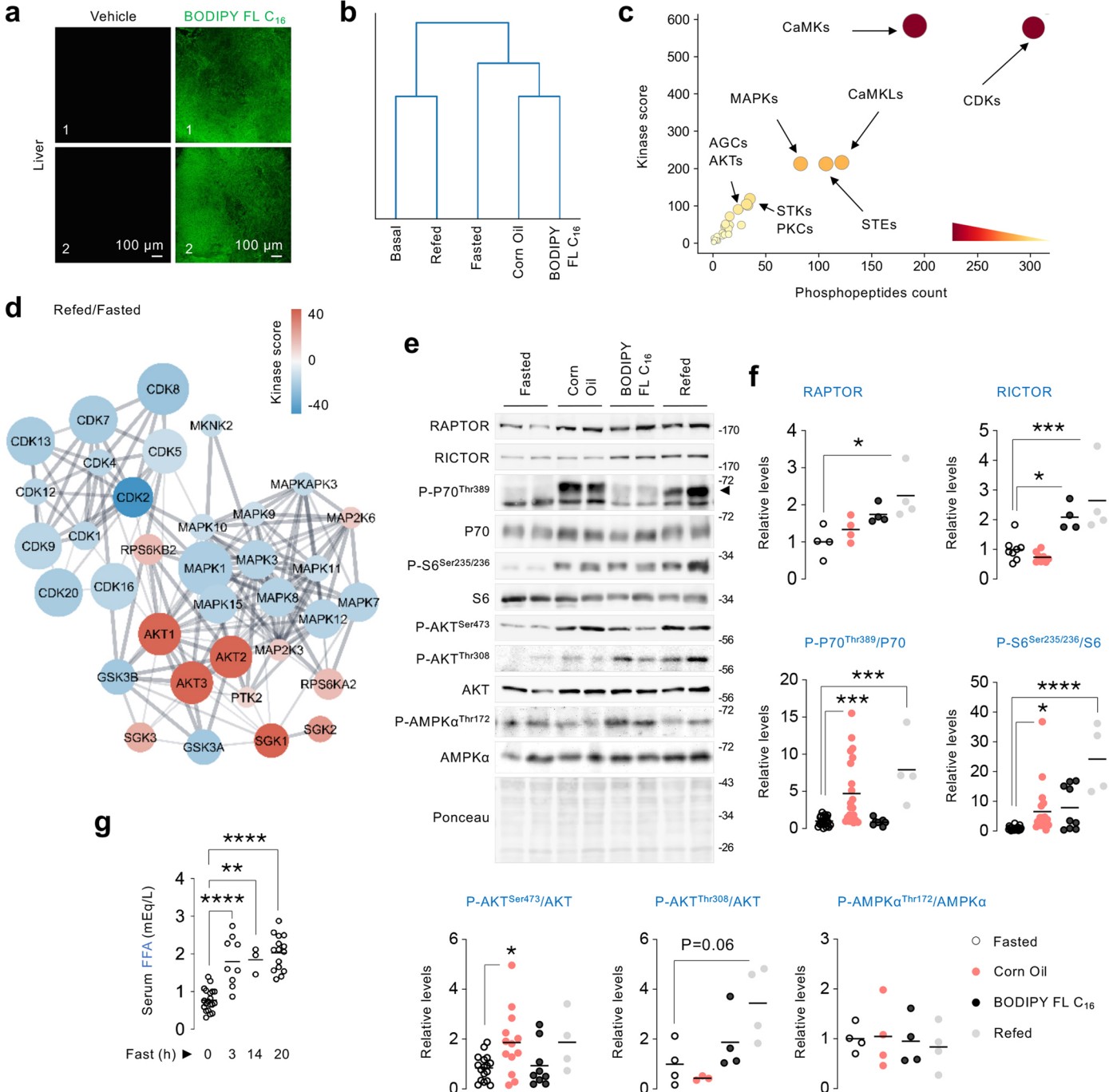

**Extended Data Fig. 1 | Kinases responsive to fasting or fatty acids. (a)** Direct BODIPY fluorescence in liver slices of 8 mo-old C57BL/6 male mice subjected to 30 min gavage with vehicle or 10 mg/kg of BODIPY FL C$_{16}$ after 14–16 h fasting (n = 5 mice). **(b)** Dendrogram of hierarchical clustering analysis (Euclidean distance) between experimental groups based on phosphorylation status of 863 phosphosites. **(c)** Prediction for kinases that putatively target the phosphosites identified within the green cluster in main Fig. 1b. Darker color intensity reflects higher kinase score. **(d)** Pairwise comparisons showing upregulated (red) or downregulated (blue) kinase networks in mice refed high fat diet (HFD) for 30 min after 14–16 h fasting. For **b-d** (n = 4 mice). **(e)** Representative IB, and **(f)** quantification for the indicated proteins in livers of 2–8 mo-old C57BL/6 male or female mice kept fasted for 14–16 h or treated with corn oil or BODIPY FL C$_{16}$ or refed a HFD for 30 min after fasting. *N* values for number of mice analyzed for individual proteins and conditions are indicated in parentheses. RAPTOR

and P-AMPK$^{Thr172}$/AMPK: n = 4 per group; RICTOR: fasted (n = 8), corn oil (n = 8), BODIPY FL C$_{16}$ (n = 4), refed (n = 4); P-P70$^{Thr389}$/P70: fasted (n = 23), corn oil (n = 22), BODIPY FL C$_{16}$ (n = 8), refed (n = 4); P-S6$^{Ser235/236}$/S6 fasted (n = 23), corn oil (n = 24), BODIPY FL C$_{16}$ (n = 9), refed (n = 4); P-AKT$^{Ser473}$/AKT: fasted (n = 17), corn oil (n = 13), BODIPY FL C$_{16}$ (n = 9), refed (n = 4); P-AKT$^{Thr308}$/AKT: fasted (n = 4), corn oil (n = 3), BODIPY FL C$_{16}$ (n = 4), refed (n = 4). Ponceau is the loading control. **(g)** Circulating free fatty acids (FFA) in 2–7 mo-old male mice fasted for: 0 h (n = 20 mice), 3 h (n = 9 mice), 14 h (n = 3 mice) or 20 h (n = 15 mice). Individual replicates and mean values are shown. *P < 0.05, ***P < 0.001 and ****P < 0.0001, One-way ANOVA followed by Šídák's multiple comparisons test **(f)**; One-way ANOVA followed by Tukey's multiple comparisons test **(g)**. Please refer to Supplementary Table 10_statistical summary, and Supplementary Tables 1, 2. Source numerical data are available in SourceData_Table 1, and unprocessed blots are available in Source Data Extended Data Fig. 1.

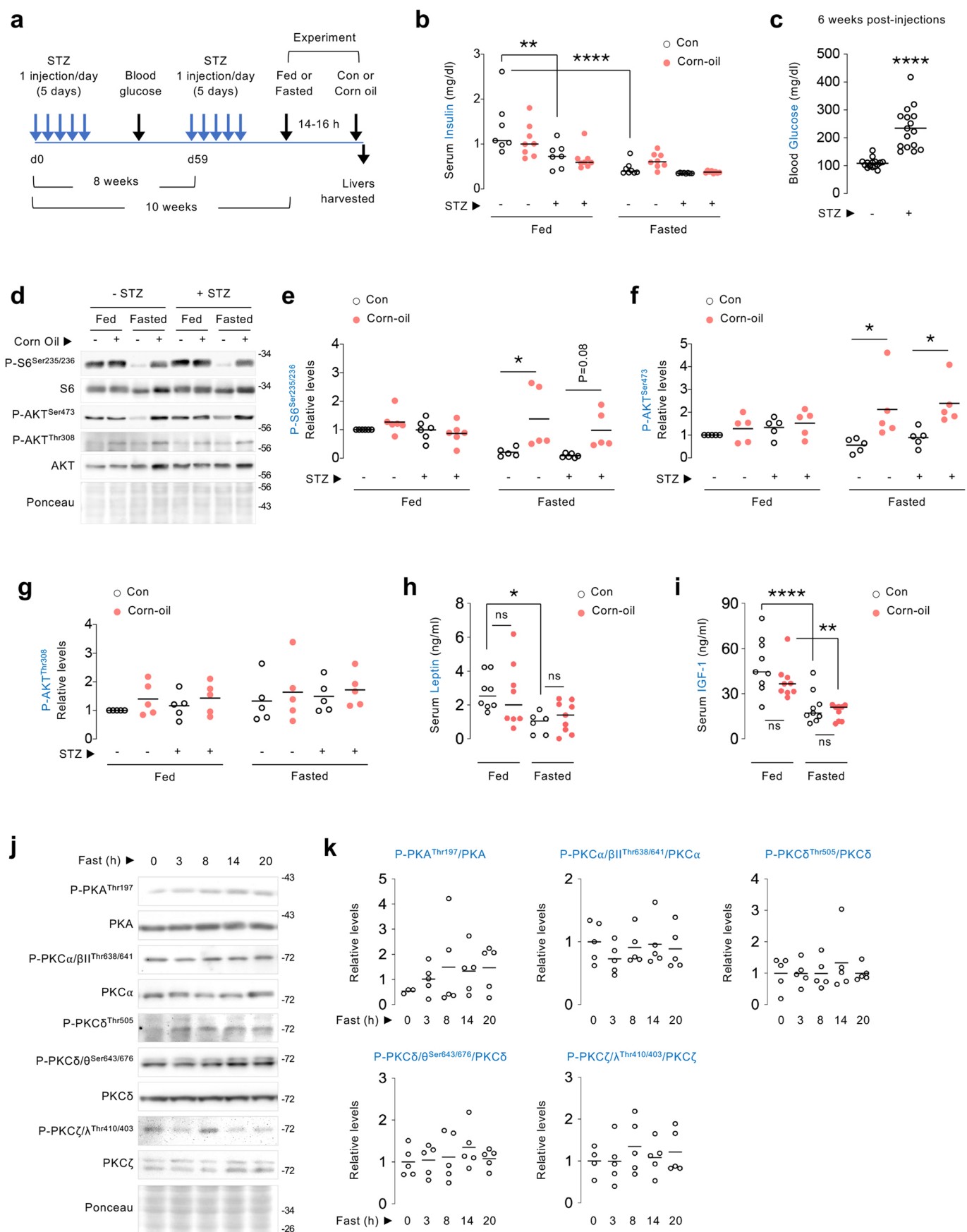

**Extended Data Fig. 2 | See next page for caption.**

**Extended Data Fig. 2 | Protein kinases A and Cs are not activated by fasting in liver, and lipid-driven mTORC1/C2 activation is leptin, IGF-1 or insulin-independent. (a)** Experimental plan for generation of insulin-deficient diabetic mice using streptozotocin (STZ). **(b)** Serum insulin levels in 3 mo-old STZ-treated C57BL/6 male mice that were fed or fasted for 14–16 h and then gavaged with corn oil for 30 min. $N$ values for number of mice in each group are in parenthesis. Fed (n = 7); fed + STZ (n = 7); fed + corn oil (n = 8); fed + STZ + corn oil (n = 8); fasted (n = 8); fasted + STZ (n = 8); fasted + corn oil (n = 8); fasted + STZ + corn oil (n = 8). **(c)** Blood glucose levels 6 weeks after the first injection with STZ (n = 16 mice). **(d)** Representative IB, and **(e-g)** quantification for indicated proteins normalized to corresponding total protein in livers of 3 mo-old STZ-treated C57BL/6 male mice that were fed or fasted for 14–16 h and then gavaged with corn oil for 30 min. For **e**, fed (n = 6); fed + STZ (n = 6); fed + corn oil (n = 6); fed + STZ + corn oil (n = 6); fasted (n = 5); fasted + STZ (n = 6); fasted + corn oil (n = 5); fasted + STZ + corn oil (n = 5). For **f** and **g**, all indicated groups consist of n = 5 mice each. **(h, i)** Serum **(h)** leptin and **(i)** IGF-1 levels in 3 mo-old C57BL/6 male mice that were fed or fasted for 14–16 h and then gavaged with corn oil for 30 min. For **h**, fed (n = 8); fed + corn oil (n = 8); fasted (n = 6); fasted + corn oil (n = 9). For **i**, all indicated groups consist of n = 9 mice each. **(j)** IB and **(k)** quantification for the indicated proteins in livers of 2–10 mo-old mice fed or fasted for the indicated durations. $N$ values for number of mice analyzed at each time-point for individual phosphoproteins are indicated in parentheses. P-PKA$^{Thr197}$/PKA: 0 h (n = 3), 3 h-20 h (n = 5); P-PKCα/βII$^{Thr638/641}$/PKCα (all time-points are n = 5); P-PKCδ$^{Thr505}$/PKCδ (all time-points are n = 5); P-PKCδ/θ$^{Ser643/676}$/PKCδ (all time-points are n = 5); and P-PKCζ/λ$^{Thr410/403}$/PKCζ: 0 h (n = 4), 3 h-20 h (n = 5). Ponceau is the loading control. Individual replicates and mean values are shown. *P < 0.05, **P < 0.01 and ****P < 0.0001, 2-way ANOVA followed by Tukey's multiple comparisons test **(b, e, f, h** and **i)**; two-tailed unpaired Student's t-test **(c)**. ns=not significant. Please refer to Supplementary Table 10_statistical summary. Source numerical data are available in SourceData_Table 1, and unprocessed blots are available in Source Data Extended Data Fig. 2.

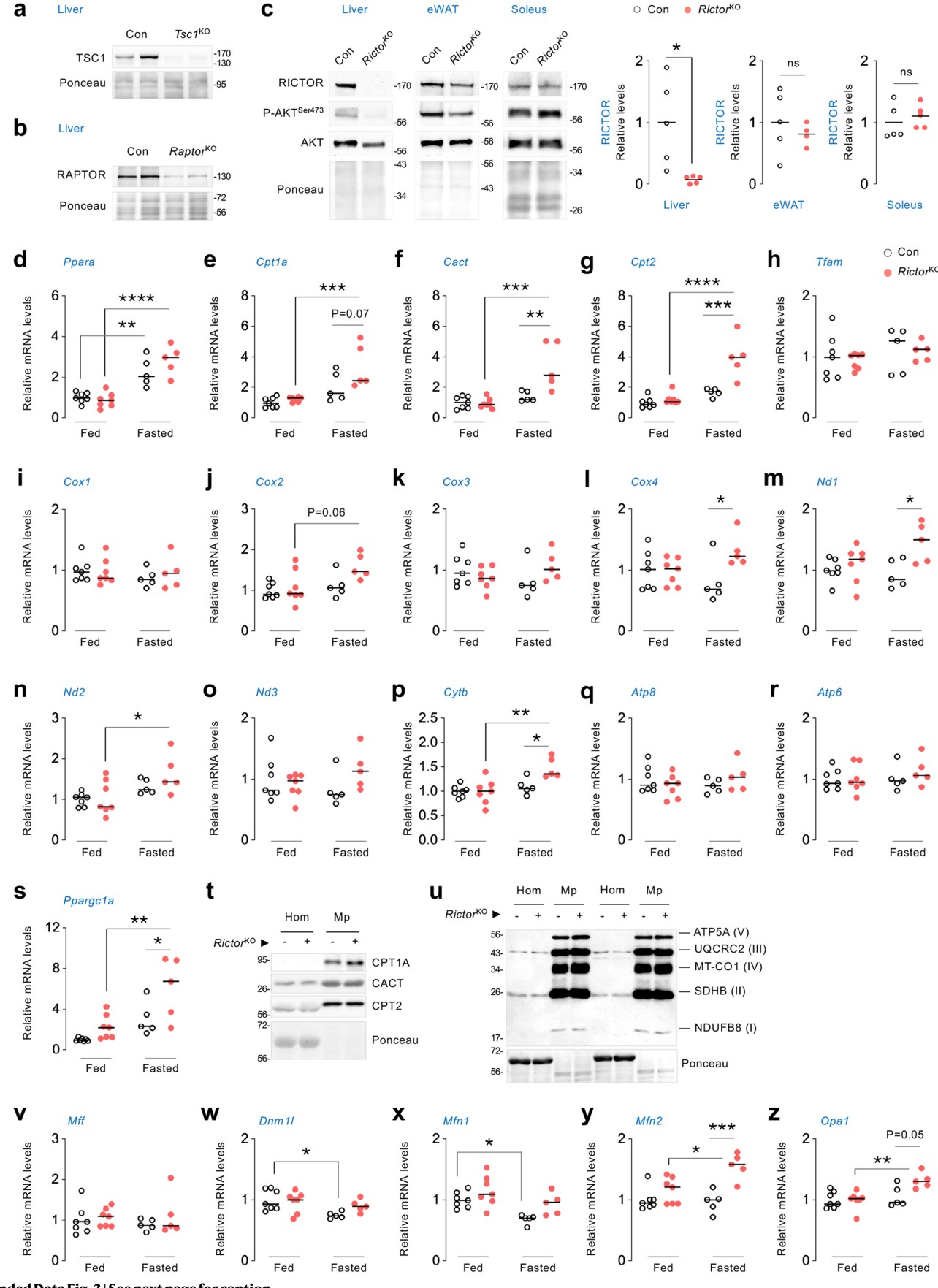

**Extended Data Fig. 3 | See next page for caption.**

**Extended Data Fig. 3 | Effect of loss of mTORC2 on expression of genes and proteins related to mitochondrial oxidative metabolism and dynamics. (a, b)** Representative IB to validate deletion of the indicated genes in livers of 4–6 mo-old Con, *Tsc1*^KO or *Raptor*^KO male and female mice. Blots for TSC1 and RAPTOR are representative of n = 7 Con or corresponding KO mice obtaining similar results. **(c)** IB for mTORC2 signaling and quantification of RICTOR protein levels in liver (n = 5 mice), epididymal white adipose tissue (eWAT) (n = 5 Con and n = 4 *Rictor*^KO mice) and soleus (n = 5 mice) of 4–6 mo-old Con and *Rictor*^KO male mice fasted for 14–16 h. **(d-s** and **v-z)** Relative mRNA expression of indicated genes in livers of 4–6 mo-old Con and *Rictor*^KO male mice that were fed or fasted for 14–16 h (n = 7 fed Con or *Rictor*^KO mice, and n = 5 fasted Con or *Rictor*^KO mice). **(t)** Representative IB for proteins involved in mitochondrial fatty acid uptake, and **(u)** OXPHOS in whole homogenates (Hom) and pure mitochondrial (Mp) fractions from livers of 5–6 mo-old Con and *Rictor*^KO male mice after 14–16 h fasting (n = 3 Con and n = 5 *Rictor*^KO mice). Ponceau is the loading control. Individual replicates and mean values are shown. *P < 0.05, **P < 0.01, ***P < 0.001 and ****P < 0.0001, two-tailed unpaired Student's t-test **(c)**; 2-way ANOVA followed by Tukey's multiple comparisons **(d-s** and **v-z)**. ns=not significant. Please refer to Supplementary Table 10_statistical summary. Source numerical data are available in SourceData_Table 1, and unprocessed blots are available in Source Data Extended Data Fig. 3.

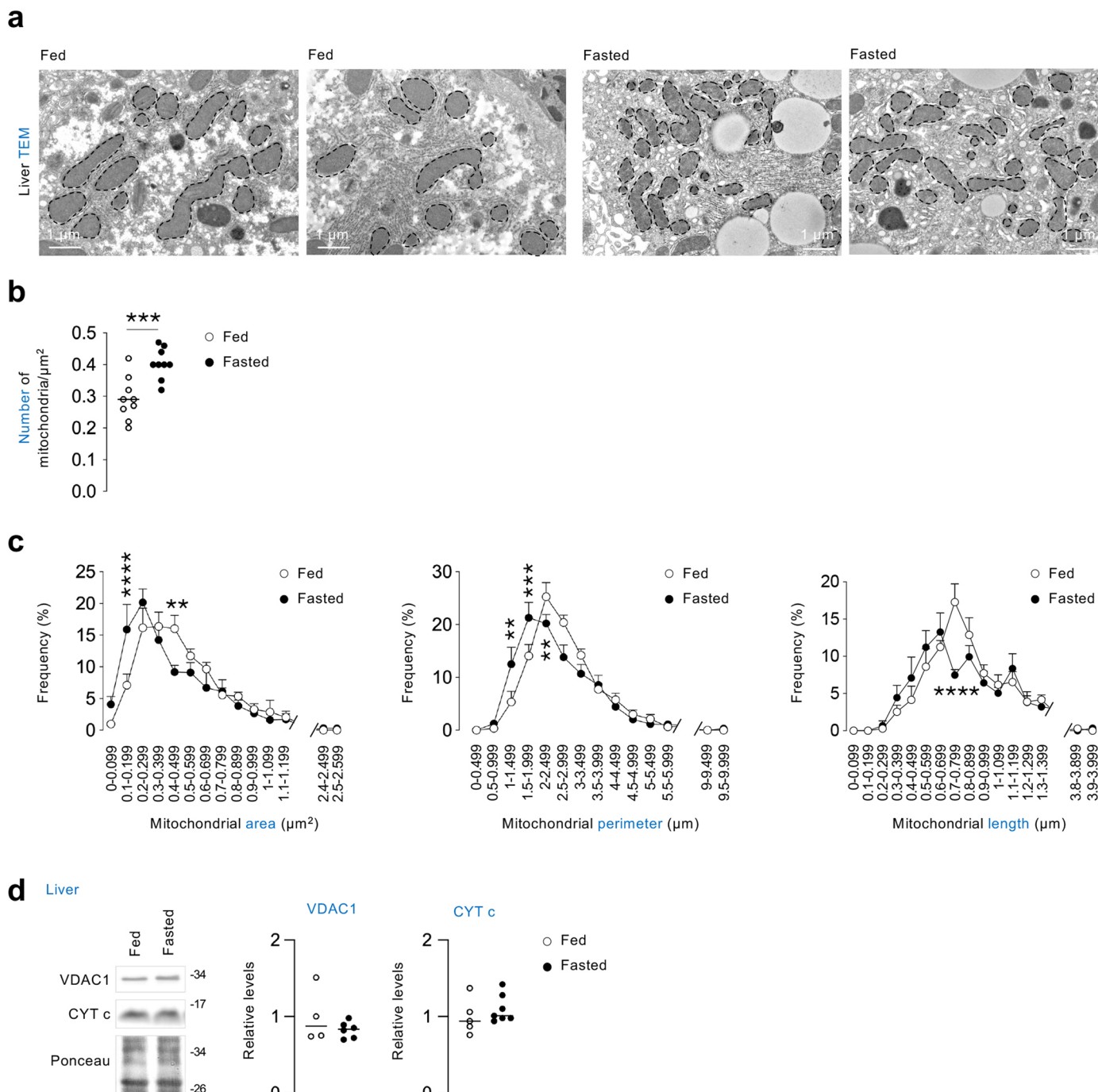

**Extended Data Fig. 4 | Fasting stimulates mitochondrial fission in livers. (a-c) (a)** Liver TEM of fed or 14–16 h fasted 4–7-mo-old male mice. Quantification for mitochondrial number is shown in **b**. Frequency histograms depicting distribution of mitochondrial area, perimeter and length are shown in **c**. Mean ± SEM is shown. For **a-c** (n = 9 mice). **(d)** Representative IB and quantification for indicated mitochondrial markers in livers of fed or 14–16 h fasted mice. *N* values for number of mice analyzed for individual proteins and conditions are indicated in parentheses. VDAC1: fed (n = 4) and fasted (n = 6); CYT c: fed (n = 5) and fasted (n = 7). Ponceau is the loading control. Individual replicates and mean values are shown in **b** and **d**. **P < 0.01, ***P < 0.001 and ****P < 0.0001, two-tailed unpaired Student's t-test **(b)**; 2-way ANOVA followed by Tukey's multiple comparisons test **(c)**. Please refer to Supplementary Table 10_ statistical summary. Source numerical data are available in SourceData_Table 1, and unprocessed blots are available in Source Data Extended Data Fig. 4.

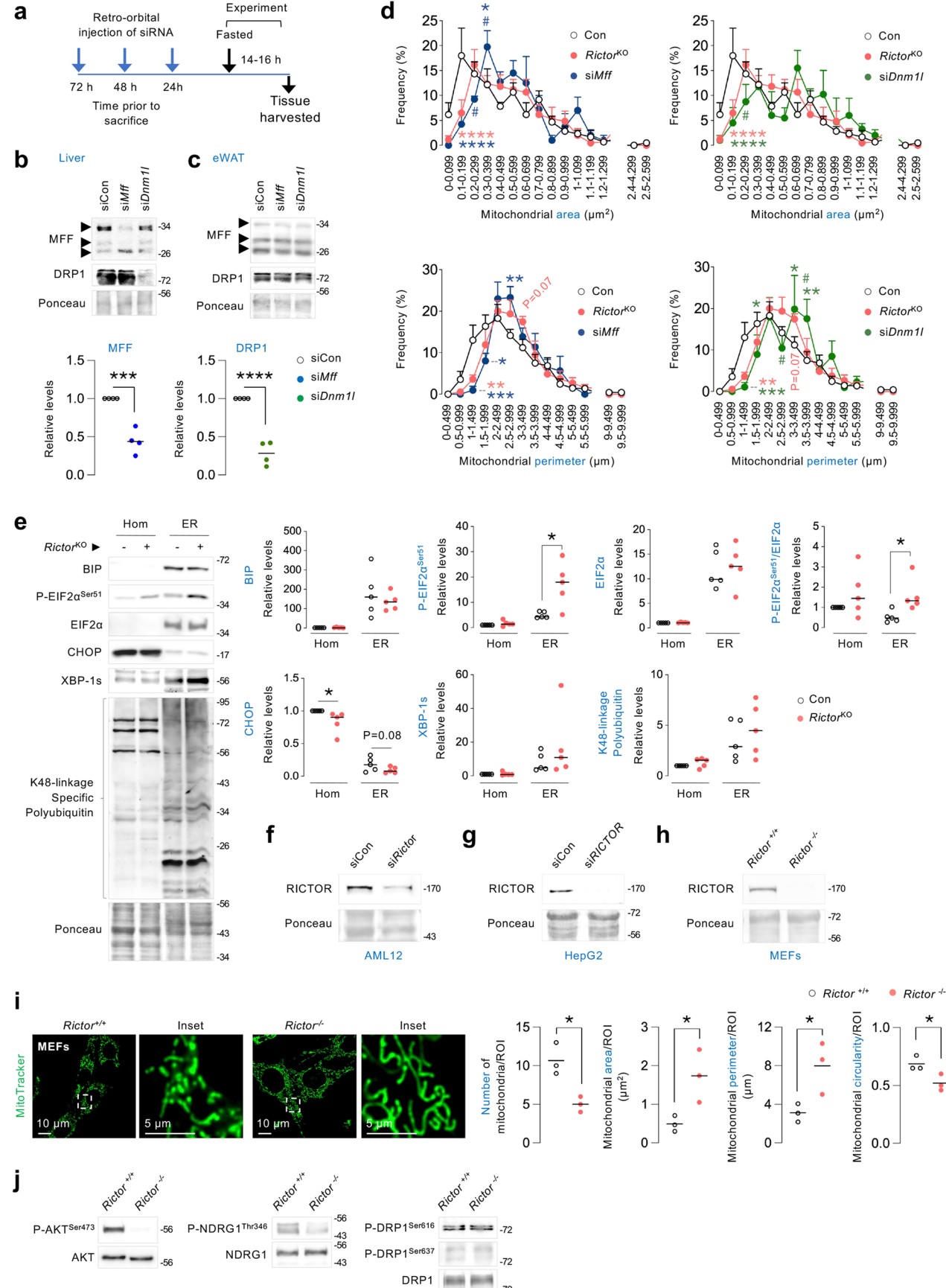

**Extended Data Fig. 5 | See next page for caption.**

**Extended Data Fig. 5 | Fasting-induced mitochondrial fission is mTORC2-dependent. (a)** Experimental plan and **(b-c)** representative IB for indicated proteins in **(b)** livers and **(c)** eWAT of 4–9 mo-old male mice injected with siCon, si*Mff* or si*Dnm1l* and fasted for 14–16 h (n = 4 mice). Quantifications for loss of MFF and DRP1 protein levels in liver are shown**. (d)** Frequency histograms depicting distribution of mitochondrial area and perimeter in Con, *Rictor*^KO and si*Mff* or si*Dnm1l* livers. *N* values for number of mice are in parenthesis: Con (n = 6), *Rictor*^KO (n = 6), si*Mff* (n = 4), and si*Dnm1l* (n = 4). Mean ± SEM is shown. **(e)** IB and quantification for indicated ER-stress and proteostasis markers in Hom and ER fractions from livers of 6 mo-old Con and *Rictor*^KO mice fasted for 14–16 h (n = 5 mice). **(f-h)** Representative IB for RICTOR in siCon or si*Rictor* **(f)** AML12 or **(g)** HepG2 cells, or in **(h)** *Rictor*^+/+ and *Rictor*^-/- MEFs. RICTOR blots are representative of independent experiments obtaining similar results in AML12 (n = 4), HepG2

(n = 3), and MEF (n = 7) cells. **(i)** Representative confocal images of *Rictor*^+/+ and *Rictor*^-/- MEFs cultured in serum-free medium for 30 min in presence of MitoTracker green. Magnified insets are shown. Quantification for mitochondrial number and mitochondria size/shape descriptors is shown (*Rictor*^+/+ 36 cells and *Rictor*^-/- 43 cells from n = 3 independent experiments). **(j)** Representative IB for indicated proteins in *Rictor*^+/+ and *Rictor*^-/- MEFs (n = 3 independent experiments). Ponceau is loading control. Individual replicates and mean values are shown. *P < 0.05, **P < 0.01, ***P < 0.001 and ****P < 0.0001 are versus Con; ^#P < 0.05 versus *Rictor*^KO, two-tailed unpaired student's t-test **(b, e** and **i)**; One-way ANOVA followed by Tukey's multiple comparisons test **(d)**. Please refer to Supplementary Table 10_statistical summary. Source numerical data are available in SourceData_Table 1, and unprocessed blots are available in Source Data Extended data Fig. 5.

**a**

Number of phosphosites
*n*=4,553   Up = 145   Down = 164

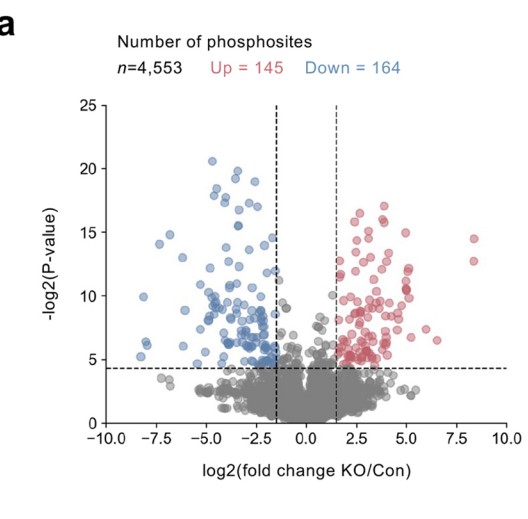

**b**

Liver BCL2/adenovirus E1B 19kDa protein-interacting protein (BNIP3):
-log2 transformed P-value

| Phospho-site | Fold-change KO/Con | P-value | Motif |
|---|---|---|---|
| Thr66 | -0.38 | 0.49 | SHCDSPPRSQTPQDTNRAEID |
| Ser77 | -3.54 | 2.25 | PQDTNRAEIDSHSFGEKNSTL |
| Ser79 | -1.57 | **5.93** | DTNRAEIDSHSFGEKNSTLSE |
| Ser85 | -3.96 | 2.28 | IDSHSFGEKNSTLSEEDYIER |
| Thr86 | 2.18 | 1.16 | DSHSFGEKNSTLSEEDYIERR |
| Ser88 | -6.07 | **8.87** | HSFGEKNSTLSEEDYIERRRE |

**c**

Liver N-Myc Downstream Regulated Gene 1 (NDRG1):
-log2 transformed P-value

| Phospho-site | Fold-change KO/Con | P-value | Motif |
|---|---|---|---|
| Thr328 | -0.87 | 1.81 | ASMTRLMRSRTASGSSVTSLE |
| Ser330 | -0.80 | 1.77 | MTRLMRSRTASGSSVTSLEGT |
| Ser332 | -1.30 | 1.62 | RLMRSRTASGSSVTSLEGTRS |
| Ser333 | 0.11 | 0.09 | LMRSRTASGSSVTSLEGTRSR |
| Ser336 | -0.91 | 1.43 | SRTASGSSVTSLEGTRSRSHT |

**d**

MAMs

Con   *Rictor*^KO

MAM fractions

nLC-MS/MS

**e**

SRTASGSSVTS(p)LEGTRSR

**f**

Liver — MAMs

○ Con
● *Rictor*^KO

NDRG1^Ser336 (Relative units)

**g**

Liver

Hom   **MAM**
Fast (h) ▶  0  3  14   0  3  14

NDRG1 — -43
CALRETICULIN — -56
FACL4 — -72
TUBULIN — -56
Ponceau — -56 / -43

NDRG1 Relative levels

Fast (h) ▶  0  3  14   0  3  14
Hom   **MAM**

**h**

NIH3T3

siCon   si*Rictor*

+ FLAG-NDRG1^WT

0.25 mM Oleic acid (2.5 h)

Pull down using FLAG-M2 agarose

Phosphoproteomics

**i**

siCon 1   RT: 61.32  AA: 12355344  BP: 716.8186
RT: 59.73  AA: 156425  BP: 716.8139
RT: 62.75  AA: 60987  BP: 716.8251

si*Rictor* 1   RT: 61.16  AA: 596117  BP: 716.8166

siCon 2   RT: 60.96  AA: 17939191  BP: 716.8181

si*Rictor* 2   RT: 60.53  AA: 294542  BP: 716.8186

Time (min)

**j**

NIH3T3

○ siCon
● si*Rictor*

NDRG1^Ser336 (Relative units)

**Extended Data Fig. 6 | See next page for caption.**

**Extended Data Fig. 6 | Phosphoproteomics reveal mTORC2 phosphorylation of NDRG1 at Ser336 in MAMs. (a)** Volcano plot for phosphoproteomics in 14–16 h fasted Con and *Rictor*$^{KO}$ livers. The numbers indicate total phosphosites. Red and blue dots represent significantly increased (P < 0.05 and log2(fold change)>1.5) and decreased (P < 0.05 and log2(fold change)<1.5) phosphosites, respectively. **(b, c)** Tables showing fold-change and log2 transformed P-values for indicated phosphorylations on liver **(b)** BNIP3, and **(c)** NDRG1 from 3–4-mo-old liver-specific *Rictor*$^{KO}$ mice fasted for 14–16 h. For **a-c** (n = 3 mice). **(d)** MAMs of 4–6 mo-old Con and *Rictor*$^{KO}$ livers from 14–16 h fasted mice were subjected to phosphoproteomics. **(e)** Annotated MS/MS spectrum of phosphopeptide SRTASGSSVT**S(p)**LEGTRSR in NDRG1, wherein **S(p)** represents phosphorylated Ser336. **(f)** Quantification for SRTASGSSVT**S(p)**LEGTRSR peptide in NDRG1 in MAMs is shown, with relative abundance of SRTASGSSVT**S(p)**LEGTRSR in Con or *Rictor*$^{KO}$ MAMs. For **d-f** (n = 3 samples wherein 2 livers were pooled to generate 1 sample). **(g)** Representative IB and quantifications for indicated proteins in Hom and MAM fractions from livers of 2–10 mo-old male mice fed or fasted for indicated time points. *N* values for number of mice at each time point are in parenthesis: Hom 0 h and 14 h (n = 5); Hom 3 h (n = 6); and all time points for MAMs (n = 6). Ponceau is the loading control. **(h)** Experimental plan to pulldown FLAG-NDRG1$^{WT}$ in siCon or si*Rictor* NIH3T3 cells co-transfected with FLAG-NDRG1$^{WT}$ plasmid for assessment of phosphorylation of FLAG-NDRG1$^{WT}$ via phosphoproteomics. FLAG-NDRG1$^{WT}$ pulled-down from total lysates of serum-deprived siCon or si*Rictor* NIH3T3 cells in presence of OA for 2.5 h. **(i)** Representative extracted ion chromatograms of SRTASGSSVT**S(p)**LEGTRSR in FLAG-NDRG1 in siCon and si*Rictor* NIH3T3 cells, and **(j)** quantification for relative abundance of SRTASGSSVT**S(p)**LEGTRSR peptide from pulled-down FLAG-NDRG1 from siCon and si*Rictor* cells expressing FLAG-NDRG1$^{WT}$. For **h-j** (n = 3 independent experiments). Individual replicates and means are shown. *P < 0.05, two-tailed unpaired Student's t-test (**a-c, f** and **j**). p=phosphorylation. Please refer to Supplementary Table 10_statistical summary. Please refer to Supplementary Table 4 (**a-c**), Supplementary Table 6 (**e** and **f**), and Supplementary Table 7 (**i** and **j**). Source numerical data are available in SourceData_Table 1, and unprocessed blots are available in Source Data Extended Data Fig. 6.

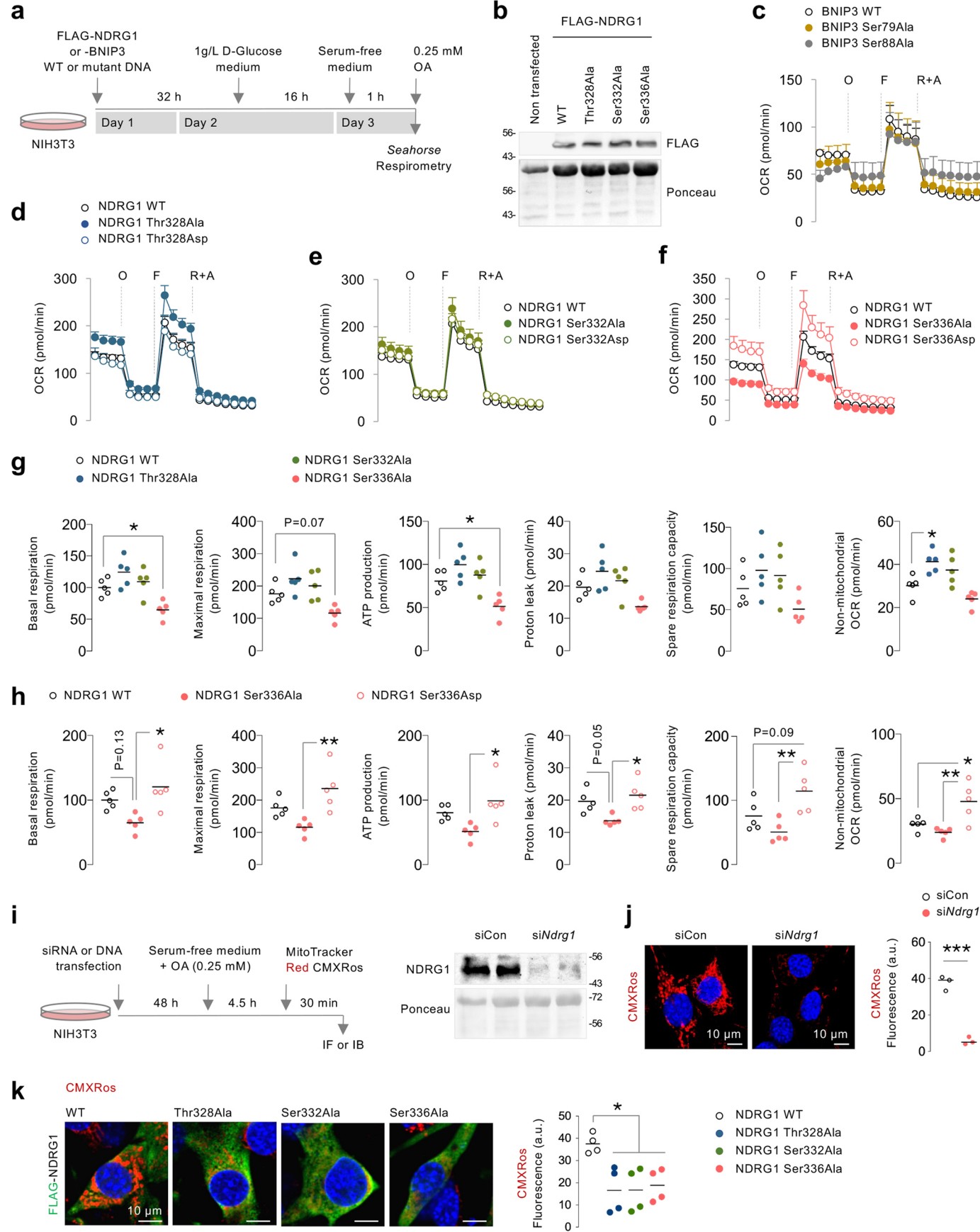

**Extended Data Fig. 7 | See next page for caption.**

**Extended Data Fig. 7 | Phosphorylation of NDRG1 at Ser336 supports mitochondrial respiration and membrane potential. (a)** Experimental plan for Seahorse mitochondrial stress tests shown through **c-h**. **(b)** Representative IB for FLAG in NIH3T3 cells expressing the indicated FLAG-tagged NDRG1 wild type (WT) or FLAG-tagged Ser/Thr>Ala mutant of NDRG1 (n = 3 independent experiments). **(c)** Seahorse mitochondrial stress test in serum-deprived NIH3T3 cells expressing FLAG-tagged BNIP3 WT or FLAG-tagged Ser79Ala or Ser88Ala mutant BNIP3 in presence of 0.25 mM OA, followed by sequential addition of oligomycin (O), FCCP (F), and rotenone + antimycin (R + A) to assess mitochondrial respiratory function (n = 5 independent experiments). **(d-f)** Seahorse mitochondrial stress tests in NIH3T3 cells expressing the indicated FLAG-tagged NDRG1 WT or Ser/Thr>Ala or Ser/Thr>Asp mutant of NDRG1. Quantifications for mitochondrial respiratory function in cells expressing **(g)** FLAG-tagged NDRG1 WT or Ser/Thr>Ala mutants of NDRG1, or **(h)** FLAG-tagged NDRG1 WT or Ser336Ala and Ser336Asp mutants of NDRG1 are shown. For **d-h**

(n = 5 independent experiments). **(i)** Cartoon depicting experimental plan for confocal microscopy performed in **j** and **k**. Representative blots for NDRG1 in siCon and si$Ndrg1$ NIH3T3 cells are shown in **i, right** (n = 5 independent experiments). **(j)** MitoTracker CMXRos red fluorescence in siCon or si$Ndrg1$ cells (siCon 58 cells and si$Ndrg1$ 62 cells from n = 3 independent experiments). **(k)** MitoTracker CMXRos red fluorescence in FLAG (green fluorescence)-tagged WT or phosphorylation-deficient mutants of NDRG1 (NDRG1$^{WT}$ 37 cells, NDRG1$^{Thr328Ala}$ 34 cells, NDRG1$^{Ser332Ala}$ 32 cells, and NDRG1$^{Ser336Ala}$ 33 cells from n = 4 independent experiments). Ponceau is loading control. For **c-f**, values are mean ± SEM. For **g, h, j** and **k**, individual replicates and means are shown. *P < 0.05, **P < 0.01 and ***P < 0.001, One-way ANOVA followed by Tukey's multiple comparisons test (**g, h** and **k**); two-tailed unpaired Student's t-test (**j**). Please refer to Supplementary Table 10_statistical summary. Source numerical data are available in SourceData_Table 1, and unprocessed blots are available in Source Data Extended Data Fig. 7.

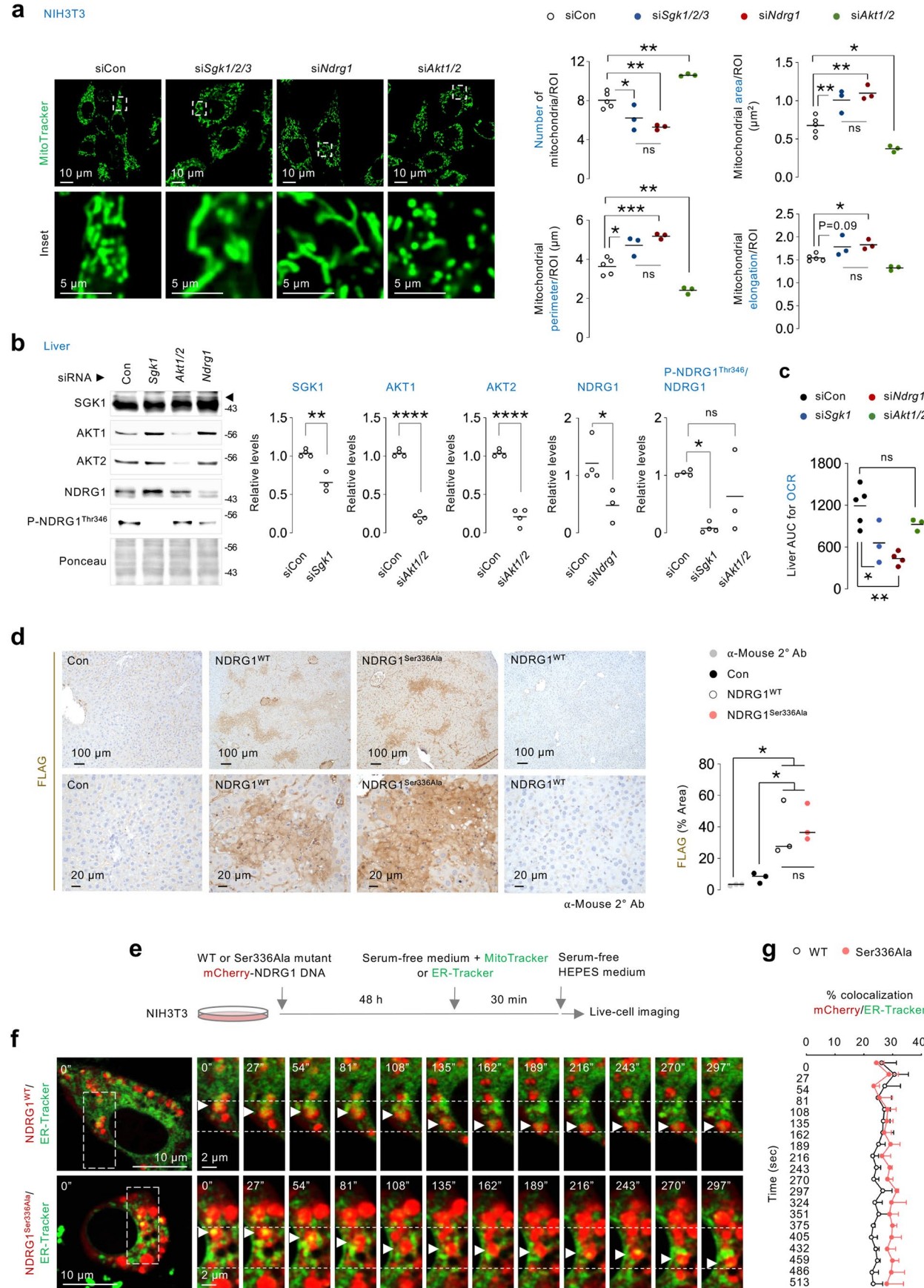

**Extended Data Fig. 8 | See next page for caption.**

**Extended Data Fig. 8 | Silencing NDRG1 and SGK1, but not AKT1/2, recapitulates mitochondrial fission failure observed in *Rictor* silenced cells.**
**(a)** Representative confocal images of NIH3T3 cells silenced for *Sgk1/2/3, Ndrg1* or *Akt1/2* and cultured in serum-free medium for 30 min in presence of MitoTracker green. Magnified insets are shown. Quantifications for mitochondrial number and mitochondrial size/shape descriptors are shown (siCon 71 cells, si*Sgk1/2/3* 36 cells, si*Ndrg1* 62 cells, and si*Akt1/2* 36 cells from n = 5 (siCon) and n = 3 (si*Sgk1/2/3*, si*Ndrg1*, and si*Akt1/2*) independent experiments). **(b)** IB and quantifications for indicated proteins in livers of 3–4 mo-old male mice injected with siRNAs against *Sgk1*, A*kt1/2* or *Ndrg1* and subjected to 14–16 h fasting. *N* values for number of mice analyzed for indicated proteins are in parentheses. SGK1: siCon (n = 4) and si*Sgk1* (n = 3); AKT1 and 2: siCon (n = 4) and si*Akt1/2* (n = 4); NDRG1: siCon (n = 4) and si*Ndrg1* (n = 3); P-NDRG1$^{Thr346}$/NDRG1: siCon (n = 4), si*Sgk1* (n = 4) and si*Akt1/2* (n = 3). **(c)** AUC for OCR in livers of 14–16 h fasted Con (n = 5), si*Sgk1* (n = 3), si*Ndrg1* (n = 4), and si*Akt1/2* (n = 3) mice. **(d)** Immunohistochemistry and quantification for equivalent FLAG expression in livers silenced for endogenous *Ndrg1* and then injected with FLAG-tagged WT or Ser336Ala NDRG1 plasmid (n = 3 mice). 2° Ab-only control is shown. **(e)** Cartoon depicting experimental plan performed in Fig. 5a and Extended Data Fig. 8f. **(f)** Representative live-cell imaging of mCherry-NDRG1$^{WT}$ or mCherry-NDRG1$^{Ser336Ala}$ and ER-Tracker green in NIH3T3 cells cultured in serum-free medium for 30 min. Magnified insets are shown. White arrowheads: NDRG1 (mCherry)/ ER (ER-Tracker) contacts. **(h)** Quantification for % colocalization of mCherry with ER-tracker in **g**. Values are mean ± SEM (n = 3 independent experiments). Please refer to Supplementary Video 7 (mCherry-NDRG1$^{WT}$/ER-Tracker), and Supplementary Video 8 (mCherry-NDRG1$^{Ser336Ala}$/ER-Tracker). Ponceau is loading control. Individual replicates and means are shown. *P < 0.05, **P < 0.01, ***P < 0.001 and ****P < 0.0001, One-way ANOVA followed by Tukey's multiple comparisons test **(a, c** and **d)**; two-tailed unpaired Student's t-test **(b);** 2-way ANOVA followed by Tukey's multiple comparisons test **(g)**. ns=not significant. Please refer to Supplementary Table 10_statistical summary. Source numerical data are available in SourceData_Table 1, and unprocessed blots are available in Source Data Extended Data Fig. 8.

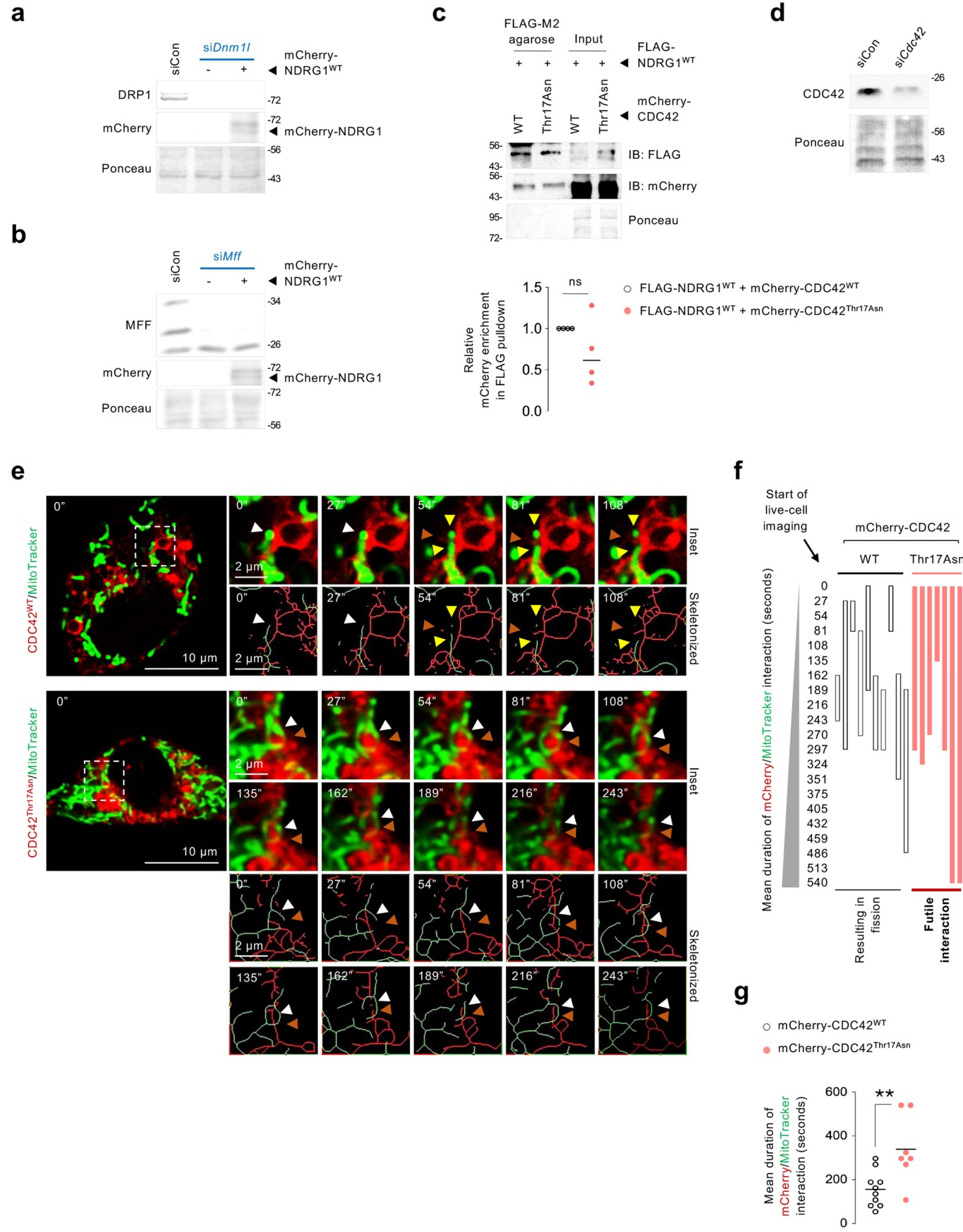

**Extended Data Fig. 9 | See next page for caption.**

**Extended Data Fig. 9 | Validating silencing of *Dnml1*, *Mff* and *Cdc42* in NIH3T3 cells, and time-lapse imaging showing loss of fission in cells expressing the mcherry-CDC42$^{Thr17Asn}$ GTP-binding mutant.** (**a, b**) Representative IB for indicated proteins in NIH3T3 cells silenced for (**a**) *Dnml1* or (**b**) *Mff* and expressing mCherry-tagged NDRG1$^{WT}$. Blots are representative of n = 3 independent experiments obtaining similar results. (**c**) IB for FLAG and mCherry in OA-treated (2.5 h) NIH3T3 cells co-expressing FLAG-NDRG1$^{WT}$ and mCherry-CDC42$^{WT}$ or mCherry-CDC42$^{Thr17Asn}$ mutant and subjected to pulldown of FLAG using FLAG M2 agarose (n = 4 independent experiments). Quantification for relative enrichment of mCherry in FLAG pulldowns was calculated by normalizing the densitometry value of mCherry to the densitometry value of FLAG. (**d**) Representative IB for indicated proteins in NIH3T3 cells silenced for *Cdc42* and expressing mCherry-tagged NDRG1$^{WT}$. CDC42 blot is representative of n = 3 independent experiments obtaining similar results. (**e**) Representative live-cell imaging in cells expressing mcherry-CDC42$^{WT}$ or dominant negative mcherry-CDC42$^{Thr17Asn}$ mutant in presence of MitoTracker green to label mitochondria. Red arrowheads: CDC42 (mCherry). White arrowheads: CDC42 (mCherry)/mitochondria (MitoTracker) contacts prior to fission. Yellow arrowheads: divided mitochondria after contact with CDC42 (mCherry). Magnified insets are shown. Please refer to Supplementary Video 14 (mCherry-CDC42$^{WT}$/MitoTracker), and Supplementary Video 15 (mCherry-CDC42$^{Thr17Asn}$/MitoTracker). (**f**) Graphical representation for duration of interaction between mCherry-CDC42$^{WT}$ or mCherry-CDC42$^{Thr17Asn}$ and mitochondria (MitoTracker), and whether interactions lead to division or are futile. (**g**) Quantification for mean duration of interaction (mCherry-CDC42$^{WT}$ 10 cells, and mCherry-CDC42$^{Thr17Asn}$ 7 cells from n = 3 independent experiments. Each tracked cell was monitored on an independent plate). Ponceau is loading control. Individual replicates and means are shown. **P < 0.01, two-tailed unpaired Student's t-test (**c** and **g**). ns=not significant. Please refer to Supplementary Table 10_statistical summary. Source numerical data are available in SourceData_Table 1, and unprocessed blots are available in Source Data Extended Data Fig. 9.

**a**

| Position | Accession | Description | Gene Symbol | Fold Change (GFP-CDC42/GFP) | P-value | Score | |
|---|---|---|---|---|---|---|---|
| 4 | Q9JM96 | CDC42 effector protein 4 | *Cdc42ep4* | 5.96 | 8.51 | 50.71 | Significantly enriched in CDC42 pull downs |
| 5 | Q91W92 | CDC42 effector protein 1 | *Cdc42ep1* | 6.08 | 7.31 | 44.46 | |
| 6 | Q9JKF1 | RAS GTPase-activating-like protein IQGAP1 | *Iqgap1* | 7.92 | 6.61 | 52.36 | |
| 7 | Q99PT1 | RHO GDP-dissociation inhibitor 1 | *Arhgdia* | 6.39 | 6.33 | 40.42 | |
| 12 | Q8JZX9 | CDC42 effector protein 2 | *Cdc42ep2* | 4.36 | 5.40 | 23.54 | |
| 22 | Q9JI08 | Bridging integrator 3 | *Bin3* | 3.77 | 3.94 | 14.83 | |
| 24 | Q9DAK3 | RHO-related BTB domain-containing protein 1 | *Rhobtb1* | 6.61 | 3.90 | 25.78 | |
| 40 | Q61599 | RHO GDP-dissociation inhibitor 2 | *Arhgdib* | 4.98 | 3.27 | 16.28 | |

*Cut off (P-value 4.32)*

**b**

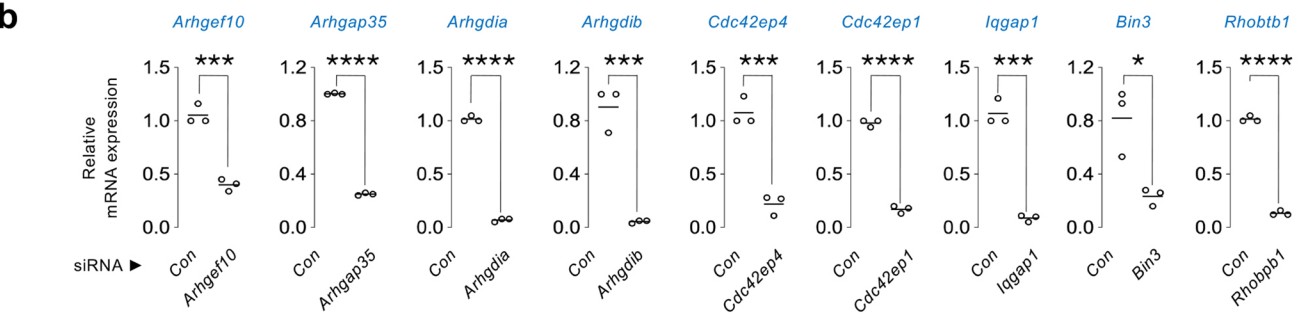

**c**

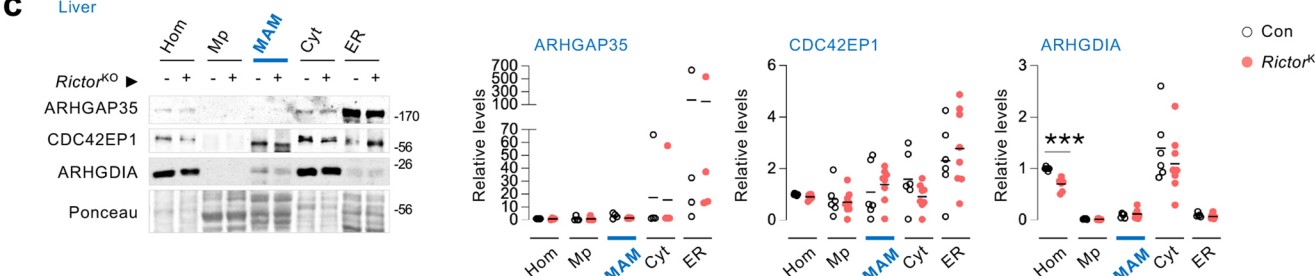

**d**

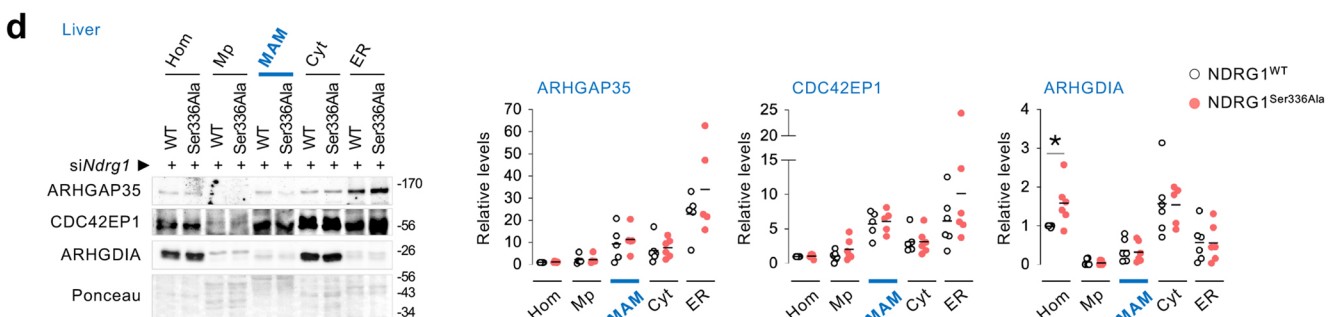

**Extended Data Fig. 10 | An siRNA screen to target interacting partners of CDC42 reveals effectors and regulators of CDC42 controlling mitochondrial dynamics. (a)** Significantly enriched interacting partners of CDC42 that belong to the RHO family of GTPases. A log2 transformed P-value cut-off of 4.32 was used as threshold (n = 4 independent experiments). **(b)** qPCR in NIH3T3 cells to validate the silencing of selected CDC42-binding partners identified by proteomics (n = 3 independent experiments). **(c, d)** IB and quantification for indicated proteins in Hom, Mp, MAMs, Cyt, and ER fractions from livers of **(c)** 5–6 mo-old Con or *Rictor*^KO^ male mice, and **(d)** 3–4 mo-old NDRG1^WT^ or NDRG1^Ser336Ala^ male mice co-injected with siRNA against endogenous *Ndrg1* and fasted for 14–16 h. *N* values for number of mice analyzed for individual proteins in indicated fractions are in parentheses. For **c**, ARHGAP35: all fractions from Con (n = 4) and *Rictor*^KO^ (n = 4) mice; CDC42EP1: all fractions, Con (n = 6) and *Rictor*^KO^ (n = 8) mice except *Rictor*^KO^ Hom where n = 7 mice; ARHGDIA: all fractions from Con

(n = 6) mice except Mp, MAMs and ER where n = 5 mice, and all fraction from *Rictor*^KO^ (n = 8) mice except Hom and ER where n = 7 mice. For **d**, ARHGAP35: all fractions from NDRG1^WT^ and NDRG1^Ser336Ala^ mice are n = 6 except Mp, MAMs and ER from NDRG1^WT^ mice where n = 5; CDC42EP1: all fractions from NDRG1^WT^ and NDRG1^Ser336Ala^ mice are n = 6 except MAMs in both groups where n = 5; ARHGDIA: all fractions from NDRG1^WT^ and NDRG1^Ser336Ala^ mice are n = 6 except Cyt in NDRG1^Ser336Ala^ mice where n = 5. Ponceau is loading control. Individual replicates and means are shown. *P < 0.05, ***P < 0.001 and ****P < 0.0001, two-tailed unpaired Student's t-test **(a-d)**. GAP: GTPase activating protein; GDI: GDP dissociation inhibitors. Please refer to Supplementary Table 10_statistical summary, and Supplementary Table 9. Source numerical data are available in SourceData_Table 1, and unprocessed blots are available in Source Data Extended Data Fig. 10.

# Reporting Summary

## Statistics

For all statistical analyses, confirm that the following items are present in the figure legend, table legend, main text, or Methods section.

| n/a | Confirmed | |
|---|---|---|
| ☐ | ☒ | The exact sample size ($n$) for each experimental group/condition, given as a discrete number and unit of measurement |
| ☐ | ☒ | A statement on whether measurements were taken from distinct samples or whether the same sample was measured repeatedly |
| ☐ | ☒ | The statistical test(s) used AND whether they are one- or two-sided<br>*Only common tests should be described solely by name; describe more complex techniques in the Methods section.* |
| ☐ | ☒ | A description of all covariates tested |
| ☐ | ☒ | A description of any assumptions or corrections, such as tests of normality and adjustment for multiple comparisons |
| ☐ | ☒ | A full description of the statistical parameters including central tendency (e.g. means) or other basic estimates (e.g. regression coefficient) AND variation (e.g. standard deviation) or associated estimates of uncertainty (e.g. confidence intervals) |
| ☐ | ☒ | For null hypothesis testing, the test statistic (e.g. $F$, $t$, $r$) with confidence intervals, effect sizes, degrees of freedom and $P$ value noted<br>*Give P values as exact values whenever suitable.* |
| ☒ | ☐ | For Bayesian analysis, information on the choice of priors and Markov chain Monte Carlo settings |
| ☒ | ☐ | For hierarchical and complex designs, identification of the appropriate level for tests and full reporting of outcomes |
| ☒ | ☐ | Estimates of effect sizes (e.g. Cohen's $d$, Pearson's $r$), indicating how they were calculated |

*Our web collection on statistics for biologists contains articles on many of the points above.*

## Software and code

Policy information about availability of computer code

| | |
|---|---|
| Data collection | Commercial acquisition softwares accompanying the following devices were used to collect data. Acquisition softwares for (a) Zeiss Axiolab 5 microscope with Axiocam 305 color camera for immunohistochemistry (Zeiss ZEN v3.7), (b) Leica TCS SP8 Confocal Laser Scanning Microscope for fluorescence images and live-cell videos (LAS X v.5.7.23225), (c) Zeiss Supra 40 Field Emission Scanning Electron Microscope to acquire transmission electron microscope images and videos (Zeiss SmartSEM v6.0) and 3D Modeling (3DMOD v4.9.10), (d) XF96 and X24 Seahorse analyzers (Agilent Technologies) to collect oxygen consumption rates (WAVE Pro v10.0.1.84; v2.6.1.56, respectively), and normalized to cell number determined by using Synergy HTX (BioTek) multi-mode plate reader (Gen5 v3.12), (e) StepOnePlus™ Real-Time PCR System (Applied Biosystems) for mRNA expression (StepOne v2.3). |
| Data analysis | Microsoft Excel v16.48, Microsoft Word v16.48, Microsoft PowerPoint v16.47 were used for processing and representation. Prism v8.4.3. was used for data analyses. Endnote X9.3.3. was used for assimilating citations. ImageJ v2.0.0-rc-69/1.52p was used for the quantification of western blot images. MitoTracker™, Red CMXRos fluorescence intensity, and mitochondria and/or ER shape parameters. The enrichment map was generated in Cytoscape v3.8.1 using Enrichment map plugin v3.3.0. Data handling and statistical analyses were performed using Python v.3.7.4 and scientific python stack: SciPy v.1.3.1, NumPy v.1.17.2, and Matplotlib v.3.1.1. Phosphosites showing significant regulation between groups were used to predict the kinase responsible for their catalysis using the iGPS software. |

For manuscripts utilizing custom algorithms or software that are central to the research but not yet described in published literature, software must be made available to editors and reviewers. We strongly encourage code deposition in a community repository (e.g. GitHub). See the Nature Portfolio guidelines for submitting code & software for further information.

## Data

Policy information about availability of data

 All manuscripts must include a data availability statement. This statement should provide the following information, where applicable:
- Accession codes, unique identifiers, or web links for publicly available datasets
- A description of any restrictions on data availability
- For clinical datasets or third party data, please ensure that the statement adheres to our policy

The mass spectrometry proteomics data have been deposited to the ProteomeXchange Consortium via the PRIDE88 partner repository, and data are available via ProteomeXchange with identifier PXD041696. All the Python codes used in proteomic analyses are fully available on Github (https://github.com/MathieuBo/mTorc2_mito_fission). No restrictions.

## Human research participants

Policy information about studies involving human research participants and Sex and Gender in Research.

| | |
|---|---|
| Reporting on sex and gender | N/A |
| Population characteristics | N/A |
| Recruitment | N/A |
| Ethics oversight | N/A |

Note that full information on the approval of the study protocol must also be provided in the manuscript.

# Field-specific reporting

Please select the one below that is the best fit for your research. If you are not sure, read the appropriate sections before making your selection.

☒ Life sciences ☐ Behavioural & social sciences ☐ Ecological, evolutionary & environmental sciences

For a reference copy of the document with all sections, see nature.com/documents/nr-reporting-summary-flat.pdf

# Life sciences study design

All studies must disclose on these points even when the disclosure is negative.

| | |
|---|---|
| Sample size | A minimum of three independent experiments were performed unless otherwise stated. N numbers indicate biological replicates. Sample sizes were determined based on power calculations from previous studies in our laboratory. Measurements were taken from distinct samples except from live cell imaging experiments when the same cell was recorded for a period of time. |
| Data exclusions | Statistically-validated outliers were determined (via Prism v8.4.3) by Grubb's method (alpha=0.05) and those verified were eliminated. |
| Replication | All replication attempts in this study were successful. In vivo experiments comprising Con and RictorKO mice were performed 6 independent times including a minimum of n=3 mice per group per repetition. Experiments comprising mice expressing WT or Ser366Ala mutant NDRG1 were performed 3 independent times including a minimum of n=3 mice per group per repetition. Experiments comprising corn oil gavage were performed 6 times including a minimum of n=3 mice per group per repetition. Experiments comprising Bodipy FL C16-gavage were performed 3 independent times including a minimum of n=3 mice per group per repetition. |
| Randomization | All mice used in this study were randomly assigned to the different experimental groups. Equal sex and age were maintained across the different animal groups within each of the experiments. |
| Blinding | The persons performing the electron microscopy imaging were unaware of the sample identity. For experiments comprising confocal imaging acquisition and image/video processing, experimental settings (i.e., laser intensity, channel intensity, etc.) were standardized using the control cells as a reference, and similar settings were applied to the rest of the experimental groups/conditions within the experiment. Quantification for mitochondrial number and shape descriptors was performed blindly. For the rest of the analyses, blinding was not relevant because sample identification was required to conduct the analyses, e.g., for MAM preparations which require pooling to mice and western blotting. |

# Reporting for specific materials, systems and methods

We require information from authors about some types of materials, experimental systems and methods used in many studies. Here, indicate whether each material, system or method listed is relevant to your study. If you are not sure if a list item applies to your research, read the appropriate section before selecting a response.

## Materials & experimental systems

| n/a | Involved in the study |
|---|---|
| ☐ | ☒ Antibodies |
| ☐ | ☒ Eukaryotic cell lines |
| ☒ | ☐ Palaeontology and archaeology |
| ☐ | ☒ Animals and other organisms |
| ☒ | ☐ Clinical data |
| ☒ | ☐ Dual use research of concern |

## Methods

| n/a | Involved in the study |
|---|---|
| ☒ | ☐ ChIP-seq |
| ☒ | ☐ Flow cytometry |
| ☒ | ☐ MRI-based neuroimaging |

## Antibodies

| | |
|---|---|
| Antibodies used | All primary and secondary antibodies were commercially available from the following sources: Rabbit polyclonal anti-AKT (#9272; Lot #30), Rabbit polyclonal anti-AKT1 (#2938; Lot #4), Rabbit polyclonal anti-AKT2 (#30638), Rabbit monoclonal anti-Phospho AKT (Thr308) (#13038; Lot #9), Rabbit monoclonal anti-Phospho AKT (Ser473) (# 4060; Lot #27), Rabbit polyclonal anti-AMPK? (#2532; Lot #21), Rabbit polyclonal anti-Phospho AMPK? (Thr172) (#2531; Lot #19), Rabbit polyclonal anti-p190-A RhoGAP (ARHGAP35) (#2513; Lot #1), Rabbit monoclonal anti-CDC42 (#2466; Lot #6), Rabbit monoclonal anti-CHOP (#5554; Lot #5), Rabbit monoclonal anti-CYTOCHROME c (#11940; Lot #4), Rabbit polyclonal anti-DYNAMIN I/II (# 2342; Lot #1), Rabbit monoclonal anti-DRP1 (#8570; Lot #5), Rabbit monoclonal anti-Phospho-DRP1 (Ser616) (# 4494; Lot #4), Rabbit monoclonal anti-Phospho-DRP1 (Ser637) (#6319; Lot #2), Rabbit monoclonal anti-DYKDDDDK Tag (#14793; Lot #5), Rabbit polyclonal anti-Phospho-eIF2α (Ser51) (#9721; Lot #21), Rabbit polyclonal anti-K48-linkage Specific Polyubiquitin (#4289; Lot #2), Rabbit monoclonal anti-MFF (#84580; Lot #1), Rabbit monoclonal anti-MFN2 (#9482; Lot #4), Rabbit polyclonal Phospho-NDRG1 (Thr346) (#5482; Lot #5), Rabbit monoclonal anti-OPA1 (#80471; Lot #1), Mouse monoclonal anti-PARKIN (#4211; Lot #8), Rabbit monoclonal anti-PKA C-α (#5842; Lot #4), Rabbit monoclonal anti-Phospho-PKA C (Thr197) (#5661, Lot # 3), Rabbit polyclonal anti-PKCα (#2056; Lot #5), Rabbit monoclonal anti-PKCδ (#9616; Lot #3), Rabbit monoclonal anti-PKCζ (#9368; Lot #4), Rabbit polyclonal anti-Phospho-PKCα/β II (Thr638/641) (#9375; Lot #5), Rabbit polyclonal anti-PKCδ (Thr505) (#9374; Lot #7), Rabbit polyclonal anti-PKCδ/θ (Ser643/676) (#9376; Lot #6), Rabbit polyclonal anti-Phospho-PKCζ/λ (Thr410/403) (#9378; Lot #10), Rabbit polyclonal anti-p70 (#9202; Lot #21), Rabbit monoclonal anti-Phospho-p70 S6 Kinase (Thr389) (#9234; Lot #12), Rabbit monoclonal anti-RAPTOR (#2280; Lot #11), Rabbit monoclonal anti-RHOA (#2117; Lot #6), Rabbit monoclonal anti-RICTOR (#2114; Lot #7), Rabbit monoclonal anti-S6 (#2217; Lot #10), Rabbit monoclonal anti-Phospho-S6 Ribosomal Protein (Ser235/236) (#4858; Lot #21), Rabbit polyclonal anti-?-Tubulin (#2144; Lot #7) and Rabbit monoclonal anti-TSC1 (#6935; Lot #4) were purchased from Cell Signaling Technology. Rabbit polyclonal anti-FACL4 (#ab227256; Lot #1035983-1), Rabbit monoclonal anti-CALRETICULIN (#ab92516; Lot # GR3287998-1), Rabbit polyclonal Anti-MTP18/MTFP1 (#ab198217; Lot # GR3381113-12), Rabbit polyclonal anti-VDAC1 (# ab15895; Lot # GR3452674-3), and Total OXPHOS Rodent WB Antibody Cocktail (#ab110413; Lot # 2101014434) were purchased from Abcam. Mouse monoclonal anti-eIF2α (#133132) was purchased from Santa Cruz Biotechnology. Mouse monoclonal anti-BiP/GRP78 (#610978; Lot #1277761) was purchased from BD Biosciences. Mouse monoclonal anti-XBP-1s (#647502; Lot B131918) was purchased from Biolegend. Rabbit polyclonal anti-MFN1 (#GTX64398; Lot # 822103891) was purchased from GeneTex. Rabbit monoclonal anti-CDC42EP1 (#MA5-37968; Lot # XA3483164) and Rat monoclonal anti-mCherry (#M11217; Lot # XJ359389) were purchased from Invitrogen. Rabbit polyclonal anti-ARHGDIA (#MBS9413899; Lot #4801) was purchased from MyBioSource. Goat polyclonal anti-GFP (#NB100-1770; Lot #48180) was purchased from Novus Biologicals. Rabbit polyclonal anti-NDRG1 (#TA327295; Lot # 3515658101) and Rabbit polyclonal anti-SGK1 (#TA326894; Lot # 3561864002) were purchased from Origene. Rabbit polyclonal anti-Phospho-SGK1 (Thr256) (#44-1260G; Lot #2530313) was purchased from ThermoFisher Scientific. Secondary HRP Antibody Goat anti-Rabbit IgG (#074-1506; Lot # 10440068) was purchased from KPL, Secondary HRP Antibody Rabbit anti-Mouse (#61-6520; Lot # XC342755A) and Secondary HRP Antibody Goat anti-Rat (#31470; Lot # XG344754) were purchased from Invitrogen. All primary antibodies were used at a concentration of 1:1000, and secondary antibodies were used at a concentration of 1:5000. See Supplementary Table 6. |
| Validation | All primary and secondary antibodies were used following Manufacturer's instructions. Molecular size markers have been included on each of the western blots, and the molecular weights for each of the antibodies were validated as per manufacturer's datasheets. Specificity for RICTOR, RAPTOR and TSC1 antibodies was also validated using knock-out mouse models for each of the corresponding gene products. Specificity for MFF, DRPL, NDRG1, SGK1, AKT1/2, CDC42, OPA1 and MFNs antibodies was confirmed knock-down mouse models or knock-down cells for each of the corresponding gene products. Additionally, application validations can be obtained from the relevant vendor websites for the antibodies using the supplied catalog numbers. |

## Eukaryotic cell lines

Policy information about cell lines and Sex and Gender in Research

| | |
|---|---|
| Cell line source(s) | NIH3T3 cells (CRL-1658), HepG2 cells (HB-8065) and AML12 cells (CRL-2254) were used in this study and purchased from ATCC. Isolation of embryonic fibroblasts from Rictor flox/flox mice was performed as described (Bio Protoc. 2013 Sep 20;3(18):e908). |
| Authentication | The cells used in this study were not independently authenticated by us. |
| Mycoplasma contamination | All experiments in this study were performed used mycoplasma-free cells. Cell culture medium was periodically subjected to PCR using REDTaq® ReadyMix™ PCR reaction mix (Sigma; R2523) for the detection of mycoplasma. |

| Commonly misidentified lines (See ICLAC register) | No misidentified cell lines were used in this study. |

# Animals and other research organisms

Policy information about studies involving animals; ARRIVE guidelines recommended for reporting animal research, and Sex and Gender in Research

| Laboratory animals | C57BL/6 (#000664), Rictor flox/flox (#020649), Raptor flox/flox (#013188) and Tsc1 flox/flox (#005680) mice were purchased from the Jackson Laboratory. Studies were performed in 2-10-month-old male and female mice fed regular chow (5058; Lab Diet, St Louis, MO, USA) and maintained in barrier facility at 22-23 ºC under 40-60% humidity and a 12h:12h light/dark cycle. |
| Wild animals | No wild animals were use in this study. |
| Reporting on sex | Studies were performed in 2-8-month-old male and female mice. |
| Field-collected samples | No field-collected samples were used in this study. |
| Ethics oversight | This research complies with all relevant ethical regulations including animal protocol approval from the IACUC of Albert Einstein College of Medicine (Protocol Number: 00001051). |

Note that full information on the approval of the study protocol must also be provided in the manuscript.

