## [Peer Review File · Nature Cell Biology]

Peer Review Information

Journal: Nature Cell Biology

Manuscript Title: mTORC2-NDRG1-CDC42 axis couples fasting to mitochondrial fission

Corresponding author name(s): Dr Rajat Singh

Editorial Notes:

Reviewer Comments & Decisions:

Decision Letter, initial version:

*Please delete the link to your author homepage if you wish to forward this email to co-authors.

Dear Rajat,

Thank you for submitting your manuscript, "mTORC2-NDRG1-Cdc42 axis couples fasting to mitochondrial fission", to the journal. I am very, very sorry for the long delay in sharing our decision with you. As you know, we had contacted three experts covering the various aspects of the study; however, Rev#3 became unresponsive. We asked another expert to step in, and it so happened that both the unresponsive Rev#3 and the newly recruited Rev #4 sent us comments at the same time. I am really so sorry for the long delay and I am so grateful for your patience.

The manuscript has now been seen by 4 referees, who are experts in Rho GTPases (Referee #1); mTORC2/metabolism (Referee #2); mitochondria/metabolism (Referee #3), and mitochondria/MAMs (Referee #4). As you will see from their comments (attached below), they found this work of potential interest, but have raised substantial concerns, which in our view would need to be addressed with considerable revisions before we can consider publication in Nature Cell Biology.

As per our standard editorial practice, we have now discussed the referee reports in detail within the editorial team, including the chief editor, to identify key referee points that should be addressed with priority, as opposed to requests that are overruled as being beyond the scope of the current study. To guide the scope of the revisions, I have listed these points below. Our standard revision period is six months; we are committed to providing a fair and constructive peer-review process, so please feel free to contact me if you would like to discuss any of the referee comments further or if you anticipate any issues or delays addressing the reviews.

I should stress that the referees' concerns regarding the strength of the core conclusions involving changes in mTORC2 activity and its downstream mechanism would need to be addressed with experiments and data, and reconsideration of the study for this journal and re-engagement of referees would depend on the strength of these revisions. In particular, we feel that efforts should be dedicated to addressing the following points in revision:

1- the reviewers did not find that the relationship between TORC2, NDRG1, and mitochondrial fission was sufficiently clear and strongly supported by the data. We feel that addressing their points will strengthen these conclusions and clarify the relationship between mTORC2 and NDR1, which may depend on other kinases:

Rev#1 points #10, 12, 15

Rev#2 point #3

Rev#3 major point #3 and minor point #11

Rev#4 points #1-2

2- It will be important to include discussion to address many of the reviewers' points and to enhance the scholarly discussion of past work, including via better integration of your significant body of work with it. Please do not worry about the reference count limit as a rich discussion of the results is very important. Please further discuss as suggested by Rev#1 (points #2, 17, 19, 23, 24, 25, 29) and to address Rev#2's interesting questions in points #5, 6, 8.

3- the mechanism of fission via Cdc42 and independently of Drp1 is called into question by the reviewers and we agree that further analyses are needed to support the conclusions:

Rev#1 points #7, 9, 14, 18, 20, 21, 30, 32, 33, 35

Rev#2 point #7

Rev#3 major point #4; minor point #9 and minor point #5 (about MTFP1)

Rev#4 (main paragraph, asking for analyses in KO cells)

4- Reviewers #2-3 ask for confirmation of the specific activity changes in mTORC2. Rev#2 points #1, 2, 4 and Rev#3 #1-2 should be addressed to bolster this central conclusion.

5- All other referee concerns pertaining to strengthening existing data, providing controls, methodological details, clarifications and textual changes, should also be addressed.

6- Finally, please pay close attention to our guidelines on statistical and methodological reporting (listed below) as failure to do so may delay the reconsideration of the revised manuscript. In particular, please provide:

- a Supplementary Figure including unprocessed images of all gels/blots in the form of a multi-page pdf file. Please ensure that blots/gels are labeled and the sections presented in the figures are clearly indicated.
- a Supplementary Table including all numerical source data in Excel format, with data for different figures provided as different sheets within a single Excel file. The file should include source data giving rise to graphical representations and statistical descriptions in the paper and for all instances where the figures present representative experiments of multiple independent repeats, the source data of all repeats should be provided.

We would be happy to consider a revised manuscript that would satisfactorily address these points, unless a similar paper is published elsewhere, or is accepted for publication in Nature Cell Biology in the meantime.

In contrast, although we agree with Rev#3 that studies of mitophagy would be very interesting, that what happens in adipose depots is also of interest, and that these analyses would provide valuable insights, we consider these points [i.e., Rev#3's points "6. Mitochondrial elongation protects mitochondria from mitophagy upon starvation (PMCID: PMC3088644). Thus, mitophagy activity should be evaluated in the RictorKO livers that display reduced mitochondrial fragmentation."; and Rev#3's question "Is mTOR activated in the adipose tissue after challenging mice on the respective diets for longer time periods?"] to be beyond the scope of the present study. Thus, addressing these points experimentally will not be necessary for reconsideration of the manuscript at this journal.

- ensure that it conforms to our format instructions and publication policies (see below and www.nature.com/nature/authors/).
- provide a point-by-point rebuttal to the full referee reports verbatim, as provided at the end of this letter.
- provide the completed Editorial Policy Checklist (found here <https://www.nature.com/authors/policies/Policy.pdf>), and Reporting Summary (found here <https://www.nature.com/authors/policies/ReportingSummary.pdf>). This is essential for reconsideration of the manuscript and these documents will be available to editors and referees in the event of peer review. For more information see <http://www.nature.com/authors/policies/availability.html> or contact me.

Nature Cell Biology is committed to improving transparency in authorship. As part of our efforts in this direction, we are now requesting that all authors identified as 'corresponding author' on published papers create and link their Open Researcher and Contributor Identifier (ORCID) with their account on the Manuscript Tracking System (MTS), prior to acceptance. ORCID helps the scientific community achieve unambiguous attribution of all scholarly contributions. You can create and link your ORCID from the home page of the MTS by clicking on 'Modify my Springer Nature account'. For more information please visit www.springernature.com/orcid.

[Redacted]

We hope that you will find our referees' comments and editorial guidance helpful. Please do not hesitate to contact me if there is anything you would like to discuss. Thank you again for considering NCB for your work and please accept my sincere apologies for the delay.

Best wishes,

Melina

Melina Casadio, PhD
Senior Editor, Nature Cell Biology
ORCID ID: <https://orcid.org/0000-0003-2389-2243>

Reviewers' Comments:

Reviewer #1:

Remarks to the Author:

This manuscript presents a phosphoproteomic study to identify mechanisms underpinning metabolic adaptation under conditions of fasting and a return to fatty acid availability. The authors report a subset of phosphosites that are modulated by diet and determine the likely kinases that phosphorylate these sites. They suggest that the data points to mTORC1 or 2 activation. Gene KOs were employed to verify that mTORC2 is in fact reactivated during fasting. They go on to show mTORC2 supports mitochondrial dynamics and respiration. Loss of mTORC2 activity leads to loss of mitochondrial fission, independent of phosphorylation of Drp1 (the GTPase known to drive fission). A second phosphoproteomic study followed by GO analysis shows a cluster of potential mTORC2 targets involved in the regulation of metabolism. The authors chose to focus on one potential target, NDRG1, and come to the conclusion that mTORC2 stimulates mitochondrial fission by phosphorylating NDRG1 at Ser336. The authors go on to determine, which GTPase is involved in this process and find that NDRG1 associates with Cdc42 as well as a RhoGEF and GAP. Cdc42 appears to be involved in mitochondrial fission. Proteomics is used to identify Cdc42 interacting proteins and the authors note five effectors/regulators, some of which are suggested to affect mitochondrial fission. The authors suggest that fission is regulated by recruitment of Cdc42 and its interacting regulators and effectors to MAMs (mitochondrial associated membranes), where they reorganize the actin cytoskeleton into ring-like structures that promote fission.

This work addresses an important area of research, where we lack molecular details of the mechanism regulating mitochondrial dynamics in response to feeding/fasting. A Drp1-independent mechanism driving mitochondrial fission under these conditions may well have a role in response to other factors and would be of interest to a wide readership. This referee is not aware of similar work that has been published.

There are however several issues that need to be addressed before this paper could be recommended for publication. If these issues are addressed I think this work will be of interest to the broad readership of Nature Cell Biology.

General Comments

It was disappointing that this study, like many unbiased screens, actually ends up cherry picking proteins that they want to work on rather than studying what the data throw up. It is however unlikely that the authors can go back and remain true to their initial unbiased approach at this point.

The manuscript is particularly impenetrable in the first sections. In fact, it feels as if there is a change of authorship at the top of page 7, where the style becomes easier to understand, with improved logic links explaining the workflow. I suggest that the first part of the manuscript needs revising so it is easier to follow, like the remainder of the manuscript.

This manuscript contains vast amounts of data and is a very dense read, in fact it is quite difficult to follow without reading more than once. Much of the supplementary data is actually vital for the story presented and I spent more time looking at supplementary figures than the main figures. Some of the data are bona fide supplementary data e.g. Ext. Fig. 7 but others are not. You cannot, in fact, evaluate the hypotheses put forward in this work without many of the supplementary data sets. Due to this, it actually feels rather random whether data is presented in main figures and which in supplementary. The data however seem sound in general and are extensive (thanks to the amount in supplementary). Guidance from NCB suggests that the number of Supplementary Figures should not exceed the number of main figures in the paper. In this case there are four main figures (with 37 panels in total) but there are 15 Ext figures with a total of 116 panels. The figures and panels are also often not cited in the order they are labelled, which makes the while story even harder to follow, e.g. Figure 4 panels are all cited out of order with 4d appearing first. It is standard convention, that figures are labelled in the order they are cited in the manuscript. Overall presentation of the data in this manuscript needs to be rethought. The amount of data has to be reduced and presented in a more easily accessible and understandable manner

Considerably more references are required in parts of the script. Many statements sound unusual and require enforcing and backing up with the relevant primary reference(s).

On a small issue, the extensive use of hyphens in the script is unusual. These should be removed and replaced with the correct grammar. In many places, small changes in vocabulary and grammar would clarify the manuscript.

I appreciated the method schematics included in the figures.

Main Specific Comments

1. The labelling and description in the Figure 1d and its legend is not clear. What do the colours/sizes of the dots indicate? In fact, Ext Figure 1c would be better than Fig. 1d in the main text, especially if more labels were added e.g. for MAPKs. What is the second unlabelled orange dot in Ext Fig 1c?
2. 'Interestingly, modest enrichment of RPS6KA2/B2, AKT1-3, PRKCB/D, and SGK1-3 with corn oil, and SGK1-3 in BODIPY C16 group, pointed to mTORC1 or mTORC2 activation (Fig. 1f, g).' – please add a reference showing these enriched hits are mTORC substrates. Also, it is not obvious that SGK1-3 are enriched in Fig. 1f – please clarify.
3. 'Refeeding reduced the overall kinase network, but expectedly activated nutrient-sensitive kinases, e.g., mTOR, and suppressed those induced by fasting, e.g., CDK2 (Fig. 1e, Extended Data Fig. 1d).' CDK2 is obviously suppressed in Ext Fig 1d but MTOR is not visible – please clarify. Fig.1e is also cited – what does this show regarding refed samples?

4. 'Given our interest in the mTOR-autophagy axis, we confirmed that corn oil exposure robustly increased levels of P-P70T389, P-S6S235/236, and P-AKTS473, and Raptor and Rictor—broadly reflecting mTORC1/C2 activation; without affecting the energy sensor7, P-AMPKT172 (Extended Data Fig. 1e, f)'. Raptor levels are unchanged (statistically) in corn fed samples – please clarify.

5. 'Refeeding or exposure to corn oil each increased P-P70T389 and P-AKTS473 to similar levels suggesting that availability of lipids originating from breakdown of dietary triglycerides leads to mTORC1/C2 activation (Extended Data Fig. 1e, f).' Please clarify this statement. Although P-P70T389 levels in refeeding and corn-fed samples differ similarly statistically to fasted, mean levels are not similar (1-5-8). Although P-AKTS473 mean levels are similar in both samples, the refed sample does not appear to be statistically different to fasted.

6. Ext Figure 7A is not cited in the text. Add to sentence 'RictorKO livers exhibited mild changes in lipid composition of mitochondrial associated membranes (MAMs)^{13,14} (as determined via unbiased lipidomics) (Extended Figure 7a)?'

7. 'Consistent with this idea, MAMs were enriched in Rictor, and in proteins regulating fission¹⁵ (mitochondrial fission factor (MFF) and Drp1) and fusion (Opa1 and mitofusins) (Fig. 2f). Strikingly, MAMs from RictorKO livers showed marked reduction in MFF levels without affecting levels of P-Drp1Ser616, Drp1, mitofusins or Opa1 (Fig. 2f)—supporting the idea that loss of mTORC2 blocks mitochondrial fission.'

Please clarify enriched – compared to what? Drp1 does not appear to be enriched in MAMs (it's higher in all other samples except total mitochondria). Opa1 is higher in total the mitochondria sample. P-Drp1Ser616 is extremely difficult to see but Drp1 levels do seem to decrease in Rictor KO cells. Although Opa1 levels appear stable in MAMs in Rictor KOs, interestingly the Opa1 bands in MAMs are upshifted compared to total mitochondria samples. Could this suggest phosphorylation of Opa1 in MAMs? Please comment.

8. Videos 3-5 and in fact the videos in general add little to the manuscript.

9. 'Since fusion rates were identical in siCon and siRictor cells, the observed mitochondrial phenotype in Rictor depleted cells is due to impaired fission and not excessive fusion.' The data in Fig. 2g suggests that fusion rates are not statistically different in control and siRictor samples. However visual inspection suggest differences are similar to that between the two samples for fission but with fusion data for the siRictor sample skewed by an outlier. Please comment.

10. 'We also focused on NDRG1 (Fig. 3e), a protein involved in lipid metabolism²⁰ and extensively phosphorylated on its C terminus by mTORC2/SGK121.' Please clarify the choice of NDRG1 for further studies in terms of the overall data set in Figure 3c. Please label the NDRG1 dot.

11. 'we sought to confirm if NDRG1S336 is indeed an mTORC2 target'. This nomenclature seems confusing; it would be simpler to say wt NDRG1.

12. '(Extended Data Fig. 8h-j) showed that this phosphopeptide is markedly less enriched in siRictor cells compared to siCon cells—confirming that mTORC2 phosphorylates NDRG1S336 in vitro and in vivo in MAMs. Unfortunately, this and much other data presented in this manuscript suggests that mTORC2 regulates NRG1 phosphorylation, not that NRG1 is a direct substrate for mTORC2. Kinase cascades are so prevalent they cannot be ruled out and only work with purified proteins will prove a direct relationship. Please edit wording here and throughout the manuscript to reflect this.

13. 'we expressed FLAG-tagged NDRG1 WT or NDRG1S336A in livers silenced for endogenous NdrG1

(Extended Data Fig. 12c) and confirmed equivalent FLAG expression by immunohistochemistry (Extended Data Fig. 12d). Data in Ext Fig 12d suggest that NDRG1S336A expresses at higher level than wt.

14. 'Since NDRG1 does not exhibit intrinsic GTPase activity, a critical requirement for membrane scission, we asked whether P-NDRG1S336 engages with proteins with intrinsic GTPase activity to facilitate fission.' Please add a reference showing GTPase activity is required for membrane scission. NDRG1 is known to be an effector protein for Rab4a. The involvement of Rab4a in this process should be investigated (see also point 17).

15. 'we used proteomics to find proteins bound to FLAG NDRG1 WT but not NDRG1S336A.' Although wt NDRG1 will have the capability of being phosphorylated, do the authors know that it is actually phosphorylated under the experimental conditions used. This would be important to verify to validate the binding results.

16. Data shown in Ext Fig. 13a should move to the main figures.

17. 'Indeed, when compared to NDRG1 WT, NDRG1S336A displayed modestly reduced binding to Cdc42, Arhgef10 (a Rho GEF that activates Rho GTPases by stimulating GDP/GTP exchange), and Arhgap35 (Rho GAP that facilitates GTP hydrolysis to inactivate Rho GTPases)'

Is it known whether ArhGEF10 and ArhGAP35 are actually active GEF/GAPs towards Cdc42 in cells. Please add references in support of statement. It would be useful to check for Rab associated proteins as well.

18. 'Furthermore, FLAG-NDRG1 WT interacts with mCherry-Cdc42 WT, but displays reduced interaction with mutant Cdc42T17N, which does not bind GTP (Extended Data Fig. 13c) indicating that NDRG1S336 phosphorylation and Cdc42-GTP binding are required for their interaction to promote fission.'

A reduced interaction with Cdc42 T17N is extremely difficult to see (even though quantification confirms apparent loss of binding). No details of how quantification was achieved are available. These details should be available for all data presented. How have the data been normalized?

A better experiment would be to use a GFP-effector construct to probe for active Cdc42 at MAMs. Currently the statement here should be modified to say the data suggest active GTP bound Cdc42 is required for fission, or that the data warrants further investigation.

19. 'Since Cdc42 is largely controlled through translocation'. This is at best only a partially true statement. Like all GTPases Cdc42 requires to be membrane bound and to be GTP loaded to be active. This statement should be qualified and references added.

20. 'Indeed, Cdc42 and RhoA26 (Fig 4b), but not dynamins, were enriched in MAMs from fasted control livers.' Please add in statistical analysis for RhoA into Fig. 4b.

21. 'By contrast, MAMs from fasted RictorKO (Fig 4b) and NDRG1S336A livers (Fig 4c) each showed markedly reduced Cdc42 levels without affecting levels of RhoA'. Please add RhoA statistical analysis to Fig. 4b as some visual reduction appears to occur in RictorKOs. There are no data for Rho A in Fig. 4c, please amend statement.

22. The authors now shift to analysis in NIH3T3s, using exogenously expressed GFP-Cdc42. Why, as you would really need to look in liver extracts as had been used throughout the study? Surely looking in the literature would be just as useful as Cdc42 pulldowns in NIH3T3s?

23. 'Of note, 5 enriched targets specifically belonged to the Cdc42 family of effectors and regulators: Cdc42 effector proteins (Cdc42ep4, Cdc42ep1, and Cdc42ep2), the Rho GDP Dissociation Inhibitor alpha (Arhgdia), and IQ motif-containing GTPase-activating protein 1 (Iqgap1), which is a downstream Cdc42 effector and an upstream Cdc42 scaffold protein (Fig 4e, Extended Data Fig 14b).'

Cdc42eps have several names, which should be noted here to aid readers understand the significance of these results: CEP4/BORG4, BORG5/MSE55/CEP1, BORG21/CEP2. Please explain how IQGAP1 is both an effector and a scaffold protein: add references to support both assertions. Arhgdia (and others) are more usually referred to as RhoGDI1 etc.

24. 'the atypical Rho GTPase/Rho GDP-dissociation inhibitor 1 (Rhotbt1)'. Presumably this is a typo – Rhotbt1 (as in Fig 4e). Please add reference to show RhoBTB1 is a RhoGDI.

25. 'In addition, we included two identified binding partners of NDRG1, Arhgef10 and Arhgap35, which serve as GEF and GAP, respectively (Extended Data Fig 13a).'

As previous – please cite references showing Arhgef10 and Arhgap35 are active against Cdc42.

26. Further characterization of the involvement of Cdc42 interactors in mitochondrial fission is undertaken in NIH3T3 cells. Please justify the use of these cells and the relevance of these results to the overall story presented.

27. 'In addition, silencing the Rho GDP dissociation inhibitor Arhgdia (but not Arhgdib), which releases Cdc42 from inhibition by Arhgdia, increased mitochondrial number'. Please clarify this statement.

28. 'Interestingly, not all Rho GTPases impact mitochondrial dynamics,' Please clarify this statement: why would the authors/readers think all GTPases would impact on one particular process?

29. 'likely because Iqgap1 serves as both downstream Cdc42 effector and upstream Cdc42 scaffold protein'. Clarify this statement with a reference.

30. 'Since Arhgap35 inactivates GTPases, we anticipated that depleting Arhgap35 would stimulate Cdc42, leading to fission; however, knocking down Arhgap35 decreased mitochondrial number—reflecting fission failure (Fig 4f, Extended Data Fig 14d-g). This likely reflects the complex regulation of Cdc42 requiring subsequent inactivation to complete its function'. Alternatively, Arhgap35 is not active towards Cdc42 and affects, instead, an antagonistic GTPase.

31. 'as well as specificity amongst the different effectors and regulators in stimulating fission during fasting.' All this analysis is performed in NIH3T3s, so has little to do with fasting in the liver.

32. 'Consistent with these findings, in addition to Cdc42, we also observed substantial enrichment in MAMs of Cdc42 effector, Cdc42ep1, and those of regulator proteins, Arhgap35 and Arhgdia (Extended Data Fig 15a, b).'

Statistical analysis presented shows no enrichment of the proteins suggested – please clarify.

33. 'Indeed, in control cells, actin assembled around mitochondrial subpopulations to generate ring-like structures and promote fission'. Although the data do suggest that ring-like actin structures are generated, the link to fission although possible or even likely is not proven. Please qualify statement.

34. 'We demonstrate that NDRG1 associates with mitochondrial constrictions to facilitate fission (Fig 4h)'. Please qualify – the data suggest this and a potential model is shown in Fig 4h.

35. The data presented do suggest a role for Cdc42 in mTORC2-NDRG1 driven regulation of

mitochondrial fission. However, the conclusions based on the final sections of the manuscript are less robust than other parts of the analysis.

Minor Comments

1. 'phosphorylation of the mitophagy adapter BNIP3 at Ser(S)79 and S88' Why revert to explaining the three letter code halfway through the manuscript? Move this to first use of S abbreviation for Serine or use three-letter code throughout (which is accessible to more readers).
2. 'SRTASGSSVTS(ph)LEGTRSR' – p is the more usual abbreviation to indicate phosphorylation in a sequence.
3. 'Several S/T residues on the NDRG1 C-terminus are phosphorylated by SGK1.' Start new paragraph here for clarity

Reviewer #2:

Remarks to the Author:

This is an extremely interesting article by Martinez-Lopez and colleagues that links mTORC2 to mitochondrial fission and shows that fasting regulates this process via mTORC2 and NDRG1. It should be noted that the processes downstream of mTORC2 that are Akt-independent have remained mysterious for some time, and thus this manuscript addresses a major gap in knowledge. That said, there are some real questions outstanding here that should be addressed prior to publication.

1. In my own personal experience, liver mTORC2 activity reaches a nadir after about 22 hours of fasting. The authors should show that data showing that mTORC2 activity reaches a minimum at 16 hours.
2. The experiment showing mTORC2 activation by corn oil gavage does not have a gavage control - the stress of gavage could be leading to glucose and insulin release and mTORC2 stimulation. This should be examined. While the authors examined and dismissed a contribution of insulin, they did not examine IGF-1 or leptin, which could regulate mTORC2 via PI3K.
3. The effect of mTORC2 activity on NDRG1 should be mediated by changes in SGK, yet no evidence of SGK phosphorylation/activation is shown.
4. Regulation of Rictor by SIRT1 in the liver has previously been shown (PMID: 21965330), and could explain the mTORC2 activation in response to feeding - this should be examined.
5. It would be interesting to compare and contrast the data gathered by the authors here with the phosphopeptide data found in fasted and refed RKO mice by others (PMID: 24072782). Among other issues, this previous study identified Ndr2 phosphosites as some of the most heavily downregulated, and the overlap between Ndr1/2/3 function is not clear.
6. A number of studies have shown that Rictor deletion shortens worm and mouse lifespan for unknown reasons - it seems logical to ask if any of these effects might be due to diminished mitochondrial fission.
7. p38 MAPK has been identified as a signaling node downstream of liver mTORC2 and possibly SGK (PMID: 24072782) and more recently DRP1 has been identified as a substrate of p38 MAPK (PMID:

31881179). Could mTORC2-p38 MAPK-DRP1 signaling be contributing to the observed phenotypes?

8. Could the authors please discuss why fasting should activate mTORC2, which would seem to be counterproductive?

Reviewer #3:

Remarks to the Author:

The study by Martinez-Lopez et al. aims to establish a link between fasting in liver and mitochondrial fission via mTORC2, via an initial unbiased phosphoproteomics screen in mice that are fed, fasting or challenged with varying diets. Fasting for prolonged periods led to a re-activation of mTORC2, and mitochondrial fission that was dependent upon the TORC1 component Rictor.

The authors show that mTORC2 phosphorylates N-myc downstream regulated gene 1 (NDRG1), an established stress-response protein. In turn, NDRG1 associates with mitochondrial constriction sites and recruits the Rho GTPase Cdc42 to promote mitochondrial fission. Overall, this manuscript suggests the existence of an mTORC2-NDRG1-Cdc42 axis to support fission during fasting, and will be of interest to the Nat Cell Bio community.

Some comments:

Major Concerns:

1. mTORC1 activity is impaired by nutrient or energy deficiency. However, here the authors suggest that mTORC2 specifically is reactivated upon fasting or short refeeding with exogenous fatty acids. The mechanism behind this TORC2 activation is unclear and requires further evaluation in the manuscript – is it due to free mTOR from turned over mTORC1?. Further, AMPK targets within the signaling cascade, beyond P-AMPK, should be measured.
2. The TORC2 effects (raptor knockouts) on fasting induced fission are striking. To ensure that this is TORC2 inhibition specific and not via changes to mTORC1 (as they did in TG studies) fasting effects on mitochondria in TSC deletions shd be examined.
3. Authors claim that “mTORC2 stimulates mitochondrial fission by phosphorylating NDRG1 at S336”. However, there is no proof that mTORC2 directly interacts and phosphorylates NDRG1 or whether it is a secondary kinase that phosphorylates NDRG1.
4. Drp1 has multiple post-translational modifications that dictate its GTPase function and fission activity. However, only Drp1 P-S616 is analyzed in this manuscript (i.e., Extended Data Fig. 6g). The Drp1 S637 site has been implicated in mitochondrial hyperfusion upon nutrient deprivation (PMCID: PMC3088644, PMCID: PMC3121813). Therefore, Drp1 P-S637 levels need to be measured. Then, analyze whether mTORC2-NDRG1-Cdc42 co-localize with Drp1 by imaging experiments.
5. In this manuscript, mice are fasted for 14-16h and then refeed the respective fatty acid/ diet treatments for 30 min. Refeeding protocols after fasting last between 1-6h. This short timing might be the reason why mTOR is not activated in the adipose tissue. Is mTOR activated in the adipose tissue after challenging mice on the respective diets for longer time periods?

Minor Concerns:

5. Previous literature demonstrated that mTORC1 stimulates mitochondrial fission via the mitochondrial fission factor MTFP1 (PMID: 28918902). What is MTFP1 expression and activity levels in the KO mice?
6. Mitochondrial elongation protects mitochondria from mitophagy upon starvation (PMCID: PMC3088644). Thus, mitophagy activity should be evaluated in the RictorKO livers that display reduced mitochondrial fragmentation.
7. Quantification of mitochondrial morphology from EM images should not only be limited to 2D parameters, but also include measurement of mitochondrial volume to assess mitochondrial ultrastructure. Further, confocal images should include circularity as an evaluator of mitochondrial fragmentation.
8. Improved pictures of BODIPY uptake in the liver should be incorporated, where the hepatocyte structure is differentiated.
9. Authors suggest Cdc42 is recruited to MAMs to promote mitochondrial fission. However, there is no clear information as to where Cdc42 resides on basal state (i.e, cytosol, or another organelle).
10. In line with the previous concern, the role of the ER is never assessed. Authors should evaluate ER localization in the mTORC2-NDRG1-Cdc42 complex and analyze whether the number of MAMs change in RictorKO livers.
11. In Extended Data Fig.11 (d), the NDRG1-mediated mitochondrial fission is not visualized clearly. Thus, authors should provide images where the mitochondrial tubule is distinctly fragmented by NDRG1.

Reviewer #4:

Remarks to the Author:

Using an unbiased phosphoproteomic approach in liver of mice that were fed, fasted, fasted and refed, or fed with specific lipids, the authors reveal mTORC2 activation upon fasting. mTORC2 activation is accompanied by increased mitochondrial fission. The comparison of the phosphoproteome of wildtype and rictor KO livers led to the identification of NDRG1, a known mTORC2 target, which is detected at ER-mito-contact sites and whose phosphorylation (specifically on S336) affected mitochondrial OXPHOS functions and fission. Unexpectedly, NDRG1-induced fission was not affected by downregulation of Dnm1l. Rather, NDRG1 interaction studies identify Cdc42, which regulates the actin cytoskeleton, as binding partner of NDRG1. Further experiments show that Cdc42 accumulates at ER-Mito-contact sites in a NDRG1-dependent manner, that some interaction partners of Cdc42 are present at these sites as well, and that Cdc42 as NDRG1 or Rictor is required for mitochondrial fission during fasting.

This is a comprehensive analysis of the effect of fasting on mitochondrial dynamics in liver. The authors provide compelling evidence for the localization of NDRG1 at fission sites as well as the importance of its site-specific phosphorylation and regulation by mTORC2 under fasting. These findings are interesting and shed new light as to how metabolic cues affect mitochondrial dynamics. However, I am less convinced about the proposed mechanisms of mitochondrial fission. The authors examine the role of the mitochondrial fission machinery by downregulating Dnm1l and failed to inhibit NDRG-driven fission. I did not find information about the efficiency of downregulation in these experiments and, in any case, it is difficult to exclude a role of Dnm1l solely based on these experiments. Other mitochondrial membrane processes were originally considered Dnm1l-independent

based on knock-down experiments, but were later shown to be mediated by Dnm1l using knockout cells. Accordingly, testing the hypothesis in Dnm1l knockout cells and examining the role of Dnm1l adaptors such as MFF appears obligatory. On the other hand, the effect of Cdc42 remains largely speculative. The authors provide convincing evidence for NDRG1 binding to Cdc42, the presence of Cdc42 (and of its regulators) at contact sites, but how Cdc42 affects fission remains speculative. I agree with the statement of the authors at the end of the manuscript that the effect of Cdc42 on mitochondrial fission is likely indirect reflecting its known effects on the actin cytoskeleton, which is known to be required for Dnm1l-mediated mitochondrial fission. Accordingly, mTORC2 activation upon fasting may affect mitochondrial fission regulating the actin cytoskeleton in a NDRG1 dependent manner.

Additional points:

1. The authors provide only correlative evidence that mTORC2 drives mitochondrial fission by phosphorylating NDRG1. To demonstrate the dependence of mTORC2-driven mitochondrial fragmentation on NDRG1-S336 phosphorylation, the authors should examine mitochondrial fission in delta-ricor cells expressing NDRG1 and its variant.
2. Although affected by NDRG1 phosphorylation at S336, the binding efficiency of NDRG1 for Cdc42 is very low. Whether NDRG1 exerts its effect on mitochondrial fission via Cdc42 at mitochondria is not unambiguously demonstrated. It therefore appears conceivable that the effects of Cdc42 depletion reflect its general role for the maintenance of the actin cytoskeleton.
3. As a minor point, how do the authors explain mitochondrial marker proteins (such as Mfn1 or MFF) in the ER fraction while ER proteins (such as calreticulin) are missing. This fraction appears to represent a general membrane fraction.

AUTHOR AFFILIATIONS – should be denoted with numerical superscripts (not symbols) preceding the

names. Full addresses should be included, with US states in full and providing zip/post codes. The corresponding author is denoted by: "Correspondence should be addressed to [initials]."

Methods should be written concisely, but should contain all elements necessary to allow interpretation and replication of the results. As a guideline, Methods sections typically do not exceed 3,000 words. The Methods should be divided into subsections listing reagents and techniques. When citing previous methods, accurate references should be provided and any alterations should be noted. Information must be provided about: antibody dilutions, company names, catalogue numbers and clone numbers for monoclonal antibodies; sequences of RNAi and cDNA probes/primers or company names and catalogue numbers if reagents are commercial; cell line names, sources and information on cell line identity and authentication. Animal studies and experiments involving human subjects must be reported in detail, identifying the committees approving the protocols. For studies involving human subjects/samples, a statement must be included confirming that informed consent was obtained. Statistical analyses and information on the reproducibility of experimental results should be provided in a section titled "Statistics and Reproducibility".

All Nature Cell Biology manuscripts submitted on or after March 21 2016 must include a Data availability statement at the end of the Methods section. For Springer Nature policies on data availability see <http://www.nature.com/authors/policies/availability.html>; for more information on this particular policy see <http://www.nature.com/authors/policies/data/data-availability-statements-data-citations.pdf>. The Data availability statement should include:

- Accession codes for primary datasets (generated during the study under consideration and designated as "primary accessions") and secondary datasets (published datasets reanalysed during the study under consideration, designated as "referenced accessions"). For primary accessions data should be made public to coincide with publication of the manuscript. A list of data types for which submission to community-endorsed public repositories is mandated (including sequence, structure, microarray, deep sequencing data) can be found here <http://www.nature.com/authors/policies/availability.html#data>.
- Unique identifiers (accession codes, DOIs or other unique persistent identifier) and hyperlinks for datasets deposited in an approved repository, but for which data deposition is not mandated (see here for details <http://www.nature.com/sdata/data-policies/repositories>).
- At a minimum, please include a statement confirming that all relevant data are available from the authors, and/or are included with the manuscript (e.g. as source data or supplementary information), listing which data are included (e.g. by figure panels and data types) and mentioning any restrictions on availability.
- If a dataset has a Digital Object Identifier (DOI) as its unique identifier, we strongly encourage including this in the Reference list and citing the dataset in the Methods.

We recommend that you upload the step-by-step protocols used in this manuscript to the Protocol Exchange. More details can found at www.nature.com/protocolexchange/about.

All imaging data should be accompanied by scale bars, which should be defined in the legend. Cropped images of gels/blots are acceptable, but need to be accompanied by size markers, and to retain visible background signal within the linear range (i.e. should not be saturated). The boundaries of panels with low background have to be demarked with black lines. Splicing of panels should only be considered if unavoidable, and must be clearly marked on the figure, and noted in the legend with a statement on whether the samples were obtained and processed simultaneously. Quantitative comparisons between samples on different gels/blots are discouraged; if this is unavoidable, it should only be performed for samples derived from the same experiment with gels/blots were processed in parallel, which needs to be stated in the legend.

Figures should be provided at approximately the size that they are to be printed at (single column is 86 mm, double column is 170 mm) and should not exceed an A4 page (8.5 x 11"). Reduction to the scale that will be used on the page is not necessary, but multi-panel figures should be sized so that

the whole figure can be reduced by the same amount at the smallest size at which essential details in each panel are visible. In the interest of our colour-blind readers we ask that you avoid using red and green for contrast in figures. Replacing red with magenta and green with turquoise are two possible colour-safe alternatives. Lines with widths of less than 1 point should be avoided. Sans serif typefaces, such as Helvetica (preferred) or Arial should be used. All text that forms part of a figure should be rewritable and removable.

SUPPLEMENTARY INFORMATION – Supplementary information is material directly relevant to the conclusion of a paper, but which cannot be included in the printed version in order to keep the manuscript concise and accessible to the general reader. Supplementary information is an integral part of a Nature Cell Biology publication and should be prepared and presented with as much care as

the main display item, but it must not include non-essential data or text, which may be removed at the editor's discretion. All supplementary material is fully peer-reviewed and published online as part of the HTML version of the manuscript. Supplementary Figures and Supplementary Notes are appended at the end of the main PDF of the published manuscript.

The total number of Supplementary Figures (not including the "unprocessed scans" Supplementary Figure) should not exceed the number of main display items (figures and/or tables (see our Guide to Authors and March 2012 editorial <http://www.nature.com/ncb/authors/submit/index.html#suppinfo>; <http://www.nature.com/ncb/journal/v14/n3/index.html#ed>). No restrictions apply to Supplementary Tables or Videos, but we advise authors to be selective in including supplemental data.

GUIDELINES FOR EXPERIMENTAL AND STATISTICAL REPORTING

REPORTING REQUIREMENTS – To improve the quality of methods and statistics reporting in our papers we have recently revised the reporting checklist we introduced in 2013. We are now asking all life sciences authors to complete two items: an Editorial Policy Checklist (found here <https://www.nature.com/authors/policies/Policy.pdf>) that verifies compliance with all required editorial policies and a reporting summary (found here <https://www.nature.com/authors/policies/ReportingSummary.pdf>) that collects information on experimental design and reagents. These documents are available to referees to aid the evaluation of the manuscript. Please note that these forms are dynamic 'smart pdfs' and must therefore be downloaded and completed in Adobe Reader. We will then flatten them for ease of use by the reviewers. If you would like to reference the guidance text as you complete the template, please access these flattened versions at <http://www.nature.com/authors/policies/availability.html>.

STATISTICS – Wherever statistics have been derived the legend needs to provide the n number (i.e. the sample size used to derive statistics) as a precise value (not a range), and define what this value represents. Error bars need to be defined in the legends (e.g. SD, SEM) together with a measure of centre (e.g. mean, median). Box plots need to be defined in terms of minima, maxima, centre, and percentiles. Ranges are more appropriate than standard errors for small data sets. Wherever statistical significance has been derived, precise p values need to be provided and the statistical test used needs to be stated in the legend. Statistics such as error bars must not be derived from $n < 3$. For

sample sizes of $n < 5$ please plot the individual data points rather than providing bar graphs. Deriving statistics from technical replicate samples, rather than biological replicates is strongly discouraged. Wherever statistical significance has been derived, precise p values need to be provided and the statistical test stated in the legend.

Author Rebuttal to Initial comments

Reviewer 1 comments

Itemized points in the Editor's summary of the concerns raised.

-  Questions on relationship between TORC2, NDRG1, and mitochondrial fission
-  Concerns regarding enhancing scholarly discussions and including references
-  Questions on mechanism of fission via Cdc42 and independency from Drp1
-  Questions regarding confirmation of the specific activity changes in mTORC2

Referee #1 (Remarks to the Author):

This manuscript presents a phosphoproteomic study to identify mechanisms underpinning metabolic adaptation under conditions of fasting and a return to fatty acid availability. The authors report a subset of phosphosites that are modulated by diet and determine the likely kinases that phosphorylate these sites. They suggest that the data points to mTORC1 or 2 activation. Gene KO's were employed to verify that mTORC2 is in fact reactivated during fasting. They go on to show mTORC2 supports mitochondrial dynamics and respiration. Loss of mTORC2 activity leads to loss of mitochondrial fission, independent of phosphorylation of Drp1 (the GTPase known to drive fission). A second phosphoproteomic study followed by GO analysis shows a cluster of potential mTORC2 targets involved in the regulation of metabolism. The authors chose to focus on one potential target, NDRG1, and come to the conclusion that mTORC2 stimulates mitochondrial fission by phosphorylating NDRG1 at Ser336. The authors go on to determine, which GTPase is involved in this process and find that NDRG1 associates with Cdc42 as well as a RhoGEF and GAP. Cdc42 appears to be involved in mitochondrial fission. Proteomics is used to identify Cdc42 interacting proteins and the authors note five effectors/regulators, some of which are suggested to affect mitochondrial fission. The authors suggest that fission is regulated by recruitment of Cdc42 and its interacting regulators and effectors to MAMs (mitochondrial associated membranes), where they reorganize the actin cytoskeleton into ring-like structures that promote fission.

This work addresses an important area of research, where we lack molecular details of the mechanism regulating mitochondrial dynamics in response to feeding/fasting. A Drp1-independent mechanism driving mitochondrial fission under these conditions may well have a role in response to other factors and would be of interest to a wide readership. This referee is not aware of similar work that has been published. There are however several issues that need to be addressed before this paper could be recommended for publication. If these issues are addressed I think this work will be of interest to the broad readership of Nature Cell Biology.

General Comments

- It was disappointing that this study, like many unbiased screens, actually ends up cherry picking proteins that they want to work on rather than studying what the data throw up. It is however unlikely that the authors can go back and remain true to their initial unbiased approach at this point.

Answer: We do realize that we cherry-picked 2 targets that appeared on the list. However, **NDRG1 is a known and previously described downstream target of mTORC2/SGK1, and its role in relation to mitochondrial metabolism is completely uncharacterized.** We were encouraged by these points, and by the fact that **NDRG1 localizes to lipid droplets (LD) to regulate their size (*Breast Cancer Res.* 2018;20(1):55).** Since the C-terminus of NDRG1 is extensively phosphorylated by the mTORC2 downstream kinase SGK1 (*Biochim Biophys Acta Mol Basis Dis.* 2018;1864(8):2644-2663), we felt that **NDRG1 and its phosphorylation by mTORC2-SGK1 axis could explain for the lipid phenotype (increased triglyceride levels during fasting) in *Rictor*^{KO} livers. The Reviewer may please also note that although NDRG1 only showed trends towards hypophosphorylation in total liver homogenates (Fig. 4e, Extended Data Fig. 6c), phosphoproteomics in *Rictor*^{KO} MAMs from livers of fasted mice showed marked NDRG1^{Ser336} hypophosphorylation when compared to fasted control MAMs (Extended Data Fig. 6d-f). Thus it is likely that we are looking at localized phosphorylation changes that are unlikely to be picked up in analyses of whole tissue homogenates (Please see Q15 on R1 – page 11). Although we cherry-picked NDRG1 we have done our due diligence in extensively characterizing the novel role of NDRG1 phosphorylation in mitochondrial function as it relates to mTORC2 signaling. While it is difficult to go back and initiate a new analysis in this manuscript; we will, in a subsequent study, initiate a mitochondrial dynamics screen to find novel targets modulating mitochondrial dynamics in an unbiased manner.**

- The manuscript is particularly impenetrable in the first sections. In fact, it feels as if there is a change of authorship at the top of page 7, where the style becomes easier to understand, with improved logic links explaining the workflow. I suggest that the first part of the manuscript needs revising so it is easier to follow, like the remainder of the manuscript.

Answer: We agree with the reviewer. We have now expanded on the parts of the manuscript that were concerning to the reviewer. We are confident that the revised manuscript will flow better.

- This manuscript contains vast amounts of data and is a very dense read, in fact it is quite difficult to follow without reading more than once. Much of the supplementary data is actually vital for the story presented and I spent more time looking at supplementary figures than the main figures. Some of the data are bona fide supplementary data e.g. Ext. Fig. 7 but others are not. You cannot, in fact, evaluate the hypotheses put forward in this work without many of the supplementary data sets. Due to this, it actually feels rather random whether data is presented in main figures and which in supplementary. The data however seem sound in general and are extensive (thanks to the amount in supplementary). Guidance from NCB suggests that the number of Supplementary Figures should not exceed the number of main figures in the paper. In this case there are four main figures (with 37 panels in total) but there are 15 Ext figures with a total of 116 panels.

The figures and panels are also often not cited in the order they are labelled, which makes the while story even harder to follow, e.g. Figure 4 panels are all cited out of order with 4d appearing first. It is standard convention, that figures are labelled in the order they are cited in the manuscript. Overall presentation of the data in this manuscript needs to be rethought. The amount of data has to be reduced and presented in a more easily accessible and understandable manner

Answer: We thank the reviewer for this comment and apologize for any inconvenience due to the extensive data presented and how these data were organized in this manuscript. We have now followed the NCB guidelines and the Reviewer's suggestion to reorganize the data panels into 7 Main figures and 10 Extended data figures. In addition, some data not directly relevant to the work have been removed, while several key pieces of Supplementary data are now presented in the Main figures. We have also labelled the figures in order as per standard convention. We are confident that these measures will allow the data to be more accessible and understandable.

- Considerably more references are required in parts of the script. Many statements sound unusual and require enforcing and backing up with the relevant primary reference(s).

Answer: We thank the reviewer for this comment. In line with increasing the number of sentences and textually revising the manuscript, we have now incorporated adequate citations to support the background work that formed the basis for the study and our findings in this paper.

- On a small issue, the extensive use of hyphens in the script is unusual. These should be removed and replaced with the correct grammar. In many places, small changes in vocabulary and grammar would clarify the manuscript.

Answer: We thank the reviewer for this comment. We have replaced extensive hyphens with correct grammar and vocabulary.

- I appreciated the method schematics included in the figures.

Answer: We thank the reviewer for this comment.

Main Specific Comments

1. The labelling and description in the Figure 1d and its legend is not clear. What do the colours/sizes of the dots indicate? In fact, Ext Figure 1c would be better than Fig. 1d in the main text, especially if more labels were added e.g. for MAPKs. What is the second unlabelled orange dot in Ext Fig 1c?

Answer: We thank the reviewer for this comment. As suggested by the Reviewer, **Extended Data Fig. 1c** is now Main **Fig. 1d**, and previous **Fig. 1d** is now **Extended Data Fig. 1c**.

We have identified the second unlabeled orange dot in new Main **Fig. 1d**. The unlabeled dot contains a mixed population of kinases: DCLK2, Obscn, PIM1, SSTK, STK33, TSSK1-3, Trad, Trb1, Trio, and VACAMKL.

Both size and color indicate a custom combined score obtained by multiplying the phosphopeptide count and kinase score from iGPS. This score was created to emphasize the importance of the different kinase groups. We have added this diagram as legend to make it clearer.

2. ‘Interestingly, modest enrichment of RPS6KA2/B2, AKT1-3, PRKCB/D, and SGK1-3 with corn oil, and SGK1-3 in BODIPY C16 group, pointed to mTORC1 or mTORC2 activation (Fig. 1f, g).’ – please add a reference showing these enriched hits are mTORC substrates. Also, it is not obvious that SGK1-3 are enriched in Fig. 1f – please clarify.

Answer: We have now provided extensive citations to support the use of AKT1-3, SGK1-3, PRKCB/D and RPS6KA2/B2 as downstream markers of mTORC1 or C2 activation. These references include: *Int J Mol Sci.* 2020;21(4):1199; *Science.* 2005;307(5712):1098-101; *Sci Signal.* 2021;14(678):eabe4509; and *Cell Signal.* 2009;21(11):1680-5.

Although we failed to observe enrichment of SGK1-3 in corn oil-treated phosphoproteomics samples, immunoblotting confirmed increases in mTORC1 and C2 activation in response to corn oil treatment as indicated by increases in levels of mTORC1 and C2 downstream targets, i.e., P-P70^{Thr389} and P-S6^{Ser235/236} (mTORC1) and P-AKT^{Ser473} (mTORC2).

3. ‘Refeeding reduced the overall kinase network, but expectedly activated nutrient-sensitive kinases, e.g., mTOR, and suppressed those induced by fasting, e.g., CDK2 (Fig. 1e, Extended Data Fig. 1d).’ CDK2 is obviously suppressed in Ext Fig 1d but MTOR is not visible – please clarify. Fig.1e is also cited – what does this show regarding refed samples?

Answer: We apologize for the confusion in relation to the absence of mTOR in **Extended Data Fig. 1d**. We clarify here that when the field refers to mTOR activation, it refers to enrichment of mTOR activation markers, AKT1-3 and SGK2-3 as well RPS6KA2/B2, and not mTOR *per se*. We have modified the text to reflect this correction.

“Refeeding reduced the overall kinase network, but expectedly enriched markers indicating activation of nutrient-sensitive kinases, e.g., AKT1-3 and SGK1-3, and suppressed those induced by fasting, e.g., CDK2 (Extended Data Fig. 1d).”

Fig. 1e was an erroneous inclusion and has been removed.

4. ‘Given our interest in the mTOR-autophagy axis, we confirmed that corn oil exposure robustly increased levels of P-P70T389, P-S6S235/236, and P-AKTS473, and Raptor and Rictor—broadly reflecting mTORC1/C2 activation; without affecting the energy sensor7, P-AMPKT172 (Extended Data Fig. 1e, f). Raptor levels are unchanged (statistically) in corn fed samples – please clarify.’

Answer: We thank the reviewer for this comment. We apologize for this oversight. We have now removed Raptor and Rictor from the statement because indeed corn oil did not increase Raptor and Rictor protein levels in liver.

5. 'Refeeding or exposure to corn oil each increased P-P70T389 and P-AKTS473 to similar levels suggesting that availability of lipids originating from breakdown of dietary triglycerides leads to mTORC1/C2 activation (Extended Data Fig. 1e, f).' Please clarify this statement. Although P-P70T389 levels in refeeding and corn-fed samples differ similarly statistically to fasted, mean levels are not similar (1-5-8). Although P-AKTS473 mean levels are similar in both samples, the refed sample does not appear to be statistically different to fasted.

Answer: We thank the reviewer for this comment. We apologize for the lack of clarity, and we have incorporated a new sentence to make this clearer. The sentence reads as follows:

"Refeeding is well known to activate mTORC1/C2 signaling; however it is unclear if dietary triglycerides per se stimulate mTOR signaling in vivo. Our observation that corn exposure activates mTORC1/C2 signaling to similar levels observed by refeeding, indicates that dietary triglycerides per se can robustly activate mTOR signaling..."

We are assuming that the reviewer is considering 1-5-8 to correspond to the lane numbers. In this case, it would seem that the reviewer is looking at Fasted (1)-BODIPY C₁₆(5)-Refed(8) samples. From our data we can see that the mean levels of BODIPY C₁₆-treated P-P70^{T389} (lanes 5 and 6) are identical to fasted state (lanes 1 and 2), whereas the corn oil (lanes 3 and 4) and refed (lanes 7 and 8)-treated samples exhibit similar significant increases in P-P70^{T389} levels (reflecting mTOR activation) as can be seen in the blots and the quantification (p=0.0008 and p=0.0006 vs fasted state, respectively).

6. Ext Figure 7A is not cited in the text. Add to sentence 'RictorKO livers exhibited mild changes in lipid composition of mitochondrial associated membranes (MAMs)13,14 (as determined via unbiased lipidomics) (Extended Figure 7a)'

Answer: We thank the reviewer for this comment. We apologize for failing to cite this figure panel in the text. We have now corrected this error as suggested by the reviewer. However, that lipidomics data are now shown in **Table 1**.

3

7. 'Consistent with this idea, MAMs were enriched in Rictor, and in proteins regulating fission15 (mitochondrial fission factor (MFF) and Drp1) and fusion (Opa1 and mitofusins) (Fig. 2f). Strikingly, MAMs from RictorKO livers showed marked reduction in MFF levels without affecting levels of P-Drp1Ser616, Drp1, mitofusins or Opa1 (Fig. 2f)—supporting the idea that loss of mTORC2 blocks mitochondrial fission.' Please clarify enriched – compared to what? Drp1 does not appear to be enriched in MAMs (it's higher in all other samples except total mitochondria). Opa1 is higher in total the mitochondria sample. P-Drp1Ser616 is extremely difficult to see but Drp1 levels do seem to decrease in Rictor KO cells. Although Opa1 levels appear stable in MAMs in Rictor KOs, interestingly the Opa1 bands in MAMs are upshifted compared to total mitochondria samples. Could this suggest phosphorylation of Opa1 in MAMs? Please comment.

Answer: We thank the reviewer for these comments. The enrichment in MAMs is in comparison to homogenates. However, to avoid confusion, we have replaced “enriched” with the term “present”. Indeed, as reported earlier (*Proc Natl Acad Sci U S A. 2013;110(31):12526-34*), Rictor is present in MAMs, as are proteins regulating mitochondrial fission and fusion. Strikingly, MAMs from KO livers show marked reduction in MFF levels when compared to control MAMs, without affecting DRP1, P-DRP1^{Ser616}, MFN1/2 and OPA1. This supports the idea that loss for mTORC2 activity results in fission failure.

In addition, OPA1 levels are higher in mitochondrial fractions when compared to homogenates likely because OPA1 resides in inner mitochondrial membranes (*Cell. 2006;126(1):177-89*) to regulate fusion of mitochondrial inner membrane and maintenance of mitochondrial cristae (*Cell. 2006;127(2):383-95*). As the Reviewer also noted, there is an upward shift of OPA1 bands in MAMs compared to mitochondria, which could be due to phosphorylation/additional post-translational modification. However, since this upward shift is not sensitive to mTORC2 loss, we suspect that this modification may not be critical to the pathway we are trying to establish.

In relation to Drp1, the reviewer is correct, we were able to see only small amounts of P-DRP1^{Ser616} and P-DRP1^{Ser637} in MAMs; however, we consistently observed no differences in levels of P-DRP1^{Ser616} and P-DRP1^{Ser637} in MAMs from Con and *Rictor*^{KO} livers. In addition, after careful assessment of quantifications of total Drp1 in MAMs, we find no differences in Drp1 levels in MAMs from Con and *Rictor*^{KO} livers. Furthermore, we found no differences in Drp1 levels in MAMs from livers expressing WT or S366A mutant NDRG1. We do agree with the reviewer that Drp1 levels *in vivo* are lower in *Rictor*^{KO} homogenates when compared to Con. However, **since fission is regulated by the MAM Drp1 pool, the physiological relevance of changes in homogenate Drp1 levels remains unclear.** Further, to this point, in relation to **Reviewer 3's Q4**, Con and *Rictor*^{-/-} MEFs as well as *Rictor*-silenced HepG2 cells (not shown here) display comparable Drp1 levels in total cell lysates despite marked reduction in fission in these cells. Moreover, **exciting new data** generated in this revision clearly indicate that fission driven by the mTORC2-NDRG1 axis is independent of Drp1 but relies on MFF (**please refer to Reviewer 4, main question**). Hence, we are confident that Drp1 does not play a role in fission mediated by NDRG1.

Legend. Immunoblots (IB) and quantification for indicated proteins in homogenates (Hom), pure mitochondria (Mp), mitochondria-associated membranes (MAM), cytosol (Cyt), and endoplasmic reticulum (ER) fractions from 14-16 h fasted Con and *Rictor*^{KO} livers (n=3-8). Two livers were pooled to generate 1 sample. Ponceau is loading control. Individual replicates and means are shown. **P<0.01 Unpaired Student's t-test.

8. Videos 3-5 and in fact the videos in general add little to the manuscript.

Answer: We thank the reviewer for this comment. The time lapse images were presented as panels to demonstrate NDRG1's ability to cut mitochondria, and although corresponding videos *per se* are not as informative as image panels, we felt that we needed to include videos as raw data for the corresponding image panels in the figure.

3

9. 'Since fusion rates were identical in siCon and siRictor cells, the observed mitochondrial phenotype in Rictor depleted cells is due to impaired fission and not excessive fusion.' The data in Fig. 2g suggests that fusion rates are not statistically different in control and siRictor samples. However visual inspection suggest differences are similar to that between the two samples for fission but with fusion data for the siRictor sample skewed by an outlier. Please comment.

Answer: Because *Rictor* loss results in elongated mitochondria that phenocopies *Drp1* loss (Refer to Fig. 3d) unlike how *Mitofusin*-KO mitochondria appear, we sought to reevaluate our live imaging data to determine if this indeed was a true outlier. Inclusion of **new data and recalculations show conclusively that *Rictor* KD leads to fission failure without affecting fusion.** This is also reflected in multiple cell types and in vivo when analyzed for mitochondrial number and size.

Legend. Live cell imaging and quantification for mitochondrial fission and fusion rates in siCon, siRictor or siDnm1l NIH3T3 cells (n=10-12 cells from n=8 independent experiments). Scale = 2 μm. Please refer to **Video 3** (siCon cells), **Video 4** (siRictor cells), and **Video 5** (siDnm1l cells). *P<0.05, One-way ANOVA followed by Tukey's multiple comparisons test.

1

10. ‘We also focused on NDRG1 (Fig. 3e), a protein involved in lipid metabolism²⁰ and extensively phosphorylated on its C terminus by mTORC2/SGK1.’ Please clarify the choice of NDRG1 for further studies in terms of the overall data set in Figure 3c. Please label the NDRG1 dot.

Answer: We have clarified the choice of NDRG1 for our studies in reviewer 1 general comments (page 2 of this document). For the Reviewer’s convenience, we have appended our response on page 2 below:

Answer: We do realize that we cherry-picked 2 targets that appeared on the list. However, **NDRG1 is a known and previously described downstream target of mTORC2/SGK1, and its role in relation to mitochondrial metabolism is completely uncharacterized.** We were encouraged by these points, and by the fact that **NDRG1 localizes to lipid droplets (LD) to regulate their size** (*Breast Cancer Res. 2018;20(1):55*). Since the C-terminus of NDRG1 is extensively phosphorylated by the mTORC2 downstream kinase SGK1 (*Biochim Biophys Acta Mol Basis Dis. 2018;1864(8):2644-2663*), we felt that **NDRG1 and its phosphorylation by mTORC2-SGK1 axis could explain for the lipid phenotype (increased triglyceride levels during fasting) in *Rictor*^{KO} livers.** The Reviewer may please also note that although NDRG1 only showed trends towards hypophosphorylation in total liver homogenates (**Fig. 4e, Extended Data Fig. 6c**), phosphoproteomics in *Rictor*^{KO} MAMs from livers of fasted mice showed marked NDRG1^{Ser336} hypophosphorylation when compared to fasted control MAMs (**Extended Data Fig. 6d-f**). Thus it is likely that we are looking at localized phosphorylation changes that are unlikely to be picked up in analyses of whole tissue homogenates (**Please see Q15 on R1 – page 11**). Although we cherry-picked NDRG1 we have done our due diligence in extensively characterizing the novel role of NDRG1 phosphorylation in mitochondrial function as it relates to mTORC2 signaling. While it is difficult to go back and initiate a new analysis in this manuscript; we will, in a subsequent study, initiate a mitochondrial dynamics screen to find novel targets modulating mitochondrial dynamics in an unbiased manner.

In addition, the NDRG1 dot has been labeled. We also noted that there was a discrepancy between the code and the excel sheet as one was using log2 transformation and the other one was using log10. We have now corrected this error and generated a New **Fig. 4c**.

11. ‘we sought to confirm if NDRG1S336 is indeed an mTORC2 target’. This nomenclature seems confusing; it would be simpler to say wt NDRG1.

Answer: We agree with the reviewer, and we have modified the sentence to make it clearer. The sentence now reads as follows:

“We sought to confirm if the mTORC2/SGK1 cascade phosphorylates NDRG1^{WT} at Serine 336 in its regulation of mitochondrial fission”.

1

12. '(Extended Data Fig. 8h-j) showed that this phosphopeptide is markedly less enriched in siRictor cells compared to siCon cells—confirming that mTORC2 phosphorylates NDRG1S336 in vitro and in vivo in MAMs. Unfortunately, this and much other data presented in this manuscript suggests that mTORC2 regulates NRG1 phosphorylation, not that NRG1 is a direct substrate for mTORC2. Kinase cascades are so prevalent they cannot be ruled out and only work with purified proteins will prove a direct relationship. Please edit wording here and throughout the manuscript to reflect this.

Answer: We agree with the reviewer, and just to clarify, we do not claim that mTORC2 directly phosphorylates NDRG1. In fact, **we indicated in the text that NDRG1 C-terminus is extensively phosphorylated by SGK1 which is the intermediate kinase between mTORC2 and NDRG1.** In support of this, we know that mTORC2 activates SGK1 and that NDRG1 is a known SGK substrate. In addition, we presented data in the previous submission that knock down of *Sgk1-3* led to similar mitochondrial phenotypes as observed in *Rictor* KD cells, *NdrG1* KD cells, and in mutant NDRG1-expressing cells/livers. In sum, as advised by the reviewer, we have edited the wording here and throughout the manuscript to reflect this.

13. 'we expressed FLAG-tagged NDRG1 WT or NDRG1S336A in livers silenced for endogenous *NdrG1* (Extended Data Fig. 12c) and confirmed equivalent FLAG expression by immunohistochemistry (Extended Data Fig. 12d).' Data in Ext Fig 12d suggest that NDRG1S336A expresses at higher level than wt.

Answer: We thank the reviewer for this comment. We have now included in the revised manuscript the quantification for the intensity of FLAG expression and our IHC data confirm **equivalent FLAG levels in livers expressing FLAG-tagged NDRG1^{WT} or NDRG1^{S336A}** (Fig 8e).

Legend. Immunohistochemistry and quantification for equivalent FLAG expression in livers silenced for endogenous *NdrG1* and then injected with Flag-tagged WT or S336A NDRG1 plasmid. 2° Ab-only control is shown (n=3 mice). Individual replicates and means are shown. *P<0.05, One-way ANOVA followed by Tukey's multiple comparison test. ns = not significant.

3

14. 'Since NDRG1 does not exhibit intrinsic GTPase activity, a critical requirement for membrane scission, we asked whether P-NDRG1S336 engages with proteins with intrinsic GTPase activity to facilitate fission.' Please add a reference showing GTPase activity is required for membrane scission. NDRG1 is known to be an effector protein for Rab4a. The involvement of Rab4a in this process should be investigated (see also point 17).

Answer: We thank the reviewer for this comment. We have now added references to support the point that GTPase activity is required for mitochondrial membrane scission (*Nat Struct Mol Biol.* 2011;18(1):20-6, *Nat Commun.* 2018;9(1):5239).

In light of the idea that NDRG1 is an effector protein for Rab4a, we have now evaluated the role of Rab4a in the process. Our new data suggest that the loss of *Rab4a* facilitates mitochondrial fission, as indicated by increases in mitochondrial number, decreased mitochondrial size parameters (area, perimeter, elongation) and increased mitochondrial circularity when compared to controls. We therefore confirm the previously shown data that loss of Rab4a stimulates mitochondrial fission (*Ann Rheum Dis.* 2014;73(10):1888-97). Consequently, **the proposed mechanism for the regulation of mitochondrial fission by NDRG1 appears to be independent of Rab4a.**

The text now reads, "Since NDRG1 is an effector for Rab4a³⁰ and known to regulate endosomal recycling³¹, we asked if Rab4a participates in NDRG1-mediated mitochondrial fission. Since loss of Rab4a increased mitochondrial fission indicated by increases in mitochondrial number, decreased mitochondrial area, perimeter, and elongation, and increased circularity (data not shown), as also described previously³², regulation of mitochondrial fission by NDRG1 appears to be Rab4a-independent."

Legend. **a.** Relative mRNA expression of *Rab4a* gene in siControl or siRab4a NIH3T3 cells. **b.** Representative MitoTracker green fluorescence in serum-starved siCon or siRab4a NIH3T3 cells. Magnified insets are shown. Quantification for mitochondrial number and mitochondrial shape descriptors is shown in **c** (n=4 independent experiments). Individual replicates and means are shown. *P<0.05 and **P<0.01, Student's t-test.

1

15. 'we used proteomics to find proteins bound to FLAG NDRG1 WT but not NDRG1S336A.' Although wt NDRG1 will have the capability of being phosphorylated, do the authors know that it is actually phosphorylated under the experimental conditions used. This would be important to verify to validate the binding results.

Answer: We thank the reviewer for this question. We draw the reviewer's attention to (Extended Data Fig 6d-f) where we performed phosphoproteomics in MAM fractions from 14-16 h fasted (*experimental condition used throughout the manuscript*) Con and *Rictor*^{KO} livers, because it is likely that localized NDRG1 pools in MAMs are phosphorylated by mTORC2. To support this, we annotated MS/MS spectrum of phosphopeptide (SRTASGSSVTS(p)LEGTRSR) in NDRG1^{WT} wherein S(p) represents Ser336, and we noted that relative abundance of this peptide was significantly reduced (P<0.05) in MAMs of *Rictor*^{KO} livers when compared to Con livers (Extended Data Fig 6d-f). In further support of this point, we also draw the reviewer's attention to Fig 1h where we show that under the experimental conditions tested, i.e., fed and fasted for 3, 8, 14 and 20 hours, we observe significant increases in SGK1 (Thr256) and NDRG1 (Thr346) phosphorylation with fasting in wildtype mice.

Finally, NDRG1 Ser/Thr C-terminus residues are known to be phosphorylated by SGK1 (*Pflugers Arch. 2009;457(6):1287-301*). Unfortunately, a phospho-specific antibody for NDRG1 Ser336 is not available.

Legend. Phosphoproteomics reveal mTORC2 phosphorylation of NDRG1 at Ser336 in MAMs. (a) MAMs from livers of 14-16 h fasted Con and *Rictor*^{KO} mice were subjected to phosphoproteomics. (b) Annotation and quantification of MS/MS spectrum of phosphopeptide SRTASGSSVTS(p)LEGTRSR in NDRG1 from Con and *Rictor*^{KO} MAMs are shown. S(p) is Ser336. (n=3 samples wherein 2 livers were pooled to generate 1 sample).

16. Data shown in Ext Fig. 13a should move to the main figures.

Answer: We thank the reviewer for this comment. Extended data Fig 13a is now Main Fig 6a.

2

17. 'Indeed, when compared to NDRG1 WT, NDRG1S336A displayed modestly reduced binding to Cdc42, Arhgef10 (a Rho GEF that activates Rho GTPases by stimulating GDP/GTP exchange), and Arhgap35 (Rho GAP that facilitates GTP hydrolysis to inactivate Rho GTPases)'

Is it known whether ArhGEF10 and ArhGAP35 are actually active GEF/GAPs towards Cdc42 in cells. Please add references in support of statement. It would be useful to check for Rab associated proteins as well.

Answer: We thank the reviewer for this comment. Recent studies have shown that there is extensive promiscuity among the regulators with greater promiscuity seen for inactivating RhoGAPs than the activating RhoGEFs (*Nat Cell Biol. 2020;22(4):498-511*). Recent studies also show that ArhGEF10 is a regulator for both RhoA and Cdc42, while ArhGAP35 is a regulator for RhoA (*Nat Cell Biol. 2020;22(4):498-511*). This coordinated and promiscuous use of different GAPs and GEFs was proposed to increase the combinatorial possibilities to control diverse downstream signaling by simultaneously engaging multiple Rho GTPases and to precisely tune their activities in physiology (*Nat Cell Biol. 2020;22(4):498-511*). Since we find that Cdc42, but not RhoA, interacts with NDRG1, we chose to focus on Cdc42. With regards to Rab-associated proteins, we checked our proteomic data (excel file included in the submission) and we found Rab-like 6 (Rabl6) and Ras-related 8a (Rab8a) in our list of proteins binding to mutant NDRG1^{Ser336Ala} normalized to NDRG1^{WT}; however, the role of Rabl6 and Rab8a in modulating NDRG1-mediated fission was not examined.

3

18. 'Furthermore, FLAG-NDRG1 WT interacts with mCherry-Cdc42 WT, but displays reduced interaction with mutant Cdc42T17N, which does not bind GTP (Extended Data Fig. 13c) indicating that NDRG1S336 phosphorylation and Cdc42-GTP binding are required for their interaction to promote fission.'

A reduced interaction with Cdc42 T17N is extremely difficult to see (even though quantification confirms apparent loss of binding). No details of how quantification was achieved are available. These details should be available for all data presented. How have the data been normalized?

A better experiment would be to use a GFP-effector construct to probe for active Cdc42 at MAMs. Currently the statement here should be modified to say the data suggest active GTP bound Cdc42 is required for fission, or that the data warrants further investigation.

Answer: We thank the reviewer for this comment. We have now included how the quantification was achieved in the Methods and the legends of the corresponding figure panels. After careful reassessment of the data and inclusion of new data from new replicates, we find that Cdc42 GTP-loading is not required for NDRG1-Cdc42 interaction; however, NDRG1^{Ser336} phosphorylation and Cdc42-GTP binding are each critical for fission. We have modified the text accordingly:

"Furthermore, FLAG-NDRG1^{WT} interacts with both mCherry-Cdc42^{WT} and mutant Cdc42^{Thr17Asn}, which does not bind GTP (Extended Data Fig. 9c) indicating that Cdc42-GTP binding is not required for Cdc42-NDRG1 interaction; however, NDRG1^{Ser336} phosphorylation and Cdc42-GTP binding are each critical for fission."

Legend: IB and quantification for FLAG and mCherry in OA-treated (2.5 h) NIH3T3 cells co-expressing FLAG-NDRG1^{WT} and mCherry-Cdc42^{WT} or mCherry-Cdc42^{Thr17Asn} mutant and subjected to pull-down of FLAG using FLAG M2 agarose (n=4). Quantification for the relative enrichment of mCherry in FLAG pull-downs was calculated by normalizing the densitometry value of mCherry to the densitometry value of FLAG. ns = not significant.

2 19. ‘Since Cdc42 is largely controlled through translocation’. This is at best only a partially true statement. Like all GTPases Cdc42 requires to be membrane bound and to be GTP loaded to be active. This statement should be qualified and references added.

Answer: We thank the Reviewer for this comment. We agree with the Reviewer’s statement and have modified the sentence and included appropriate references to support the same. The sentence reads as follows:

“Since Cdc42 action is regulated by membrane binding (*J Biol Chem.* 2012;287(8):5764-74) and GTP-loading (*J Biol Chem.* 2013;288(12):8531-8543), we suspect that mTORC2-driven NDRG1^{Ser336} phosphorylation is a signal to recruit Cdc42 to MAMs for its activation to drive fission.”

3 20. ‘Indeed, Cdc42 and RhoA26 (Fig 4b), but not dynamins, were enriched in MAMs from fasted control livers.’ Please add in statistical analysis for RhoA into Fig. 4b.

Answer: We apologize to the reviewer for not incorporating the quantification for RhoA in old Fig 4b. Please see the corresponding figure panel (now Fig 7d and 7e) in the answer to the next question, Q21.

3 21. ‘By contrast, MAMs from fasted RictorKO (Fig 4b) and NDRG1S336A livers (Fig 4c) each showed markedly reduced Cdc42 levels without affecting levels of RhoA’. Please add RhoA statistical analysis to Fig. 4b as some visual reduction appears to occur in RictorKOs. There are no data for Rho A in Fig. 4c, please amend statement.

Answer: We apologize to the reviewer for not incorporating the RhoA data in Fig 4c, and for not including the quantification for RhoA in Fig 4b and 4c (now Fig 7d and 7e). This has now been included in the revised figure.

Legend for Figures A and B. IB and quantification for indicated proteins in Hom, Mp, MAMs, Cyt and ER from livers of 14-16 h-fasted (A) 5-6 mo-old control and *Rictor*^{KO} (n=3-5 mice), or (B) 3-4 mo-old mice expressing *NDRG1*^{WT} or *NDRG1*^{S336A} plasmids co-injected with siRNA against endogenous *NdrG1* (n=4-6 mice).

22. The authors now shift to analysis in NIH3T3s, using exogenously expressed GFP-Cdc42. Why, as you would really need to look in liver extracts as had been used throughout the study? Surely looking in the literature would be just as useful as Cdc42 pulldowns in NIH3T3s?

Answer: We thank the reviewer for this comment. The reason why we moved to NIH3T3 cells is because the in vitro system is a cleaner model to carry out live cell imaging and co-IPs without the interference from lipids or bile acids that we typically see in liver. Nevertheless, the NIH3T3 mouse fibroblasts have been used as a model to elucidate roles for Cdc42 in: **A)** cellular growth and transformation (*Mol Cell Biol.* 1998;18(8):4689-97, *Mol Cell Biol.* 2002;22(19):6895-905, *Cell Cycle.* 2005;4(11):1675-82, *J Biol Chem.* 2012;287(8):5764-74), **B)** intestine regeneration during aging (*iScience.* 2021;24(4):102362), and **C)** cytoskeletal reorganization and cell motility during cell migration (*Mol Cancer Res.* 2008;6(4):555-67).

2

23. Of note, 5 enriched targets specifically belonged to the Cdc42 family of effectors and regulators: Cdc42 effector proteins (Cdc42ep4, Cdc42ep1, and Cdc42ep2), the Rho GDP Dissociation Inhibitor alpha (Arhgdia), and IQ motif-containing GTPase-activating protein 1 (Iqgap1), which is a downstream Cdc42 effector and an upstream Cdc42 scaffold protein (Fig 4e, Extended Data Fig 14b).'

Cdc42eps have several names, which should be noted here to aid readers understand the significance of these results: CEP4/BORG4, BORG5/MSE55/CEP1, BORG21/CEP2. Please explain how IQGAP1 is both an effector and a scaffold protein: add references to support both assertions. Arhgdia (and others) are more usually referred to as RhoGDI1 etc.

Answer: We thank the reviewer for this comment. We have now adopted the appropriate nomenclature for each of the effector proteins so as to aid readers to better understand the significance of these results.

Furthermore, we have explained in the manuscript how IQGAP1 is both an effector and a scaffold protein. Proper references have been added to support both assertions. The text now reads as follows:

"This is likely because IQGAP1 plays a dual role in regulation of Cdc42 as a scaffold upstream and effector downstream of Cdc42 (reviewed in Adv Biol Regul. 2016;60:29-35). As a scaffold protein, IQGAP1 provides a molecular link between Ca²⁺/calmodulin and Cdc42-mediated processes (J Biol Chem. 1999;274(1):464-70), while also serving as a downstream effector whereby Cdc42 enhances the F-actin-cross-linking activity of IQGAP1 during actin reorganization (J Biol Chem. 1997;272(47):29579-83)."

2

24. 'the atypical Rho GTPase/Rho GDP-dissociation inhibitor 1 (Rhotbt1)'. Presumably this is a typo – Rhobtb1 (as in Fig 4e). Please add reference to show RhoBTB1 is a RhoGDI.

Answer: We thank the reviewer for this comment. We apologize for this mistake because we were referring to RhoBTB1 (Rho-Related BTB Domain Containing 1) as opposed to the incorrect protein mentioned in the previous submission, i.e., atypical Rho GTPase/Rho GDP-dissociation inhibitor 1. We have now looked into the description of the identity of RhoBTB1 across several protein sites including NCBI, Genecards and UniProt, and as per all three sites, RhoBTB1 is Rho-Related BTB Domain Containing 1, and not Rho GDP-dissociation inhibitor 1. We apologize for this error on our part, and we have corrected this error in our text.

2

25. 'In addition, we included two identified binding partners of NDRG1, Arhgef10 and Arhgap35, which serve as GEF and GAP, respectively (Extended Data Fig 13a).' As previous – please cite references showing Arhgef10 and Arhgap35 are active against Cdc42.

Answer: We thank the reviewer for this question. We have addressed this question in question 17.

26. Further characterization of the involvement of Cdc42 interactors in mitochondrial fission is undertaken in NIH3T3 cells. Please justify the use of these cells and the relevance of these results to the overall story presented.

Answer: We thank the reviewer for this question. We carried out the bulk of the mechanistic analysis in NIH3T3 cells (example, NDRG1^{WT} binds to Cdc42 in NIH3T3 cells but mutant NDRG1^{S336A} fails to bind to Cdc42, and mitochondrial membrane potential is lost in NIH3T3 cells silenced for *Cdc42*). This is because cell biological analyses via imaging (*critical for studies into mitochondrial dynamics*) are difficult in livers, and we considered it reasonable to perform the screen to find Cdc42 interactors using the same cell line. In any case, candidates identified as Cdc42 interactors in NIH3T3 cells were tested for their enrichment in MAMs from livers in response to fasting. In addition, we validated phenotypes obtained in NIH3T3 cells by performing experiments in livers. Hence, we are confident that our findings in NIH3T3 cells are relevant to what we observe in livers.

Finally, as also pointed out in 22 earlier (**previous page**), NIH3T3 mouse fibroblasts have been used as a model to elucidate roles for Cdc42 in: **A) cellular growth and transformation** (*Mol Cell Biol.* 1998;18(8):4689-97, *Mol Cell Biol.* 2002;22(19):6895-905, *Cell Cycle.* 2005;4(11):1675-82, *J Biol Chem.* 2012;287(8):5764-74), **B) intestine regeneration during aging** (*iScience.* 2021;24(4):102362), and **C) cytoskeletal reorganization and cell motility during cell migration** (*Mol Cancer Res.* 2008;6(4):555-67).

27. 'In addition, silencing the Rho GDP dissociation inhibitor Arhgdia (but not Arhgdib), which releases Cdc42 from inhibition by Arhgdia, increased mitochondrial number'. Please clarify this statement.

Answer: We thank the reviewer for this comment. We have now clarified the statement in our manuscript. The statement now reads as follows:

*"Rho GDP dissociation inhibitor (RhoGDI) proteins are negative regulators of Rho GTPases by retaining Rho GTPases in the cytosol, inhibiting their GTPase activity, and preventing their interaction with GEFs, GAPs and effectors (reviewed in *Biochem J.* 2005;390(Pt 1):1-9). In keeping with this, we suspect that silencing the RhoGDI alpha (Arhgdia) releases Cdc42 from the inhibitory effect of Arhgdia/RhoGDI1, which leads to increased mitochondrial fission."*

28. 'Interestingly, not all Rho GTPases impact mitochondrial dynamics,' Please clarify this statement: why would the authors/readers think all GTPases would impact on one particular process?

Answer: We thank the reviewer for this comment. We agree with the reviewer that not all GTPases would not affect one particular process. Because a number of GTPases were found to interact with WT Cdc42, we used this experiment as an opportunity to demonstrate the specificity of one GTPase towards this process. We have modified the sentence as follows:

"Because a number of GTPases interact with Cdc42, it is interesting that not all Rho GTPases impact mitochondrial dynamics, since depletion of Rho GTPase family Rhobtb1 gene had no effect on mitochondrial morphology, while silencing IQGAP1 increased mitochondrial number (Fig 7c)."

2 **29.** ‘likely because Iqgap1 serves as both downstream Cdc42 effector and upstream Cdc42 scaffold protein’. Clarify this statement with a reference.

Answer: We thank the reviewer for this comment. As addressed in question 23, we have added references to clarify this statement.

3 **30.** ‘Since Arhgap35 inactivates GTPases, we anticipated that depleting Arhgap35 would stimulate Cdc42, leading to fission; however, knocking down Arhgap35 decreased mitochondrial number—reflecting fission failure (Fig 4f, Extended Data Fig 14d-g). This likely reflects the complex regulation of Cdc42 requiring subsequent inactivation to complete its function’. Alternatively, Arhgap35 is not active towards Cdc42 and affects, instead, an antagonistic GTPase.

Answer: We thank the reviewer for this insight. This is certainly a possibility and we have now included this sentence in our discussion.

31. ‘as well as specificity amongst the different effectors and regulators in stimulating fission during fasting.’ All this analysis is performed in NIH3T3s, so has little to do with fasting in the liver.

Answer: We thank the reviewer for this comment. **We are confident that effectors and regulators identified via experiments in NIH3T3 cells perform similar functions in the fasted liver.** The basis for this statement is: **A)** Knockdown of each of the proteins in our axis, i.e., *Rictor*, *Ndr1*, *Cdc42*, in NIH3T3 cells resulted in similar fission failure phenotypes and respiratory defects in liver, and **B)** the identified candidates binding to NDRG1 or Cdc42 in NIH3T3 cells were independently validated and found to be enriched in MAMs isolated from livers. Furthermore, in our previous studies into the role of lipophagy (*Nature*. 2009;458(7242):1131-5), we have found that NIH3T3 fibroblasts faithfully reflect changes in autophagy and lipid metabolism in liver. Hence, we are more than confident that findings in NIH3T3 cells replicate changes in mitochondrial dynamics, energetics and lipid metabolism as we have observed in liver over the years.

3 **32.** ‘Consistent with these findings, in addition to Cdc42, we also observed substantial enrichment in MAMs of Cdc42 effector, Cdc42ep1, and those of regulator proteins, Arhgap35 and Arhgdia (Extended Data Fig 15a, b).’ Statistical analysis presented shows no enrichment of the proteins suggested – please clarify.

Answer: We thank the reviewer for this comment. We agree with the reviewer. We apologize for the use of the term enrichment because we are not comparing the extent of accumulation in MAMs versus homogenates. Instead, we evaluated changes in levels of Cdc42 and its corresponding effectors and regulators in MAMs from control versus test conditions. Consequently, we have eliminated “enrichment of” and replaced it with “presence”. The sentence now reads:

“Consistent with these findings, in addition to Cdc42, we detected the presence in MAMs of Cdc42 effector, Cdc42ep1/BORG5, and Arhgap35 and Arhgdia/RhoGDI1 (Extended Data Fig 10c, d) ”.

33. 'Indeed, in control cells, actin assembled around mitochondrial subpopulations to generate ring-like structures and promote fission'. Although the data do suggest that ring-like actin structures are generated, the link to fission although possible or even likely is not proven. Please qualify statement.

Answer: We thank the reviewer for this comment. We have now incorporated a clear description of the known role of actin in mitochondrial fission, and on this basis, we have described how best our new data fit in this existing model of actin-mediated fission. The sentence now reads as follows:

"Since dynamic cycling of actin through populations of mitochondria controls fission^{41,42}, we asked whether Cdc42 mediates the effect of mTORC2-NDRG1^{Ser336} on mitochondrial fission by remodeling the actin cytoskeleton. Indeed, in control cells, actin assembled around mitochondrial subpopulations to generate ring-like structures consistent with maintained fission⁴² (Fig 7f), while silencing Cdc42 decreased colocalization of actin with mitochondria, which correlated with elongated mitochondria reflecting fission failure (Fig 7f), supporting our notion that Cdc42 facilitates the organization of actin around mitochondria to enable fission."

34. 'We demonstrate that NDRG1 associates with mitochondrial constrictions to facilitate fission (Fig 4h)'. Please qualify – the data suggest this and a potential model is shown in Fig 4h.

Answer: We thank the reviewer for this comment. We have expanded on the description of these results, and referred to appropriate figure panels so that it is easily conceived by the readers.

35. The data presented do suggest a role for Cdc42 in mTORC2-NDRG1 driven regulation of mitochondrial fission. However, the conclusions based on the final sections of the manuscript are less robust than other parts of the analysis.

Answer: We thank the reviewer for this comment. The focus of this work was on establishing a role of the mTORC2-NDRG1-Cdc42 axis in mitochondrial fission, which was mechanistically tested in this work using epistasis experiments, however, we have been clear in our statements that the role of the interactors and effectors of Cdc42 appears to correlate with those of mTORC2, NDRG1 and Cdc42.

Minor Comments

1. 'phosphorylation of the mitophagy adapter BNIP3 at Ser(S)79 and S88' Why revert to explaining the three letter code halfway through the manuscript? Move this to first use of S abbreviation for Serine or use three-letter code throughout (which is accessible to more readers).

Answer: We thank the reviewer for this comment. We have now corrected the discrepancy and used the three-letter code for a given amino acid throughout the manuscript.

2. 'SRTASGSSVTS(ph)LEGTRSR' – p is the more usual abbreviation to indicate phosphorylation in a sequence.

Answer: We thank the reviewer for this comment. We have now used "p" as the abbreviation to indicate phosphorylation in the sequence.

3. 'Several S/T residues on the NDRG1 C-terminus are phosphorylated by SGK1.' Start new paragraph here for clarity.

Answer: We thank the reviewer for this comment. We have now started this sentence as a new paragraph for clarity.

Reviewer 2 comments

Referee #2 (Remarks to the Author):

This is an extremely interesting article by Martinez-Lopez and colleagues that links mTORC2 to mitochondrial fission and shows that fasting regulates this process via mTORC2 and NDRG1. It should be noted that the processes downstream of mTORC2 that are Akt-independent have remained mysterious for some time, and thus this manuscript addresses a major gap in knowledge. That said, there are some real question outstanding here that should be addressed prior to publication.

4 **1. In my own personal experience, liver mTORC2 activity reaches a nadir after about 22 hours of fasting. The authors should show that data showing that mTORC2 activity reaches a minimum at 16 hours.**

Answer: We thank the reviewer for this comment. The reviewer is correct. As can be seen in the original submission in **Fig. 1h**, fasting-induced increase in mTORC2 signaling as determined by P-AKT^{Ser473}/AKT is highest at 14 hours (h) (p=0.0424 compared to fed mice), and P-SGK1^{Thr256}/SGK1 and P-NDRG1^{Thr346}/NDRG1 are highest at 20 h (p=0.0276 and p=0.0249, respectively). This effect is lost by 20 h suggesting that, as indicated by Reviewer 2, mTORC2 activity is beginning to decline with progressing periods of fasting beyond 14 h, and likely reaches a minimum at 16 h.

4 **2. The experiment showing mTORC2 activation by corn oil gavage does not have a gavage control - the stress of gavage could be leading to glucose and insulin release and mTORC2 stimulation. This should be examined. While the authors examined and dismissed a contribution of insulin, they did not examine IGF-1 or leptin, which could regulate mTORC2 via PI3K.**

Answer: We thank the reviewer for this comment. These controls are important, and we have now included this in our manuscript. We apologize for not including the description of the controls in the methods of the original submission, which we have now included in the methods. These are: **A)** Animals were acclimatized to gavage by mock handling and gavages for 5 days prior to experiment, and **B)** on the day of the experiment, control animals were gavaged with vehicle (10% DMSO-saline solution) to match for volume and stomach distension.

Furthermore, as suggested by Reviewer 2, we tested if **reactivation of mTORC2 is coupled to changes in leptin or IGF-1. To test this possibility, we measured circulating levels of leptin and IGF-1 in fed and fasted mice that were gavaged or not with corn oil, and as can be seen from our data, fasting *per se* reduces levels of leptin (Fig A) (J Clin Endocrinol Metab. 1996;81(9):3419-23) and IGF-1 (Fig B) (Ageing Res Rev. 2019;53:100910) when compared to fed controls, and corn oil gavage does not lead to increases in leptin or IGF-1 in the fed or fasted state.**

Legend for Figures A and B. mTORC1/C2 signaling in presence of lipid appears to be leptin and IGF-1-independent. Circulating (A) leptin and (B) IGF-1 levels in 3 mo-old C57BL/6 male mice that were fed or fasted for 14-16 h and then gavaged with corn oil for 30 min (n=6-9). *P<0.05, **P<0.01 and ****P<0.0001, 2-way ANOVA followed by Tukey's correction. ns = not significant.

1

3. The effect of mTORC2 activity on NDRG1 should be mediated by changes in SGK, yet no evidence of SGK phosphorylation/activation is shown.

Answer: We thank the reviewer for this comment. We have now included in the revised manuscript, our assessment of phosphorylation of SGK1 (Thr256) in response to fasting, and our new data indicate that P-SGK1^{Thr256}/SGK1 levels begin to increase by 14 h and are significantly higher at 20 h of fasting compared to fed controls (**Fig 1h**). This is consistent with significant increases in phosphorylation of the SGK1 substrate, NDRG1^{Thr346}, at 20 h (**Fig 1h**).

Legend. IB and quantification for indicated proteins in livers of 5-10 months (mo)-old male and mice that were fed or fasted for the indicated durations (n=5-12 mice). Ponceau is loading control. Individual replicates and means are shown. *P<0.05, One-way ANOVA followed by Tukey's correction.

In addition, we individually silenced SGK, AKT or NDRG1 in vitro (**Extended Data Fig. 8a**) and in vivo (**Extended Data Fig. 8b-c**), and we noted that cells silenced for *NdrG1* or *Sgk1-3* exhibited the fission failure phenotype observed in *Rictor* KD or KO cells, whereas silencing *Akt1/2* did not have any effect on fission. Similarly, livers of fasted mice knocked down for *NdrG1* or *Sgk1* showed reduced respiration (**Extended Data Fig. 8c**) as observed in livers of fasted *Rictor*^{KO} mice, while loss of *Akt1/2* had no effect on mitochondrial respiration during fasting. **Hence, our results suggest that the mTORC2-SGK-NDRG1 cascade, and not the mTORC2-AKT cascade, is required for mitochondrial fission during fasting.**

Legend. Silencing NDRG1 and SGK1, but not AKT1/2, recapitulates mitochondrial fission failure observed in Rictor silenced cells. (a) Representative confocal images of NIH3T3 cells silenced for *Sgk1/2/3*, *NdrG1* or *Akt1/2* and cultured in serum-free medium for 30 min in presence of MitoTracker green. Magnified insets are shown. Quantifications for mitochondrial number and mitochondrial size/shape descriptors are shown (36-71 cells from 3-5 independent experiments). (b) IB and quantifications for the indicated proteins in livers of 3-4 mo-old male mice injected with siRNAs against *Sgk1*, *Akt1/2* or *NdrG1* and subjected to 14-16 h fasting (n=3-4). (c) AUC for OCR in siSgk1, siAkt1/2 or siNdrG1 livers of 14-16 h fasted mice (n=3-5). Ponceau is loading control. Individual replicates and means are shown. *P<0.05, **P<0.01, ***P<0.001 and ****P<0.0001, One-way ANOVA and Tukey's multiple comparisons test (a, c); Unpaired Student's t-test (b); ns = not significant.

4

4. Regulation of Rictor by SIRT1 in the liver has previously been shown (PMID: 21965330), and could explain the mTORC2 activation in response to feeding - this should be examined.

Answer: The mechanism of activation of mTORC2 or mTORC1 by changes in nutrient availability are highly complex processes that may require their localization to specific organelles, which as we know for mTORC1 are lysosomes, and it may be possible that for mTORC2 are ER or ER-mitochondrial contact sites. Hence, mechanisms of mTORC2 activation are a whole different project on its own, and this is the reason why this paper focusses on the downstream effects of fasting-driven signaling on mitochondrial fission and not on how fasting reactivates mTORC2. Consequently, we respectfully urge the reviewer to consider that extensive exploration of upstream mechanisms of mTORC2 activation are beyond our means and the scope of this manuscript.

However, because SIRT1 is induced by fasting (*Mol Cell Biochem.* 2010;339(1-2):285-92), and because of existing data in literature showing a role of SIRT1 in driving *Rictor* gene expression, we assessed the regulation of *Rictor* gene by SIRT1 in our cells. As shown previously (*J Clin Invest.* 2011;121(11):4477-90), our new data show that cells knocked down for *Sirt1* display decreased *Rictor* mRNA expression when compared to siControl cells. Hence, it is likely that SIRT1 induction during fasting plays a permissive role in fasting-induced mTORC2 reactivation during fasting.

Legend. Relative mRNA expression of the indicated genes in NIH3T3 cells transfected or not with siRNAs against *Sirt1* (n=4-8). Individual replicates and mean values are shown. **P<0.01 and ****P<0.0001, Student's t-test.

2

5. It would be interesting to compare and contrast the data gathered by the authors here with the phosphopeptide data found in fasted and refed RKO mice by others (PMID: 24072782). Among other issues, this previous study identified NdrG2 phosphosites as some of the most heavily downregulated, and the overlap between NdrG1/2/3 function is not clear.

Answer: We thank the reviewer for this comment. As rightfully indicated by reviewer 2, phosphorylation of NDRG1/2/3 as indicated in PMID: 24072782 is not aligned with what we see in our paper, and this is because:

A) Direct comparison of our paper with PMID: 24072782 is difficult **because phosphoproteomic analyses in both manuscripts are done in completely different nutritional conditions.** In PMID: 24072782, analysis was done in mice fasted overnight and then refed for 3 hours with the focus on feeding-driven mTORC2 signaling, whereas our phosphoproteomic analysis was performed in livers of 14-16 h fasted Con and *Rictor*^{KO} mice, and we focused on reactivation of mTORC2 during fasting. As we all know, 3 h of refeeding will stimulate growth factor/insulin signaling leading to phosphorylation of completely different subsets of proteins when compared to fasting-induced reactivation of mTORC2. Finally, the *Rictor*^{KO} model used in PMID: 24072782 is based on loss of *Rictor* from birth (Albumin-Cre line) while we used livers that were silenced conditionally during adulthood for a period of 8 weeks. Consequently, for many of these reasons, it may be difficult to compare the proteomic analysis from both studies.

B) Although NDRG2 was identified as a substrate of mTORC2 in the 3 h refed state in PMID: 24072782; in our study, **loss of mTORC2 resulted in increased NDRG2 phosphorylation at Ser352, Ser353, Ser355 and Thr357** when compared to controls. It is clear that the C-terminus of NDRG2 is phosphorylated in *Rictor*^{KO} livers as can be seen below, and as suggested previously, it would seem that this part of NDRG2 is mostly disordered and a known target for additional kinases (<https://doi.org/10.1074/jbc.M401504200>).

Furthermore, calculation the deltaP for both NDRG1 and NDRG2 suggests that NDRG1 is hypophosphorylated in *Rictor*^{KO} liver while NDRG2 is hyperphosphorylated. Because of these reasons, **we focused on the downregulated NDRG1 phosphosites** and not on NDRG2 in our study.

2

6. A number of studies have shown that Rictor deletion shortens worm and mouse lifespan for unknown reasons - it seems logical to ask if any of these effects might be due to diminished mitochondrial fission.

Answer: We thank the reviewer for this exciting comment. This is a very important point that reviewer 2 has raised, and we are currently in process of generating data to support the idea that fasting-induced mTORC2 reactivation is impacted with aging and correlates with reduced mitochondrial fission in aged livers.

If this holds true, then it is logical to consider that the shortened lifespan in worms and mice upon *Rictor* loss could be in part due to impaired mitochondrial fission/function. In preliminary support of this exciting point raised by reviewer 2, we present two pieces of preliminary data (not in the manuscript) that we are working on for a different manuscript where we observe that **mTORC2 reactivation by fasting is dampened with age, as indicated by the failure of aged livers to increase their P-AKT^{Ser473}/AKT levels at 14 hours of fasting when compared to young controls**. We also observe a varying trend of decreased Rictor protein levels with age, which, as per this reviewer's point in a previous question, maybe due to age-related reduction in Sirt1 activity. Importantly, **age-related loss of mTORC2 activity correlates with impaired mitochondrial fission during fasting, as indicated by decreased number of mitochondria (please see next page)**. Consistent with this point, **we see reduced frequency of smaller mitochondria and higher frequency of larger mitochondria, as reflected by changes in mitochondrial area, perimeter and circularity, in aged livers, reflecting impaired fission (please see next page)**.

Legend. IB and quantification for indicated proteins of the mTORC2 pathway in livers of young (4-7-mo-old) and aged (30-mo-old) mice that were fed or fasted for the indicated durations (n=3-4 mice). Ponceau is loading control. Individual replicates and means are shown. *P<0.05 and **P<0.01, 2-way ANOVA followed by Tukey's multiple comparisons test.

2 Answer to Q6 continued:

Legend. (a) Representative EMs in fed or 14-16 h fasted young (4-7-mo-old) and aged (30-mo-old) livers (n=3-9 mice). Quantification for mitochondrial number (b) and frequency histograms depicting distribution of mitochondrial (c) area, (d) perimeter and (e) circularity are shown. Individual replicates and means are shown in b. Mean \pm SEM is shown in c-d. * $P < 0.05$, ** $P < 0.01$, *** $P < 0.001$ and **** $P < 0.0001$. 2-way ANOVA followed by Tukey's (b) or Šidák's multiple comparison test (c-d).

3

7. p38 MAPK has been identified as a signaling node downstream of liver mTORC2 and possibly SGK (PMID: 24072782) and more recently DRP1 has been identified as a substrate of p38 MAPK (PMID: 31881179). Could mTORC2-p38 MAPK-DRP1 signaling be contributing to the observed phenotypes?

Answer: We thank the reviewer for this question. It is possible that mTORC2 signaling also converges on p38 MAPK-DRP1 axis in mitochondrial fission; however, because our MAMs analysis showed no differences in levels of phosphorylated or total Drp1 in fasted *Rictor*^{KO} livers when compared to controls, we focused on identifying additional GTPases that lie downstream of NDRG1. Having said that, as per Reviewer 2's suggestion, we tested whether P-p38 MAPK^{T180/Y182} levels are reduced in fasted *Rictor*^{KO} mice, and our data show **similar P-p38 MAPK^{T180/Y182}/P38 levels in Con and *Rictor*^{KO} fasted livers, likely excluding the possibility of the contribution of the mTORC2-p38 MAPK-DRP1 axis to the observed phenotype in our studies.**

Legend. IB and quantification for the indicated proteins in livers of 4-6 mo-old Con and *Rictor*^{KO} male mice fasted for 14-16 h (n=8-9 mice). Ponceau is loading control. Individual replicates and means are shown. ns = not significant.

2

8. Could the authors please discuss why fasting should activate mTORC2, which would seem to be counterproductive?

Answer: We thank the reviewer for this comment. We have included a discussion of why we feel mTORC2 is reactivated during fasting. The text now reads as follows:

*“Fasting and feeding are metabolically and hormonally very distinct physiological states¹. While nutrient deprivation in cultured cells has been shown to block mitochondrial fission so as to preserve ATP synthesis and cell viability^{44,45}, cells in culture do not completely recapitulate the complex physiology of an intact whole organism. In fact, in context of the liver, a highly integrated organ system that plays a key role in handling metabolic transition states, our findings demonstrate marked increases in fission in response to an acute fast. Indeed, switching between nutrient availability and its absence thereof has been shown to modulate mitochondrial cristae and ER contacts, which per se impact mitochondrial dynamics^{46,47}. In keeping with this, we suspect that fasting-induced increases in adipose lipolysis, and supply of copious amounts of circulating FFA and various other lipid species to liver, leads to activation of key signaling modules, for instance mTOR. Indeed, cholesterol⁴⁸ and phosphatidic acid⁴⁹ have each been shown to activate mTORC1, and we show here that exposure to dietary corn oil or exogenously supplied FFA or fasting each activates mTORC2 signaling in liver, as has been shown for mTORC1 in starved cultured cells⁵⁰. In addition, fasting stimulates SIRT1 activity⁵¹, which in turn drives mTORC2 signaling in liver⁵². Since silencing *Sirt1* reduced expression of *Rictor* in NIH3T3 cells (data not shown), it is also possible that fasting-induced increases in SIRT1 activity facilitates mTORC2-driven fission.”*

Reviewer 3 comments

Referee #3 (Remarks to the Author):

The study by Martinez-Lopez et al. aims to establish a link between fasting in liver and mitochondrial fission via mTORC2, via an initial unbiased phosphoproteomics screen in mice that are fed, fasting or challenged with varying diets. Fasting for prolonged periods led to a re-activation of mTORC2, and mitochondrial fission that was dependent upon the TORC1 component Rictor.

The authors show that mTORC2 phosphorylates N-myc downstream regulated gene 1 (NDRG1), an established stress-response protein. In turn, NDRG1 associates with mitochondrial constriction sites and recruits the Rho GTPase Cdc42 to promote mitochondrial fission. Overall, this manuscript suggests of the existence of an mTORC2-NDRG1-Cdc42 axis to support fission during fasting and will be of interest to the Nat Cell Bio community.

Some comments:**Major Concerns:****4**

1. mTORC1 activity is impaired by nutrient or energy deficiency. However, here the authors suggest that mTORC2 specifically is reactivated upon fasting or short refeeding with exogenous fatty acids. The mechanism behind this TORC2 activation is unclear and requires further evaluation in the manuscript – is it due to free mTOR from turned over mTORC1?. Further, AMPK targets within the signaling cascade, beyond P-AMPK, should be measured.

Answer: We thank the reviewer for this question. The mechanism by which lipids/fasting directly reactivate mTORC2 is beyond the scope of this manuscript, and this is another project that is currently under investigation in our laboratory. In this work, the focus is to understand the function and relevance of fasting-induced reactivation of mTORC2.

Although the reviewer raises an interesting point, we tend to disagree with the possibility that mTORC2 activation during fasting is because more of mTOR protein is available due to the assumption that mTORC1 is suppressed during fasting. This is because only short-term fasting (3-6 h) shuts down mTORC1 activity (as indicated by reduced levels of the mTORC1 downstream target P-S6^{S235/236}/S6 when compared to fed controls); however, **as described previously (Nature. 2010;465, 942-946), and validated further in our manuscript (Fig. 1h), there is sustained reactivation of mTORC1 beyond 6 hours of fasting. Hence, it is unlikely that mTORC2 reactivation is due to free mTOR protein.**

As suggested the Reviewer, we have now included here our analysis of phosphorylation of the AMPK downstream signaling target, acetyl-CoA carboxylase (ACC), at Ser79, and **our new data (not included in manuscript) show trends towards increases in p-ACC^{Ser79}/ACC levels at 20 hours of fasting**, however this remained statistically insignificant (P=0.103).

Legend. IB and quantification for the indicated proteins in livers of 5-10 mo-old male mice that were fed or fasted for the indicated durations (n=5-12 mice). Ponceau is loading control. Individual replicates and means are shown.

4 **2.** The TORC2 effects (raptor knockouts) on fasting induced fission are striking. To ensure that this is TORC2 inhibition specific and not via changes to mTORC1 (as they did in TG studies) fasting effects on mitochondria in TSC deletions should be examined.

Answer: We thank the reviewer for this comment. As pointed by the reviewer, specific inactivation of mTORC2 (*Rictor*^{KO}), but not mTORC1 (*Raptor*^{KO}) markedly increased liver TGs and impaired mitochondrial respiration (Fig. 2c-e), pointing to a selective role of mTORC2 in mitochondrial function.

This is also an interesting suggestion from the Reviewer. To this end, we have now included new data showing that loss of *Tsc1* (hyperactivation of mTORC1) results in increased mitochondrial number and reduced mitochondrial size (diminished area and perimeter) when compared to controls (Fig A). These results are consistent with the known role for mTORC1 in stimulating mitochondrial fission (Mol Cell. 2017;67(6):922-935.e5). This described mechanism involves the regulation of levels of the fission factor protein MTFP1 by mTORC1. As can be seen in response to question 5 of this document, MTFP1 protein levels remained unaltered in *Rictor*^{KO} livers when compared to controls. In addition, loss of *Rictor* did not result in changes in the phosphorylation of mTORC1 downstream target P-P70 at Thr389 (Fig B), suggesting that the effect of the loss of mTORC2 activity on mitochondrial fission is not due to changes in mTORC1 activity.

Legend for Fig A. **a.** Representative blot for Tsc1 protein levels in siControl or siTsc1-transfected NIH3T3 cells. Ponceau is loading control. **b.** Representative MitoTracker green fluorescence in serum-starved siCon or siTsc1 NIH3T3 cells. Magnified insets are shown. Quantification for mitochondrial number and mitochondrial shape descriptors is shown in **c** (n=4 independent experiments). Individual replicates and means are shown. *P<0.05, Student's t-test.

Legend for Fig B. Representative blot and quantification for the indicated proteins in livers of 4-6 mo-old Con and *Rictor*^{KO} male mice fasted for 14-16 h (n=7-10 mice). Ponceau is loading control. Individual replicates and means are shown. ns = not significant.

1

3. Authors claim that “mTORC2 stimulates mitochondrial fission by phosphorylating NDRG1 at S336”. However, there is no proof that mTORC2 directly interacts and phosphorylates NDRG1 or whether it is a secondary kinase that phosphorylates NDRG1

Answer: We thank the reviewer for this comment. Based on literature, we know that NDRG1 is phosphorylated by a kinase that lies downstream of mTORC2, i.e., SGK1. **Please see our response to Reviewer 1 question 12. For your convenience, answer to Reviewer 1 question 12 is appended below:**

Reviewer 1 Q12. (Extended Data Fig. 8h-j) showed that this phosphopeptide is markedly less enriched in *siRictor* cells compared to *siCon* cells—confirming that mTORC2 phosphorylates NDRG1S336 in vitro and in vivo in MAMs. Unfortunately, this and much other data presented in this manuscript suggests that mTORC2 regulates NRG1 phosphorylation, not that NRG1 is a direct substrate for mTORC2. Kinase cascades are so prevalent they cannot be ruled out and only work with purified proteins will prove a direct relationship. Please edit wording here and throughout the manuscript to reflect this.

Answer: We agree with the reviewer, and just to clarify, we do not claim that mTORC2 directly phosphorylates NDRG1. In fact, **we indicated in the text that NDRG1 C-terminus is extensively phosphorylated by SGK1 which is the intermediate kinase between mTORC2 and NDRG1.** In support of this, we know that mTORC2 activates SGK1 and that NDRG1 is a known SGK substrate. In addition, we presented data in the previous submission that knock down of *Sgk1-3* led to similar mitochondrial phenotypes as observed in *Rictor* KD cells, *NdrG1* KD cells, and in mutant NDRG1-expressing cells/livers. In sum, as advised by the reviewer, we have edited the wording here and throughout the manuscript to reflect this.

3

4. Drp1 has multiple post-translational modifications that dictate its GTPase function and fission activity. However, only Drp1 P-S616 is analyzed in this manuscript (i.e., Extended Data Fig. 6g). The Drp1 S637 site has been implicated in mitochondrial hyperfusion upon nutrient deprivation (PMCID: PMC3088644, PMCID: PMC3121813). Therefore, Drp1 P-S637 levels need to be measured. Then, analyze whether mTORC2-NDRG1-Cdc42 co-localize with Drp1 by imaging experiments.

Answer: We thank the reviewer for this important question. The reviewer is right that Drp1 Ser637 site has been implicated in mitochondrial hyperfusion upon nutrient deprivation, and we apologize for not including this phosphorylation in our manuscript. Our new data indicate that, as expected, loss of *Rictor* (*Rictor*^{-/-}) significantly decreases the phosphorylation of mTORC2-downstream targets AKT (at Ser473) and NDRG1 (at Thr346), **however, the phosphorylation of Drp1 at Ser616 (drives fission) or Ser637 (blocks fission) remain unaltered when compared to control cells (*Rictor*^{+/+}) (Fig A, next page 5 of this document).** Similarly, *Rictor*^{KO} livers from fasted mice showed no differences in Drp1 Ser616 or Ser637 phosphorylation compared to Con mice (Fig B, next page 5 of this document). These data support that mTORC2-NDRG1 axis drives mitochondrial fission in the absence of any changes to DRP1 phosphorylation.

3 Answer to **Q4** continued:

Legend for Fig A. IB for the indicated proteins in *Rictor*^{+/+} and *Rictor*^{-/-} MEFs (n=4-7). Ponceau is loading control. *P<0.05, **P<0.01 and ***P<0.001, Unpaired Student's t-test.

Legend for Fig B. IB and quantification for the indicated proteins in liver of 4-6 mo-old Con and *Rictor*^{KO} male mice fasted for 14-16 h (n=7-9 mice). Ponceau is loading control. Individual replicates and means are shown.

5. In this manuscript, mice are fasted for 14-16h and then refeed the respective fatty acid/ diet treatments for 30 min. Refeeding protocols after fasting last between 1-6h. This short timing might be the reason why mTOR is not activated in the adipose tissue. Is mTOR activated in the adipose tissue after challenging mice on the respective diets for longer time periods?

Answer: We thank the reviewer for this question. However, in discussing with the Editor, we feel that addressing the question if mTOR is activated in adipose tissue after challenging the mice on the respective diets is not directly relevant to the topic of investigation.

Minor Concerns:

3

5. Previous literature demonstrated that mTORC1 stimulates mitochondrial fission via the mitochondrial fission factor MTFP1 (PMID: 28918902). What is MTFP1 expression and activity levels in the KO mice?

Answer: We thank the reviewer for this question. Immunoblotting for MTFP1 expression in Con and *Rictor*^{KO} livers revealed no changes in MTFP1 protein levels in *Rictor*^{KO} when compared to control livers, which excludes the possibility that impairment in mitochondrial fission in *Rictor*^{KO} livers is due to mTORC1-mediated regulation of MTFP1 protein levels.

Legend. Immunoblot and quantification for MTFP1 protein levels in livers of 4-6 mo-old Con and *Rictor*^{KO} male mice fasted for 14-16 h (n=7-10 mice). Ponceau is loading control. Individual replicates and means are shown. ns = not significant.

6. Mitochondrial elongation protects mitochondria from mitophagy upon starvation (PMCID: PMC3088644). Thus, mitophagy activity should be evaluated in the RictorKO livers that display reduced mitochondrial fragmentation.

Answer: We thank the reviewer for this question. However, in discussing with the Editor, we feel that addressing the question whether mitophagy activity should be evaluated in the *Rictor*^{KO} livers is beyond the scope of this manuscript.

7. Quantification of mitochondrial morphology from EM images should not only be limited to 2D parameters, but also include measurement of mitochondrial volume to assess mitochondrial ultrastructure. Further, confocal images should include circularity as an evaluator of mitochondrial fragmentation.

Answer: We thank the reviewer for this comment. There are extensive literature supporting that changes in mitochondrial characteristics such as **mitochondrial number** and **mitochondrial size** reflect changes in mitochondrial dynamics using 2D imaging using confocal and/or transmission electron microscopy. To this end, **tabulated below (please see page 8 of this document) are papers from established labs from the mitochondrial field that have used mitochondrial number, mitochondrial area and mitochondrial length or perimeter to reflect changes in mitochondrial dynamics.** Accordingly, we have performed all our experiments and analyses in all our models, and in parallel, included positive controls for loss of fission and fusion by knocking-down *Drp1*, *Mff* or *Mitofusin1* and *Opa1*, respectively, to show that loss of the mTORC2-NDRG1-Cdc42 axis results in a mitochondrial phenotype that mimics those observed when *Drp1* or *Mff* is knocked down in vitro (**Fig. 6f**) and in vivo (**Fig. 3a**).

Answer to Q7 continued:

MITOCHONDRIAL PARAMETER	FISSION-DEFICIENT MODELS	FUSION-DEFICIENT MODELS Hyper-fission (Drp1-oe) models
NUMBER	Decreased number	Increased number
	Drp1-KD HeLa cells Metabolites. 2021; 11(5): 322 EMBO J. 2019;38(8):e99748 Drp1-KD cells Nature. 2019;570(7761):E34-E42 Nature. 2016; 540(7631): 139–143 Drp1-KO Liver Diabetologia. 2015;58(10):2371-80 Mff-KO or Drp1-KO or Fis1-KO MEFs Mol Biol Cell. 2013;24(5):659-67.	Mfn1-KO liver Diabetes. 2016;65(12):3552-3560 Mfn2-KO liver or Mfn2-KO muscle Proc Natl Acad Sci U S A. 2012;109(14):5523-8 Opa1-KD HeLa cells Methods. 2008;46(4):295-303 Opa1-KO muscle Cells. 2021; 10(9): 2177 Drp1-overexpressing HeLa cells Mol Biol Cell. 2016;27(1):20-34.
AREA	Increased	Decreased
	Mff-KO Liver Diabetologia. 2021;64(9):2092-2107 Drp1-, Mff- or Fis-KD HeLa cells Mitochondrion. 2021;59:216-224 EMBO J. 2019;38(8):e99748 Drp1-KD cells Nature. 2019;570(7761):E34-E42 Nature. 2016; 540(7631): 139–143 Drp1-KO muscle Nature Communications. 2019;10(1):2576 Drp1-KO Liver Diabetologia. 2015;58(10):2371-80 Mff-KO or Drp1-KO or Fis1-KO MEFs Mol Biol Cell. 2013;24(5):659-67.	Mfn2- or Opa1-KD HeLa cells Mitochondrion. 2021;59:216-224 Opa1-KO muscle Cell Metab. 2017;25(6):1374-1389.e6 Cells. 2021; 10(9): 2177 Mfn1/Mfn2-KO heart Cell Death Dis. 2016;7(5):e2238 Drp1-overexpressing HeLa cells Mol Biol Cell. 2016;27(1):20-34.
LENGTH OR PERIMETER	Increased	Decreased
	Mff-KO Liver Diabetologia. 2021;64(9):2092-2107 Drp1-KD cells Nature. 2019;570(7761):E34-E42	Mfn1-KO or Opa1-KO human HCC cell lines and human CCA tumor organoids Cells. 2020; 9(1): 121 Mfn2-KO adipose tissue EMBO J. 2017;36(11):1543-1558 Mfn1-KO liver Diabetes. 2016;65(12):3552-3560 Opa1-KD HeLa cells Methods. 2008;46(4):295-303

Answer to **Q7** continued:

Answer: We agree on this point about the circularity, and we have now performed the analysis of mitochondrial circularity for each of our experiments, and these data are now included in the manuscript. Briefly, analysis for circularity shows that ***Rictor*, *Ndrg1* or *Cdc42* knock down each phenocopied the reduced circularity seen in cells silenced for the fission factors *Mff* and *Dnm11* (Fig quantifications b-f)**. Please see also in the following page (page number 10) circularity data for MEFs, AML12 and HepG2 cells where we consistently see **reduced circularity upon loss of *Rictor* in MEFs (page 10, Fig A), AML12 (page 10, Fig B), and HepG2 (page 10, Fig C) cells when compared to each of their corresponding controls.**

Legend. Loss of *Rictor* or *Ndrg1* or *Cdc42* each shows the same mitochondrial fission failure phenotype as when known fission factors, *Drp1* or *Mff*, are silenced.

(a) Representative confocal images of NIH3T3 cells transfected with the indicated siRNAs and cultured in serum-free medium for 30 min in the presence of MitoTracker green to stain mitochondria. Magnified insets are shown. **b-f.** Quantification for **(b)** mitochondria number, **(c)** area, **(d)** perimeter, **(e)** circularity and **(f)** elongation (64-142 cells from n= 5-8 independent experiments). Gray areas indicate mitochondria fission-deficient models. Individual replicates and means are shown. * $P < 0.05$, ** $P < 0.01$, *** $P < 0.001$ and **** $P < 0.0001$, One-way ANOVA followed by Dunnett's multiple comparisons test. Scale = 10 μm .

Answer to Q7 continued:

Legend for Figures A, B and C. Loss of *Rictor* blocks mitochondrial fission in multiple cell lines.

Representative blots for Rictor protein levels and confocal images and quantification for mitochondrial shape descriptors in (A) *Rictor*^{+/+} and *Rictor*^{-/-} MEFs, or (B) AML12 cells or (C) HepG2 cells transfected with siRNAs against *Rictor* and their corresponding controls and cultured in serum-free medium for 30 min in the presence of MitoTracker green to stain mitochondria. Magnified insets are shown. AML12 = 45 cells; HepG2 = 35-38 cells; MEFs = 36-43 cells from n=3 independent experiments. Individual replicates and means are shown. *P<0.05, **P<0.01 and ****P<0.001, Student's t-test.

8. Improved pictures of BODIPY uptake in the liver should be incorporated, where the hepatocyte structure is differentiated.

Answer: We thank the reviewer for this comment. We employed confocal imaging of freshly isolated liver slices as a positive control to demonstrate the delivery of BODIPY-C₁₆ FL into the liver. It is difficult to obtain high resolution images for BODIPY uptake in tissue slices because the BODIPY signal is lost through the process of fixation, cutting and visualization.

3

9. Authors suggest Cdc42 is recruited to MAMs to promote mitochondrial fission. However, there is no clear information as to where Cdc42 resides on basal state (i.e, cytosol, or another organelle).

Answer: We thank the reviewer for this question. As can be seen in the figure below, in basal conditions (0 h) in wildtype mice, Cdc42 primarily localizes in MAMs fractions followed by the ER and, to a lesser extent, the cytosol when compared to total homogenates (Fig below). Upon fasting, we do not see changes in levels of Cdc42 in the MAMs, ER or cytosol. The ability of Cdc42 to facilitate mitochondrial fission during fasting is likely through binding to phosphorylated NDRG1^{S336} and GTP-loading at MAMs. Indeed, in absence of Rictor or mutation at Ser336 in NDRG1, the abundance of Cdc42 in MAMs is reduced (Fig 7d and 7e in manuscript).

Legend. IB and quantification of Cdc42 in Hom, Mp, MAMs, Cyt and ER fractions from livers of 5-10-month old male mice that were fed or fasted for the indicated durations (h, hours) (n=5-6 mice). Ponceau is loading control. Individual replicates and means are shown. *P<0.05, **P<0.01, ***P<0.001 and ****P<0.0001 vs Hom fraction within each time point, One-way ANOVA followed by Tukey's multiple comparisons test.

10. In line with the previous concern, the role of the ER is never assessed. Authors should evaluate ER localization in the mTORC2-NDRG1-Cdc42 complex and analyze whether the number of MAMs change in *Rictor*KO livers.

Answer: We thank the reviewer for this comment. Our analysis of ER fractions shows that ER lipids, ER stress or ER volume do not change in *Rictor*^{KO} livers when compared to controls. We now show new immunoblot and TEM data revealing reduced levels of the ER-shaping protein Reticulon 4 in MAMs (Fig A) and reduced MAM-ER contacts (Fig B) in *Rictor*^{KO} livers when compared to controls. It is well-established that mitochondrial fission is mediated by the ER, and its likely that ER plays a biophysical role in driving fission downstream in our model; however, investigations into any direct contributions of the ER in the context of mTORC2 signaling are beyond our means and the scope of this manuscript. On the basis of multiple phosphoproteomic analysis, NDRG1-Cdc42 pathway appears to be critical for mitochondrial fission during fasting, and this pathway has thus remained the focus of our investigation in this manuscript.

Legend for Fig A. IB for the indicated ER-shaping proteins in Hom, Mp, MAMs, Cyt and ER from livers of 14-16 h-fasted 5-6 mo-old control and *Rictor*^{KO} (n=3-5 mice). Quantification for Reticulon 4 is shown. Ponceau is loading control. Individual replicates and means are shown. *P<0.05, unpaired Student's t-test.

Using established methods (*Science*. 2011;334(6054):358-62), we have quantified the ER-mitochondrial interactions, and our data reveals reduced number of ER-mitochondria contact sites in *Rictor*^{KO} livers during fasting when compared to controls, thus confirming fission failure in *Rictor*^{KO} livers (Fig. B).

Legend for Fig B. TEMs and quantification for % of mitochondria-ER contacts in Con and *Rictor*^{KO} livers from fed or 14-16 h fasted 4-9 mo-old male mice (n=4-9 mice). A Mitochondria-ER distance <30 nm is a contact site (*Front Cell Dev Biol*. 2020;8:428). Red arrowheads depict contact sites. Individual replicates and means are shown.

1

11. In Extended Data Fig.11 (d), the NDRG1-mediated mitochondrial fission is not visualized clearly. Thus, authors should provide images where the mitochondrial tubule is distinctly fragmented by NDRG1

Answer: We thank the reviewer for this comment. We have included in the revised manuscript images of greater resolution to demonstrate NDRG1-mediated mitochondrial fission. The new **Figure panel** looks as follows:

Legend. Fig. 5 | Mitochondrial fission through the mTORC2-NDRG1^{Ser336} axis occurs independently of Drp1. (a) Live cell imaging of mCherry-NDRG1^{WT} or mCherry-NDRG1^{Ser336Ala} and MitoTracker green in siCon NIH3T3 cells or live cell imaging of mCherry-NDRG1^{WT} and MitoTracker in siRictor or siDnm1l NIH3T3 cells cultured in serum-free medium for 30 min. Orange arrowheads: NDRG1. White arrowheads: NDRG1-mitochondria contact prior to fission. Yellow arrowheads: divided mitochondria after scission by NDRG1. Magnified insets are shown. Please refer to Video 6 (NDRG1^{WT}/MitoTracker), Video 9 (NDRG1^{Ser336Ala}/MitoTracker), Video 10 (siRictor; NDRG1^{WT}/MitoTracker) and Video 11 (siDnm1l; NDRG1^{WT}/MitoTracker).

Reviewer 4 comments

Referee #4 (Remarks to the Author):

Using an unbiased phosphoproteomic approach in liver of mice that were fed, fasted, fasted and refed, or fed with specific lipids, the authors reveal mTORC2 activation upon fasting. mTORC2 activation is accompanied by increased mitochondrial fission. The comparison of the phosphoproteome of wildtype and rictor KO livers led to the identification of NDRG1, a known mTORC2 target, which is detected at ER-mito-contact sites and whose phosphorylation (specifically on S336) affected mitochondrial OXPHOS functions and fission. Unexpectedly, NDRG1-induced fission was not affected by downregulation of Dnm1l. Rather, NDRG1 interaction studies identify Cdc42, which regulates the actin cytoskeleton, as binding partner of NDRG1. Further experiments show that Cdc42 accumulates at ER-Mito-contact sites in a NDRG1-dependent manner, that some interaction partners of Cdc42 are present at these sites as well, and that Cdc42 as NDRG1 or Rictor is required for mitochondrial fission during fasting.

This is a comprehensive analysis of the effect of fasting on mitochondrial dynamics in liver. The authors provide compelling evidence for the localization of NDRG1 at fission sites as well as the importance of its site-specific phosphorylation and regulation by mTORC2 under fasting. These findings are interesting and shed new light as to how metabolic cues affect mitochondrial dynamics. However, I am less convinced about the proposed mechanisms of mitochondrial fission. **The authors examine the role of the mitochondrial fission machinery by downregulating Dnm1l and failed to inhibit NDRG-driven fission. I did not find information about the efficiency of downregulation in these experiments, and, in any case, it is difficult to exclude a role of Dnm1l solely based on these experiments.** Other mitochondrial membrane processes were originally considered Dnm1l-independent based on knock-down experiments but were later shown to be mediated by Dnm1l using knockout cells.

Accordingly, **testing the hypothesis in Dnm1l knockout cells and examining the role of Dnm1l adaptors such as MFF appears obligatory.** On the other hand, the effect of Cdc42 remains largely speculative. The authors provide convincing evidence for NDRG1 binding to Cdc42, the presence of Cdc42 (and of its regulators) at contact sites, but **how Cdc42 affects fission remains speculative.** I agree with the statement of the authors at the end of the manuscript that the effect of Cdc42 on mitochondrial fission is likely indirect reflecting its known effects on the actin cytoskeleton, which is known to be required for Dnm1l-mediated mitochondrial fission. Accordingly, mTORC2 activation upon fasting may affect mitochondrial fission regulating the actin cytoskeleton in a NDRG1 dependent manner.

Answer: We thank the reviewer for this comment. We have now provided evidence of **knockdown of *Dnm1l* gene by ~90 % in the experiments where cells are expressing NDRG1^{WT} and elsewhere where we silenced *Dnm1l* in NIH3T3 cells and liver.** One representative knockdown of Drp1 is show below. Also, consistent with ~90 % loss of Drp1 protein, we observed decreases in mitochondrial number and increases in mitochondrial area, length and perimeter (**Fig. 6f**), reflecting marked loss of fission in our *Dnm1l* knocked-down cells. These mitochondrial phenotypes were also observed in livers upon Invivofectamine-mediated *Dnm1l* silencing (**Fig. 3a, Extended data Fig. 5b, 5d**). These data support genetic and functional silencing of Drp1 in vitro and in vivo systems used across this manuscript. In relation to *Dnm1l* silencing in cells expressing NDRG1^{WT}, **we have now new data showing that expressing NDRG1^{WT} in *Dnm1l*-depleted cells results in complete restoration of mitochondrial fission, supporting our idea that NDRG1-mediated mitochondrial fission can occur independently of Drp1.** Details of this experiment are shown in new figure panel (please see page 4 of this document).

Legend. Representative IB and quantification for Drp1 protein levels in NIH3T3 cells transfected with siRNAs against *Dnm1l*. Scramble siRNA (siCon) was used as control. Ponceau is loading control. (n=3). ****P<0.0001, unpaired student's t-test.

Answer continued:

Legend. (a) Experimental plan and (b) representative IB and quantification for indicated proteins in livers of 4-9 mo-old male mice injected with siCon, siMff or siDnm1l and fasted for 14-16 h (n=3-4). ***P<0.001 and ****P<0.0001 are versus siCon, unpaired student's t-test.

Answer continued:

3

We absolutely agree that DRP1 plays a central role in mitochondrial fission. **In fact, we have used knock downs of *Dnm1l* and *Mff* as positive controls in vitro (Fig. 6f) and in vivo (Fig. 3a, Extended Data 5a-d) to demonstrate that phenotypes obtained upon depletion of *Rictor*, *NdrG1* or *Cdc42* phenocopy the established fission failure phenotype observed in cells knocked-down for *Dnm1l* or *Mff*.** Although our data show that under conditions of *Dnm1l* downregulation NDRG1^{WT} is still able to divide mitochondria using live cell imaging (Fig. 5a), we have now formally tested if reconstitution of NDRG1^{WT} under conditions of *Dnm1l* loss (Fig. A) leads to recovery of mitochondrial fission reflected by increased mitochondrial number and decreased size. Hence, we have now performed experiments where we have expressed mCherry-NDRG1^{WT} in *Dnm1l*-deficient cells, and our data (Fig. B) suggest that **the overexpression of NDRG1^{WT} in *Dnm1l*-depleted cells reverses the mitochondrial fission failure phenotype observed when cells are deficient in *Dnm1l* alone**, as indicated by increased mitochondrial number, decreased size (area, perimeter and elongation), and increased circularity in si*Dnm1l* + NDRG1^{WT} compared to si*Dnm1l* cells (Fig. B). Accordingly, we stand by our interpretation that **the mTORC2-NDRG1-Cdc42 axis is DRP1-independent**.

Legend for Figures A and B. **A.** Representative blots and quantification for Drp1 and mCherry in NIH3T3 cells transfected with siRNAs against *Dnm1l* and transfected or not with mCherry-NDRG1^{WT} plasmid (n=3). Ponceau is loading control. **B.** Representative confocal images of NIH3T3 cells knocked-down for *Dnm1l* expressing or not mCherry-NDRG1^{WT} DNA and cultured in serum-free medium for 30 min in presence of MitoTracker green to stain mitochondria. Magnified insets are shown. Quantification for mitochondrial number and mitochondrial shape descriptors is shown (n=4 independent experiments). Individual replicates and means are shown. *P<0.05, **P<0.01 and ***P<0.001, One-way ANOVA followed by Tukey's multiple comparisons test.

Answer continued:

3

By contrast, reconstitution of NDRG1^{WT} under conditions of *Mff* loss (Fig. A) fails to recover mitochondrial fission as reflected by the inability of NDRG1^{WT} to increase mitochondrial number and decrease mitochondrial size (area, perimeter and elongation) in *Mff*-deficient cells (Fig. B), suggesting that the mTORC2-NDRG1-Cdc42 axis drives mitochondrial fission in an MFF-dependent manner but independent of DRP1.

Legend for Figures A and B. **A.** Representative blots and quantification for MFF and mCherry in NIH3T3 cells transfected with siRNAs against *Mff* and transfected or not with mCherry-NDRG1^{WT} plasmid (n=3). Ponceau is loading control. **B.** Representative confocal images of NIH3T3 cells knocked-down for *Mff* expressing or not mCherry-NDRG1^{WT} DNA and cultured in serum-free medium for 30 min in presence of MitoTracker green to stain mitochondria. Magnified insets are shown. Quantification for mitochondrial number and mitochondrial shape descriptors is shown (n=3 independent experiments). Individual replicates and means are shown. *P<0.05, **P<0.01, ***P<0.001 and ****P<0.0001, One-way ANOVA followed by Tukey's multiple comparisons test.

Additional points:

1

1. The authors provide only correlative evidence that mTORC2 drives mitochondrial fission by phosphorylating NDRG1. To demonstrate the dependence of mTORC2-driven mitochondrial fragmentation on NDRG1-S336 phosphorylation, the authors should examine mitochondrial fission in delta-ric1 cells expressing NDRG1 and its variant.

Answer: We thank the reviewer for this comment. We have already performed this experiment and quantified mitochondria cutting events by NDRG1^{WT} in cells knocked down for *Rictor*, and our data clearly show that **NDRG1^{WT} fails to divide mitochondria in cells knocked down for *Rictor* when compared to siControl cells.** These data can be found in **Fig. 5a**. Furthermore, mitochondrial division is substantially dampened when cells express NDRG1^{Ser336Ala} mutant *in vitro* (**Fig. 5a**) and *in vivo* (**Fig. 3a**).

1

2. Although affected by NDRG1 phosphorylation at S336, the binding efficiency of NDRG1 for Cdc42 is very low. Whether NDRG1 exerts its effect on mitochondrial fission via Cdc42 at mitochondria is not unambiguously demonstrated. It therefore appears conceivable that the effects of Cdc42 depletion reflect its general role for the maintenance of the actin cytoskeleton.

Answer: We thank the reviewer for this comment. We have performed similar epistasis experiments where **the ability of NDRG1^{WT} to cut mitochondria was significantly reduced when *Cdc42* was silenced (Fig. A next page).** This mechanistically couples Cdc42 to NDRG1 in context of mitochondrial division.

Furthermore, using live cell imaging, **we show that Cdc42^{WT} engages with mitochondria resulting in its division, whereas the Cdc42^{Thr17Asn} mutant, which fails to bind to GTP, was unable to divide mitochondria (Extended Data Fig. 9e-g).** The quantifications of these live cell imaging experiments revealed that, as we noticed in *Rictor* silenced cells (**Fig. 5a**), *Cdc42* silenced cells or *Cdc42^{Thr17Asn}* mutant expressing cells, each showed prolonged and futile interactions of NDRG1^{WT} with mitochondria without cutting events.

Furthermore, silencing *Cdc42* markedly reduced mitochondrial membrane potential (**Fig. 6g**) as observed when *Rictor* or *Ndr1* was silenced (**Fig. 2i, Extended Fig. 7j**). **On the basis of data from these epistasis experiments interrogating the interplay between *Rictor* and NDRG1, and *Cdc42* and NDRG1, we are able to confidently state that mTORC2-NDRG1 engages with *Cdc42* to divide mitochondria.**

The execution of mitochondrial division appears to be dependent on *Cdc42*'s ability to interface with actin because the ring-like structures formed by actin fail to organize around mitochondria when *Cdc42* is silenced (**Fig. 7f**). **Hence, we agree with the reviewer that the effects of *Cdc42* depletion reflect its general role for the maintenance of the actin cytoskeleton, however, our data suggest that NDRG1 directs, perhaps a pool of, *Cdc42* to regulate mitochondrial fission in response to nutritional cues.** In sum, our epistasis experiments demonstrate a role of the mTORC2-NDRG1-*Cdc42* axis in mitochondrial fission.

1 Answer to Q2 continued:

Legend. Live-cell imaging of mCherry-NDRG1^{WT} and MitoTracker green in siCon or siCdc42 NIH3T3 cells. Magnified insets are shown. Red arrowhead: NDRG1 mediating fission. White arrowhead: NDRG1-mitochondrial colocalization before fission. Yellow arrowheads: divided mitochondria after fission. Please refer to Video 14 (siCon cells; NDRG1^{WT}) and Video 15 (siCdc42 cells; NDRG1^{WT}).

(A) Quantification for duration and fate, i.e., fission vs. no fission, of interaction between mCherry-NDRG1^{WT} and mitochondria. A description of this quantitative approach is as follows: We quantified the duration of NDRG1-mitochondrial interaction events and whether each interaction led to mitochondrial fission (useful) or not (futile) as recorded via live cell imaging. The X-axis represents time in seconds—reflecting the duration of contact of NDRG1 with mitochondria; while individual-colored bars in the Y-axis represent the different cells/conditions. The length of each colored bar (when tracked from left to right) represents the time between the initiation of interaction of NDRG1 with mitochondria to the end of this interaction, which may or may not lead to mitochondrial division depending on the cell/condition. In addition to the time taken for mitochondrial fission by NDRG1, we noted if interaction with NDRG1 was **useful** (resulted in fission) or **futile** (no fission occurred). Quantification for mean duration of mcherry-NDRG1/mitochondria (MitoTracker) interaction is shown (n=6-11 cells from 3 independent experiments). Individual replicates and means are shown. *P<0.05. Student's t-test.

3. As a minor point, how to the authors explain mitochondrial marker proteins (such as Mfn1 or MFF) in the ER fraction while ER proteins (such as calreticulin) are missing. This fraction appears to represent a general membrane fraction.

Answer: We thank the reviewer for this comment. **Enrichment of mitochondrial fission markers MFF and Drp1 in the ER fraction has been previously described by *J Cell Biol.* 2017 Dec 4; 216(12): 4123–4139.** As per this paper, “Here, we find that a population of Drp1 oligomers is associated with ER in mammalian cells and is distinct from mitochondrial or peroxisomal Drp1 populations. Subpopulations of Mff and Fis1, which are tail-anchored proteins, also localize to ER”.

We have generated MAM preps using protocols described by *Wieckowski, M., et al (Nat Protoc.* 2009;4(11):1582-90), and our immunoblot analysis of markers for organelle/membrane fraction purity and enrichment, including calreticulin, that we present in the entire manuscript replicate those reported in *Nat Protoc.* 2009;4(11):1582-90 (please see side by side comparison in the figure below). In addition, we have now improved the quality of the immunoblot for calreticulin in **Fig. 3c**, and our data show comparable calreticulin enrichment as observed in *Nat Protoc.* 2009;4(11):1582-90.

Finally, changes in Cdc42 and MFF in response to knocking out *Rictor* or expressing mutant NDRG1 S336A were specifically observed in the MAMs fractions and not in ER or mitochondria. Hence, **we are confident that our fractions do not represent general membrane fractions.**

Pre-Decision Letter:

Dear Rajat,

We now have received all reviews from the original revs on your revision "mTORC2-NDRG1-Cdc42 axis couples fasting to mitochondrial fission" -- we have been discussing the reviewers' comments editorially and apologize for the delay in sending our decision to you.

I am writing because Reviewer #4 shared some persistent concerns that we found important regarding the need to use Drp1 KO cells to make claims related to the dispensability of its activity. I am pasting Rev#4's comments below. At this stage, we were hoping to ask if you would be willing to please send us responses to the comments pasted below, focusing on the Drp1 comment.

To be clear, we are not asking for a revised manuscript/new experiments/new data now. We are simply interested in hearing your thoughts on these points within 1-2 business days if possible - including if you tried these experiments and they could not be done, or if you may have these data, or if you think you would be able to provide minimal experiments along these lines in a reasonable time frame if needed.

This would be extremely informative to us as we continue the editorial process. We might then discuss your response with the reviewers again before reaching our decision editorially.

Please let me know if you have any questions and I look forward to hearing your thoughts on the points below. Thank you so much for your time and consideration,

Best wishes,
Melina

--

Melina Casadio, PhD
Senior Editor, NCB

REVIEWER #4 COMMENTS TO THE AUTHORS

The authors have significantly improved the description of their findings in the revised manuscript and added additional data in support of some of their claims. However, the experiments did not address my main concern on the original version that mitochondrial fission induced along the mTORC2-NDRG1-CDC42 axis occurs in a DRP1/DNML1-independent manner. The authors base this conclusion on knockdown experiments only and did not generate knockout cells to substantiate this conclusion. They show now that overexpression of NDRG1 can suppress fission defects in DRP1- but not in MFF depleted cells and take this as evidence in support of DRP1-independent fission. Although the effects of NDRG1 overexpression on mitochondrial morphology are convincing, I do not agree with the conclusion of the authors that this support a DRP1-independent fission mechanism. As pointed out in my original comments, low amounts of DRP1 were found to promote membrane fission in other systems that were originally described to be DRP1-independent. Moreover, the authors do not provide strong evidence for an alternative fission mechanism and the role of the rho GTPase CDC42 in this process. It remains unclear whether it acts via the actin cytoskeleton, its canonical function, or directly affects the fission process, as the authors discuss. Although I do appreciate the comprehensive

analysis of the effects of fasting on mitochondrial dynamics in liver, I remain less convinced about the mechanistic aspects of this work.

Author rebuttal, pre-decision:

Dear Rajat,

We now have received all reviews from the original revs on your revision "mTORC2-NDRG1-Cdc42 axis couples fasting to mitochondrial fission" -- we have been discussing the reviewers' comments editorially and apologize for the delay in sending our decision to you.

I am writing because Reviewer #4 shared some persistent concerns that we found important regarding the need to use Drp1 KO cells to make claims related to the dispensability of its activity. I am pasting Rev#4's comments below. At this stage, we were hoping to ask if you would be willing to please send us responses to the comments pasted below, focusing on the Drp1 comment.

To be clear, we are not asking for a revised manuscript/new experiments/new data now. We are simply interested in hearing your thoughts on these points within 1-2 business days if possible - including if you tried these experiments and they could not be done, or if you may have these data, or if you think you would be able to provide minimal experiments along these lines in a reasonable time frame if needed.

This would be extremely informative to us as we continue the editorial process. We might then discuss your response with the reviewers again before reaching our decision editorially.

Please let me know if you have any questions and I look forward to hearing your thoughts on the points below. Thank you so much for your time and consideration,

Best wishes,
Melina

--

Melina Casadio, PhD
Senior Editor, NCB

REVIEWER #4 COMMENTS TO THE AUTHORS

The authors have significantly improved the description of their findings in the revised manuscript and added additional data in support of some of their claims. However, the experiments did not address my main concern on the original version that **mitochondrial fission induced along the mTORC2-NDRG1-CDC42 axis occurs in a DRP1/DNML1-**

independent manner. The authors base this conclusion on knockdown experiments only and did not generate knockout cells to substantiate this conclusion. They show now that overexpression of NDRG1 can suppress fission defects in DRP1- but not in MFF depleted cells and take this as evidence in support of DRP1-independent fission. Although the effects of NDRG1 overexpression on mitochondrial morphology are convincing, I do not agree with the conclusion of the authors that this support a DRP1-independent fission mechanism. As pointed out in my original comments, **low amounts of DRP1 were found to promote membrane fission in other systems that were originally described to be DRP1-independent.** Moreover, **the authors do not provide strong evidence for an alternative fission mechanism and the role of the rho GTPase CDC42 in this process.** It remains **unclear whether it acts via the actin cytoskeleton, its canonical function, or directly affects the fission process,** as the authors discuss. Although I do appreciate the comprehensive analysis of the effects of fasting on mitochondrial dynamics in liver, I remain less convinced about the mechanistic aspects of this work.

Answer: We do not agree with Reviewer's concerns that our experiments do not show that mitochondrial fission induced by the mTORC2-NDRG1-CDC42 axis occurs in a DRP1/DNML1-independent manner. We are providing the following points to support our conclusions:

1. First, we have provided strong evidence of knockdown of *Dnm1l* gene by ~90 % in the experiments where we silenced *Dnm1l* in NIH3T3 cells. The **deletion of *Dnm1l* gene results in all the HALLMARK FEATURES of fission failure *in vitro* and *in vivo*.** This is supported by the decreased mitochondrial number, increased mitochondrial area, perimeter and length, and decreased in circularity as observed in *siDnm1l* cells when compared to controls (*siCon*) (Fig. 6f). Consistent with these changes in mitochondrial morphology, *siDnm1l* cells display significant reduction in fission rates (Fig. 3d), thus strongly supporting the notion that **a 90% loss of *Dnm1l* gene is sufficient to impair mitochondrial fission.** In relation to the experiments where we demonstrate the independency of NDRG1 from Drp1 (Fig. 5d, Reviewer 4 comments, page 4), we included data showing that **expressing NDRG1^{WT} in cells depleted of Drp1 restores mitochondrial fission originating from loss of Drp1.** There can only be one conclusion drawn from this outcome: NDRG1 is able to divide mitochondria in cells where there is fission failure due to Drp1 deficiency. We are happy to not use the term "Drp1-independent" but state verbatim that, "*NDRG1 divides mitochondria in cells and normalizes mitochondrial phenotypes in cells exhibiting fission failure due to Drp1 deficiency.*"
2. Importantly, we validated our findings *in vivo* by knocking down *Dnm1l* gene in C57 mice using siRNAs delivered retro-orbitally using *invivolectamine*TM (developed for liver-targeted transfection) (Fig. 3a, Extended Data Fig. 5a-d), where we noted a 71% reduction in Drp1 protein levels in liver. In this model, **our transmission electron microscopy studies revealed HALLMARK FEATURES of fission failure, i.e., decreased mitochondrial number and mitochondrial ballooning in livers of *siDnm1l* mice (Fig. 3a Extended Data**

Fig. d), which are qualitatively and quantitatively identical to what has been previously reported in $Drp1^{KO}$ livers (*Diabetologia*. 2015;58(10):2371-80). **PLEASE NOTE:** If residual Drp1 was sufficient to drive fission, then fission failure phenotypes would not have been noted in our models *in vitro* and *in vivo*. In sum, these data support use of genetic and functional silencing of Drp1 *in vitro* and *in vivo* without the need to generate Drp1 knock-out cells.

3. We find there is no scientific rigor in the reviewer's statement that "low amounts of DRP1 were found to promote membrane *fission in other systems that were originally described to be DRP1-independent*" since the reviewer fails to provide the corresponding literature to support his statement. First of all, it is UNFAIR to compare a manuscript with unrelated outcomes from other labs. Moreover, mitochondrial fission through **Drp1-independent mechanisms has been demonstrated in *Drp1*^{-/-} mouse embryonic fibroblasts (*J Biol Chem*. 2011;286(23):20710-26, *Proc Natl Acad Sci U S A*. 2013;110(40):16003-8)**. As alluded to earlier in point 2: We will state verbatim that, "*NDRG1 divides mitochondria in cells and normalizes mitochondrial phenotypes in cells exhibiting fission failure due to Drp1 deficiency.*"
4. We strongly disagree with the reviewer's logicless comment that "*the authors do not provide strong evidence for an alternative fission mechanism and the role of the rho GTPase CDC42 in this process*". **We can confidently state that mTORC2-NDRG1 engages with Cdc42 to divide mitochondria since we demonstrate that:**
 - a) NDRG1^{WT} fails to cut mitochondria when *Cdc42* is silenced (**Fig. 6c-e**).
 - b) *Cdc42*^{WT} engages with mitochondria resulting in its division, while the *Cdc42*^{Thr17Asn} mutant, which fails to bind to GTP, is unable to divide mitochondria (**Extended Data Fig. 9e-g**).
 - c) *Cdc42* silenced cells or *Cdc42*^{Thr17Asn} mutant expressing cells, each showed prolonged and futile interactions of NDRG1^{WT} with mitochondria without cutting events.
 - d) Extensive characterization of *Cdc42*-interacting partners further validate its role in fission.
5. Finally, the inclusion of the role of actin in mitochondrial division in our studies was simply aimed at providing closure to our proposed model where we are showing a Rictor-NDRG1-Cdc42 axis drives mitochondrial fission. **While a role for actin in mitochondrial division has been already established to regulate mitochondrial fission (*Nat Commun*. 2016;7:12886); besides actin is not focal center of our study.** Due to the known roles for *Cdc42* in actin polymerization, we interrogated if the regulation of *Cdc42* in mitochondrial division was dependent on *Cdc42*'s ability to interface with actin. In support of this hypothesis, we observed that the ring-like structures formed by actin fail to organize around mitochondria when *Cdc42* is silenced (**Fig. 7f**). Hence, we agree with the reviewer that the effects of *Cdc42* depletion may reflect its canonical function in the maintenance of the actin cytoskeleton, while it is also possible that *Cdc42* regulates mitochondrial fission directly by other mechanisms.

We did attempt to generate Drp1KO cells using CRISPR but failed to generate cells that looked healthy enough for experiments, but given the robust outcomes from siRNA use in vitro and in vivo, we felt that a KO system may not be necessary after all. This was also because the **focus of the work is on mTORC2-NDRG1**.

Decision Letter, first revision:

Our ref: NCB-A49199A

24th March 2023

Dear Rajat,

Thank you for submitting your revised manuscript "mTORC2-NDRG1-Cdc42 axis couples fasting to mitochondrial fission" (NCB-A49199A). It has now been seen by the original referees and their comments are below. The reviewers find that the paper has improved in revision, and Reviewer #1 shares suggestions for clarifications and edits regarding the revision data that we find valuable. Reviewer #4, as you know from our correspondence, is not convinced that the study establishes a Drp1-independent mechanism of action. The reviewer also asks about the mechanism by which Cdc42 may impact mitochondrial fission; however, we do not think that further mechanistic insights along these lines are within the scope of this work.

The Drp1 independence is an important part of the study and we appreciate that it has been challenging to establish a KO cell line, and that the evidence does support a potentially Drp1-independent mechanism. We have discussed the reviewer's remarks as well as your responses to them in detail within the editorial team, and find that we can move forward with the study at NCB provided that you please address this remaining caveat with careful text edits. We encourage you to state in the manuscript preferably (or Methods if needed) that your attempts to generate a full KO line were unsuccessful, and therefore this limit in interpretation remains.

Overall, we'll be happy in principle to publish the study in Nature Cell Biology, pending minor revisions to satisfy the referees' final requests and to address the above, and to comply with our editorial and formatting guidelines.

****The current version of your manuscript is in a PDF format. Could you please email us a copy of the file in an editable format (Microsoft Word or LaTeX)? We cannot proceed with PDFs at this stage.****
(Please email us the current Word version, not a revised manuscript -- this is so that we can initiate all our checks)

With the Word file, we will be performing detailed checks on your paper and will send you a checklist detailing our editorial and formatting requirements within 1-2 weeks of reception of the file. Please do not upload the final materials and make any revisions until you receive this additional information from us.

Thank you again for your interest in Nature Cell Biology. Please do not hesitate to contact me if you have any questions.

Sincerely,

Melina

Melina Casadio, PhD
Senior Editor, Nature Cell Biology
ORCID ID: <https://orcid.org/0000-0003-2389-2243>

Reviewer #1 (Remarks to the Author):

The revised manuscript submitted by Martinez-Lopez et al. is vastly improved over the original submission. Most of the suggestions made (by all reviewers) have been acted upon or answered satisfactorily. The impact of the manuscript has been considerably raised by the rewrite and the inclusion of new data.

Below are further points (not in any order of importance), mainly occurring in newly added sections, that should be addressed before publication.

1. References are cited in the summary, which is not usual for Nature Cell Biology.
2. 'Strikingly, fasting for 14 h robustly reactivated mTORC1 and mTORC2 signaling in liver, indicated by phosphorylation of their respective targets³, P70Thr389 (mTORC1), and AKTSer473, SGK1Thr256 and NDRG1Thr346 (mTORC2) (Fig 1h).'
- Please temper this sentence. The signal for P-P70Thr389 is not clear- which band is the arrow indicating? The weaker signal just under the arrow line is coming back at 14 h but is still less than at 0 h? Increases in P-AKTSer473 and P-SGK1Thr256 are visible. P-NDRG1Thr346 has actually reduced at 14 h although is notable at 20 h. Statistical analysis suggests that although many mice were used, the data at 14 h is only just significantly different for these substrates and has to be compared to 3 h in some cases to achieve significance. Overall the data is, at best, suggestive of reactivated MTORC1/2 signalling.
3. '(Extended Data Fig. 4c), reflecting increased fission during fasting.' Please comment on any (possible) role for smaller mitochondria during fasting.
4. Figure 3b: it is not clear in the quantification which data are for fed and which for fasted mice. Please change key and add colours/shades to indicate this.
5. New sections added on page 9 and all throughout the remains of the manuscript cite 'data not shown'. To my knowledge this is not allowed in Nature Cell Biology, nor should it be. If data is necessary for the assessment of the work presented, then it has to be included in extended data. If this results in too many extended data figures, then a judgement has to be made on what to exclude. In some instances, e.g. 'and increased circularity (data not shown), as also described previously⁴⁵

the 'data not shown' is not required if it has already been published.

6. p13: 'To that purpose, we used proteomics to identify proteins bound to FLAG-tagged NDRG1WT but not NDRG1Ser336Ala.' Explain the methods involved in 'proteomics'. Was this just co-IP and mass spec? An explanation akin to that in the next section and in Fig 7a would be useful. This may also need to be added to the Method section

7. p13: 'Indeed, when compared to NDRG1WT, NDRG1Ser336Ala displayed modestly reduced binding to Cdc42, Arhgef10 (a Rho GEF that activates Rho GTPases by stimulating GDP/GTP exchange)⁴⁹, and Arhgap35 (Rho GAP that facilitates GTP hydrolysis to inactivate Rho GTPases)⁴⁹ likely due to the transient and dynamic nature of these interactions.'

Please clarify what is meant here. What is the significance of the interactions being transient and dynamic?

8. 'and mutant Cdc42Thr17Asn, which does not bind GTP (Extended Data Fig. 9c)'. Please edit this statement. T17N mutants form unproductive complexes with GEFs but can bind GTP. A statement such as 'expected to be predominantly GDP bound' would be more accurate.

9. 'Rho GDP dissociation inhibitor (RhoGDI) proteins are negative regulators of Rho GTPases by retaining Rho GTPases in the cytosol, inhibiting their GTPase activity, and preventing their interaction with GEFs, GAPs and effectors⁵¹.' Please edit this sentence. The reference is quite old and GDI proteins have been well characterized as also acting as chaperones and therefore acting as positive regulators by correctly localizing small G proteins to the correct membrane and protecting them from degradation.

10. 'Because a number of GTPases interact with Cdc42, it is interesting that not all Rho GTPases impact mitochondrial dynamics, since depletion of Rho GTPase family Rhobtb1 gene had no effect on mitochondrial morphology, while silencing IQGAP1 increased mitochondrial number (Fig 7c).'

Please clarify this statement. The data in Fig 7 does not show that a number of GTPases interact with Cdc42, only RhoBTB1. Why is it 'interesting' that not all Rho GTPases impact mitochondrial dynamics? It seems unlikely that they would. IQGAP1 is not a GTPase and yet this is implied in this sentence.

11. There are multiple incidences of incorrect unit nomenclature in the methods: ml, µl and ug

Reviewer #2 (Remarks to the Author):

This is possibly the most insanely great response document of all time. Forty six pages of beautifully formatted experiments carefully addressing every single comment of four reviewers. I hope the reviewers opt in to publishing this review document.

I am fully satisfied that the revised manuscript addresses all of my comments and, I think, those of my colleagues. I compliment the authors on their detailed and fascinating findings.

Reviewer #3 (Remarks to the Author):

This revision represents a substantially improved and at times toned down paper, which addressed most of my original concerns. Aspects of my initial review that the authors did not do as deemed outside of the scope would have made the manuscript stronger but I do not see as deal breakers given the revision of the language.

I recommend for publication

Reviewer #4 (Remarks to the Author):

The authors have significantly improved the description of their findings in the revised manuscript and added additional data in support of some of their claims. However, the experiments did not address my main concern on the original version that mitochondrial fission induced along the mTORC2-NDRG1-CDC42 axis occurs in a DRP1/DNML1-independent manner. The authors base this conclusion on knockdown experiments only and did not generate knockout cells to substantiate this conclusion. They show now that overexpression of NDRG1 can suppress fission defects in DRP1- but not in MFF depleted cells and take this as evidence in support of DRP1-independent fission. Although the effects of NDRG1 overexpression on mitochondrial morphology are convincing, I do not agree with the conclusion of the authors that this support a DRP1-independent fission mechanism. As pointed out in my original comments, low amounts of DRP1 were found to promote membrane fission in other systems that were originally described to be DRP1-independent. Moreover, the authors do not provide strong evidence for an alternative fission mechanism and the role of the rho GTPase CDC42 in this process. It remains unclear whether it acts via the actin cytoskeleton, its canonical function, or directly affects the fission process, as the authors discuss. Although I do appreciate the comprehensive analysis of the effects of fasting on mitochondrial dynamics in liver, I remain less convinced about the mechanistic aspects of this work.

Decision Letter, final checks:

Our ref: NCB-A49199A

4th April 2023

Dear Dr. Singh,

Thank you for your patience as we've prepared the guidelines for final submission of your Nature Cell Biology manuscript, "mTORC2-NDRG1-Cdc42 axis couples fasting to mitochondrial fission" (NCB-A49199A). Please carefully follow the step-by-step instructions provided in the attached file, and add a response in each row of the table to indicate the changes that you have made. Please also check and comment on any additional marked-up edits we have proposed within the text. Ensuring that each point is addressed will help to ensure that your revised manuscript can be swiftly handed over to our production team.

In recognition of the time and expertise our reviewers provide to Nature Cell Biology's editorial process, we would like to formally acknowledge their contribution to the external peer review of your manuscript entitled "mTORC2-NDRG1-Cdc42 axis couples fasting to mitochondrial fission". For those reviewers who give their assent, we will be publishing their names alongside the published article.

Nature Cell Biology offers a Transparent Peer Review option for new original research manuscripts submitted after December 1st, 2019. As part of this initiative, we encourage our authors to support increased transparency into the peer review process by agreeing to have the reviewer comments, author rebuttal letters, and editorial decision letters published as a Supplementary item. When you submit your final files please clearly state in your cover letter whether or not you would like to participate in this initiative. Please note that failure to state your preference will result in delays in accepting your manuscript for publication.

Cover suggestions

As you prepare your final files we encourage you to consider whether you have any images or illustrations that may be appropriate for use on the cover of Nature Cell Biology.

Nature Cell Biology has now transitioned to a unified Rights Collection system which will allow our Author Services team to quickly and easily collect the rights and permissions required to publish your work. Approximately 10 days after your paper is formally accepted, you will receive an email in providing you with a link to complete the grant of rights. If your paper is eligible for Open Access, our Author Services team will also be in touch regarding any additional information that may be required to arrange payment for your article.

Please note that *Nature Cell Biology* is a Transformative Journal (TJ). Authors may publish their

research with us through the traditional subscription access route or make their paper immediately open access through payment of an article-processing charge (APC). Authors will not be required to make a final decision about access to their article until it has been accepted. Find out more about Transformative Journals

Please use the following link for uploading these materials:
[Redacted]

Best regards,

Kendra Donahue
Staff
Nature Cell Biology

On behalf of

Melina Casadio, PhD
Senior Editor, Nature Cell Biology
ORCID ID: <https://orcid.org/0000-0003-2389-2243>

Reviewer #1:

Remarks to the Author:

The revised manuscript submitted by Martinez-Lopez et al. is vastly improved over the original submission. Most of the suggestions made (by all reviewers) have been acted upon or answered satisfactorily. The impact of the manuscript has been considerably raised by the rewrite and the inclusion of new data.

Below are further points (not in any order of importance), mainly occurring in newly added sections, that should be addressed before publication.

1. References are cited in the summary, which is not usual for Nature Cell Biology.
2. 'Strikingly, fasting for 14 h robustly reactivated mTORC1 and mTORC2 signaling in liver, indicated by phosphorylation of their respective targets³, P70Thr389 (mTORC1), and AKTSer473, SGK1Thr256 and NDRG1Thr346 (mTORC2) (Fig 1h).'

Please temper this sentence. The signal for P-P70Thr389 is not clear- which band is the arrow indicating? The weaker signal just under the arrow line is coming back at 14 h but is still less than at 0 h? Increases in P-AKTSer473 and P-SGK1Thr256 are visible. P-NDRG1Thr346 has actually reduced at 14 h although is notable at 20 h. Statistical analysis suggests that although many mice were used, the data at 14 h is only just significantly different for these substrates and has to be compared to 3 h in some cases to achieve significance. Overall the data is, at best, suggestive of reactivated MTORC1/2 signalling.

3. '(Extended Data Fig. 4c), reflecting increased fission during fasting.' Please comment on any (possible) role for smaller mitochondria during fasting.
4. Figure 3b: it is not clear in the quantification which data are for fed and which for fasted mice. Please change key and add colours/shades to indicate this.
5. New sections added on page 9 and all throughout the remains of the manuscript cite 'data not shown'. To my knowledge this is not allowed in Nature Cell Biology, nor should it be. If data is necessary for the assessment of the work presented, then it has to be included in extended data. If this results in too many extended data figures, then a judgement has to be made on what to exclude. In some instances, e.g. 'and increased circularity (data not shown), as also described previously⁴⁵' the 'data not shown' is not required if it has already been published.
6. p13: 'To that purpose, we used proteomics to identify proteins bound to FLAG-tagged NDRG1WT but not NDRG1Ser336Ala.' Explain the methods involved in 'proteomics'. Was this just co-IP and mass spec? An explanation akin to that in the next section and in Fig 7a would be useful. This may also need to be added to the Method section
7. p13: 'Indeed, when compared to NDRG1WT, NDRG1Ser336Ala displayed modestly reduced binding to Cdc42, Arhgef10 (a Rho GEF that activates Rho GTPases by stimulating GDP/GTP exchange)⁴⁹, and Arhgap35 (Rho GAP that facilitates GTP hydrolysis to inactivate Rho GTPases)⁴⁹ likely due to the transient and dynamic nature of these interactions.'

Please clarify what is meant here. What is the significance of the interactions being transient and dynamic?

8. 'and mutant Cdc42Thr17Asn, which does not bind GTP (Extended Data Fig. 9c)'. Please edit this statement. T17N mutants form unproductive complexes with GEFs but can bind GTP. A statement such as 'expected to be predominantly GDP bound' would be more accurate.
9. 'Rho GDP dissociation inhibitor (RhoGDI) proteins are negative regulators of Rho GTPases by

retaining Rho GTPases in the cytosol, inhibiting their GTPase activity, and preventing their interaction with GEFs, GAPs and effectors⁵¹. Please edit this sentence. The reference is quite old and GDI proteins have been well characterized as also acting as chaperones and therefore acting as positive regulators by correctly localizing small G proteins to the correct membrane and protecting them from degradation.

10. 'Because a number of GTPases interact with Cdc42, it is interesting that not all Rho GTPases impact mitochondrial dynamics, since depletion of Rho GTPase family Rhobtb1 gene had no effect on mitochondrial morphology, while silencing IQGAP1 increased mitochondrial number (Fig 7c).' Please clarify this statement. The data in Fig 7 does not show that a number of GTPases interact with Cdc42, only RhoBTB1. Why is it 'interesting' that not all Rho GTPases impact mitochondrial dynamics? It seems unlikely that they would. IQGAP1 is not a GTPase and yet this is implied in this sentence.

11. There are multiple incidences of incorrect unit nomenclature in the methods: ml, µl and ug

Reviewer #2:

Remarks to the Author:

This is possibly the most insanely great response document of all time. Forty six pages of beautifully formatted experiments carefully addressing every single comment of four reviewers. I hope the reviewers opt in to publishing this review document.

I am fully satisfied that the revised manuscript addresses all of my comments and, I think, those of my colleagues. I compliment the authors on their detailed and fascinating findings.

Reviewer #3:

Remarks to the Author:

This revision represents a substantially improved and at times toned down paper, which addressed most of my original concerns. Aspects of my initial review that the authors did not do as deemed outside of the scope would have made the manuscript stronger but I do not see as deal breakers given the revision of the language.

I recommend for publication

Reviewer #4:

Remarks to the Author:

The authors have significantly improved the description of their findings in the revised manuscript and added additional data in support of some of their claims. However, the experiments did not address my main concern on the original version that mitochondrial fission induced along the mTORC2-NDRG1-CDC42 axis occurs in a DRP1/DNML1-independent manner. The authors base this conclusion on knockdown experiments only and did not generate knockout cells to substantiate this conclusion. They show now that overexpression of NDRG1 can suppress fission defects in DRP1- but not in MFF depleted cells and take this as evidence in support of DRP1-independent fission. Although the effects

of NDRG1 overexpression on mitochondrial morphology are convincing, I do not agree with the conclusion of the authors that this support a DRP1-independent fission mechanism. As pointed out in my original comments, low amounts of DRP1 were found to promote membrane fission in other systems that were originally described to be DRP1-independent. Moreover, the authors do not provide strong evidence for an alternative fission mechanism and the role of the rho GTPase CDC42 in this process. It remains unclear whether it acts via the actin cytoskeleton, its canonical function, or directly affects the fission process, as the authors discuss. Although I do appreciate the comprehensive analysis of the effects of fasting on mitochondrial dynamics in liver, I remain less convinced about the mechanistic aspects of this work.

Author Rebuttal, second revision:

Dear Rajat,

Thank you for submitting your revised manuscript "mTORC2-NDRG1-Cdc42 axis couples fasting to mitochondrial fission" (NCB-A49199A). It has now been seen by the original referees and their comments are below. The reviewers find that the paper has improved in revision, and Reviewer #1 shares suggestions for clarifications and edits regarding the revision data that we find valuable. Reviewer #4, as you know from our correspondence, is not convinced that the study establishes a Drp1-independent mechanism of action. The reviewer also asks about the mechanism by which Cdc42 may impact mitochondrial fission; however, we do not think that further mechanistic insights along these lines are within the scope of this work.

The Drp1 independence is an important part of the study and we appreciate that it has been challenging to establish a KO cell line, and that the evidence does support a potentially Drp1-independent mechanism. We have discussed the reviewer's remarks as well as your responses to them in detail within the editorial team, and find that **we can move forward with the study at NCB provided that you please address this remaining caveat with careful text edits. We encourage you to state in the manuscript preferably (or Methods if needed) that your attempts to generate a full KO line were unsuccessful, and therefore this limit in interpretation remains.**

Overall, we'll be happy in principle to publish the study in Nature Cell Biology, pending minor revisions to satisfy the referees' final requests and to address the above, and to comply with our editorial and formatting guidelines.

The current version of your manuscript is in a PDF format. Could you please email us a copy of the file in an editable format (Microsoft Word or LaTeX)? We cannot proceed with PDFs at this stage (Please email us the current Word version, not a revised manuscript -- this is so that we can initiate all our checks)

With the Word file, we will be performing detailed checks on your paper and will send you a checklist detailing our editorial and formatting requirements within 1-2 weeks of reception of the file. Please do not upload the final materials and make any revisions until you receive this additional information from us.

Thank you again for your interest in Nature Cell Biology. Please do not hesitate to contact me if you have any questions.

Sincerely,

Melina

Melina Casadio, PhD

Senior Editor, Nature Cell Biology

ORCID ID: <https://orcid.org/0000-0003-2389-2243>

Reviewer #1 (Remarks to the Author):

The revised manuscript submitted by Martinez-Lopez et al. is vastly improved over the original submission. Most of the suggestions made (by all reviewers) have been acted upon or answered satisfactorily. The impact of the manuscript has been considerably raised by the rewrite and the inclusion of new data.

Below are further points (not in any order of importance), mainly occurring in newly added sections, that should be addressed before publication.

1. References are cited in the summary, which is not usual for Nature Cell Biology.

Answer: We thank the Reviewer for bringing this to our attention. All references have now been removed from the summary, and incorporated into the Introduction and/or Results section.

2. ‘Strikingly, fasting for 14 h robustly reactivated mTORC1 and mTORC2 signaling in liver, indicated by phosphorylation of their respective targets³, P70Thr389 (mTORC1), and AKTSer473, SGK1Thr256 and NDRG1Thr346 (mTORC2) (Fig 1h).’

Please temper this sentence. The signal for P-P70Thr389 is not clear- which band is the arrow indicating? The weaker signal just under the arrow line is coming back at 14 h but is still less than at 0 h? Increases in P-AKTSer473 and P-SGK1Thr256 are visible. P-NDRG1Thr346 has actually reduced at 14 h although is notable at 20 h. Statistical analysis suggests that although many mice were used, the data at 14 h is only just significantly different for these substrates and has to be compared to 3 h in some cases to achieve significance. Overall the data is, at best, suggestive of reactivated MTORC1/2 signalling.

Answer: As suggested by the Reviewer, we have tempered this sentence by removing the word “robustly” in describing mTORC1/mTORC2 reactivation. In addition, we have added the sentence: *“Our data suggest that fasting for 14 h reactivates mTORC1/C2 signaling in liver, indicated by phosphorylation of their respective targets²³,...”*

We apologize to the reviewer for the shift in the arrow depicting the P-P70^{Thr389} band in Fig. 1h while preparing the figures. As shown in the Supplementary Image Source Data File 1 containing the uncropped full-length immunoblot membranes (**please see next page of this document**), the arrow points to the band of P-P70^{Thr389} that comes back up at 14 h of fasting (**Fig A**). This is the same band that increases in livers of mice gavaged with corn oil or refed a high fat diet when compared to controls (**Fig B**). As a reference, **Fig C** depicts immunoblotting data from *Cell Signaling Technology* showing the P-P70^{Thr389} band using the same Rabbit antibody used in our manuscript.

A Fig 1h. P-P70^{Thr389}

B Extended Data Fig. 1e. P-P70^{Thr389}

Legend for Figures A and B. Uncropped full-length picture of immunoblot membranes for P-P70^{Thr389} in **A**) Fig 1h and **B**) Extended Data Fig. 1e of the manuscript. Arrow depicts P-P70^{Thr389} band.

C Phospho-p70 S6 Kinase (Thr389) (108D2) Rabbit mAb #9234
Cell signaling Technology

Legend for Figure C. Immunoblot for P-P70^{Thr389} levels in the indicated cell lines using Phospho-p70 S6 Kinase (Thr389) (108D2) Rabbit antibody (Cell signaling Technology, #9234).

3. '(Extended Data Fig. 4c), reflecting increased fission during fasting.' Please comment on any (possible) role for smaller mitochondria during fasting.

Answer: We thank the Reviewer for bringing this to our attention. We have added a comment on a possible role for smaller mitochondria during fasting. The sentence in the text now reads, "...we demonstrate that paradoxical reactivation of mTORC2 during fasting is required for mitochondrial

remodeling to possibly support the increased energetic demands of fasting. In fact, enzymatic reactions, for example, those part of the Krebs cycle, appear to be sensitive to changes in mitochondrial shape, volume and connectivity⁶¹. Consistently, not only does loss of Rictor impact mitochondrial fission, we also noted marked accumulation of acylcarnitines, a metabolic signature consistent with dampened mitochondrial respiration.”

4. Figure 3b: it is not clear in the quantification which data are for fed and which for fasted mice. Please change key and add colours/shades to indicate this.

Answer: We thank the Reviewer for bringing this to our attention. The Fed and Fasted groups have now been clearly indicated in Fig 3b. We apologize for failing to include the identity of the groups in the first place.

5. New sections added on page 9 and all throughout the remains of the manuscript cite ‘data not shown’. To my knowledge this is not allowed in Nature Cell Biology, nor should it be. If data is necessary for the assessment of the work presented, then it has to be included in extended data. If this results in too many extended data figures, then a judgement has to be made on what to exclude. In some instances, e.g. ‘and increased circularity (data not shown), as also described previously⁴⁵’ the ‘data not shown’ is not required if it has already been published.

Answer: We thank the Reviewer for bringing this point to our attention. We agree with the Reviewer on this point, and accordingly, we have now excluded the term, “data not shown” for instances where data were not included in the manuscript. These specifically pertain to: **(a)** data showing that loss of *Rab4a* induces fission (opposite to what would be expected if RAB4A is mediating the effect of NDRG1 on fission); **(b)** data showing that silencing *Cdc42ep2* severely reduces viability; and **(c)** data showing that loss of *Tsc1* induces fission (opposite to what would be expected if mTORC1 is mediating the effect observed in our model). We eliminated these data since these were not directly relevant to our finding.

6. p13: ‘To that purpose, we used proteomics to identify proteins bound to FLAG-tagged NDRG1WT but not NDRG1Ser336Ala.’ Explain the methods involved in ‘proteomics’. Was

this just co-IP and mass spec? An explanation akin to that in the next section and in Fig 7a would be useful. This may also need to be added to the Method section.

Answer: We have now included in detail in the methods section that proteomics was performed on pulldowns of FLAG-tagged WT-NDRG1 and mutant NDRG1^{Ser336Ala} using S-trap columns followed by Nanoscale liquid chromatography-mass spectrometry (nLC-MS/MS).

7. p13: ‘Indeed, when compared to NDRG1WT, NDRG1Ser336Ala displayed modestly reduced binding to Cdc42, Arhgef10 (a Rho GEF that activates Rho GTPases by stimulating GDP/GTP exchange)⁴⁹, and Arhgap35 (Rho GAP that facilitates GTP hydrolysis to inactivate Rho GTPases)⁴⁹ likely due to the transient and dynamic nature of these interactions.’ Please clarify what is meant here. What is the significance of the interactions being transient and dynamic?

Answer: We have modified this sentence for clarity; and the sentence now reads: “Indeed, when compared to NDRG1^{WT}, NDRG1^{Ser336Ala} displayed reduced, albeit modest, binding to CDC42, ARHGEF10 (RHO GEF that activates RHO GTPases by stimulating GDP/GTP exchange)⁴¹, and ARHGAP35 (RHO GAP that facilitates GTP hydrolysis to inactivate RHO GTPases)⁴¹”

8. ‘and mutant Cdc42Thr17Asn, which does not bind GTP (Extended Data Fig. 9c)’. Please edit this statement. T17N mutants form unproductive complexes with GEFs but can bind GTP. A statement such as ‘expected to be predominantly GDP bound’ would be more accurate.

Answer: We thank the reviewer for this comment. The sentence now reads: “Furthermore, FLAG-NDRG1^{WT} interacts with both mCherry-CDC42^{WT} and mutant CDC42^{Thr17Asn}, which is expected to be predominantly GDP-bound (**Extended Data Fig. 9c**), indicating that CDC42-GTP loading is not required for CDC42-NDRG1 interaction.”

9. ‘Rho GDP dissociation inhibitor (RhoGDI) proteins are negative regulators of Rho GTPases by retaining Rho GTPases in the cytosol, inhibiting their GTPase activity, and

preventing their interaction with GEFs, GAPs and effectors⁵¹.' Please edit this sentence. The reference is quite old and GDI proteins have been well characterized as also acting as chaperones and therefore acting as positive regulators by correctly localizing small G proteins to the correct membrane and protecting them from degradation.

Answer: We have now edited this sentence and included a more recent reference to support the same. The sentence now reads: "*Rho GDP dissociation inhibitor (RhoGDI) proteins can act as negative regulators of Rho GTPases by retaining Rho GTPases in the cytosol, inhibiting their GTPase activity, and preventing their interaction with GEFs, GAPs and effectors*^{43,44}."

10. 'Because a number of GTPases interact with Cdc42, it is interesting that not all Rho GTPases impact mitochondrial dynamics, since depletion of Rho GTPase family Rhobtb1 gene had no effect on mitochondrial morphology, while silencing IQGAP1 increased mitochondrial number (Fig 7c).' Please clarify this statement. The data in Fig 7 does not show that a number of GTPases interact with Cdc42, only RhoBTB1. Why is it 'interesting' that not all Rho GTPases impact mitochondrial dynamics? It seems unlikely that they would. IQGAP1 is not a GTPase and yet this is implied in this sentence.

Answer: We have now modified the sentence to make it more clear. The relevant sentences now read as follows: "*Not all RHO GTPases impact mitochondrial dynamics, since depleting the RHO GTPase family Rhobtb1 gene had no effect on mitochondrial morphology, suggesting specificity of RHO GTPase CDC42 towards fission. We also found that silencing Iqgap1, which regulates CDC42 as an upstream scaffold and as a downstream effector of CDC42⁴², increased mitochondrial number (Fig 7c). Indeed, as a scaffold protein, IQGAP1 provides a molecular link between Ca²⁺/calmodulin and CDC42-mediated processes⁴⁵, while as a downstream effector, CDC42 enhances the F-ACTIN-cross-linking activity of IQGAP1 during ACTIN reorganization⁴⁶."*

11. There are multiple incidences of incorrect unit nomenclature in the methods: ml, µl and ug

Answer: We thank the reviewer for this comment. We have now included the correct nomenclature for the units.

Reviewer #2 (Remarks to the Author):

This is possibly the most insanely great response document of all time. Forty-six pages of beautifully formatted experiments carefully addressing every single comment of four reviewers. I hope the reviewers opt in to publishing this review document.

I am fully satisfied that the revised manuscript addresses all of my comments and, I think, those of my colleagues. I compliment the authors on their detailed and fascinating findings.

We are immensely grateful to Reviewer 2 for her/his kind comments.

Reviewer #3 (Remarks to the Author):

This revision represents a substantially improved and at times toned down paper, which addressed most of my original concerns. Aspects of my initial review that the authors did not do as deemed outside of the scope would have made the manuscript stronger, but I do not see as deal breakers given the revision of the language.

I recommend for publication

We are immensely grateful to Reviewer 3 for kindly recommending our manuscript for publication.

Reviewer #4 (Remarks to the Author):

The authors have significantly improved the description of their findings in the revised manuscript and added additional data in support of some of their claims. However, the experiments did not

address my main concern on the original version that mitochondrial fission induced along the mTORC2-NDRG1-CDC42 axis occurs in a DRP1/DNML1-independent manner. The authors base this conclusion on knockdown experiments only and did not generate knockout cells to substantiate this conclusion. They show now that overexpression of NDRG1 can suppress fission defects in DRP1- but not in MFF depleted cells and take this as evidence in support of DRP1-independent fission. Although the effects of NDRG1 overexpression on mitochondrial morphology are convincing, I do not agree with the conclusion of the authors that this support a DRP1-independent fission mechanism. As pointed out in my original comments, low amounts of DRP1 were found to promote membrane fission in other systems that were originally described to be DRP1-independent. Moreover, the authors do not provide strong evidence for an alternative fission mechanism and the role of the rho GTPase CDC42 in this process. It remains unclear whether it acts via the actin cytoskeleton, its canonical function, or directly affects the fission process, as the authors discuss. Although I do appreciate the comprehensive analysis of the effects of fasting on mitochondrial dynamics in liver, I remain less convinced about the mechanistic aspects of this work.

Answer: We have included careful edits to the manuscript, in that, instead of use of the phrase “*Drp1-independent mechanism*”, we have stated that “*By contrast, NDRG1^{WT} failed to restore the alterations in mitochondrial number, area, perimeter and circularity in siMff cells (Fig 5e, Extended data Fig 9b) suggesting that although DRP1 is a key regulator of fission, it appears to not influence mitochondrial fission via the mTORC2-NDRG1 axis.*” This change was included wherever necessary in the text. Furthermore, we have included a statement in the text that although we attempted to generate Dnml1^{KO} cells using CRISPR, we failed to generate KO cells that were healthy enough for experiments.

Final Decision Letter:

Dear Dr Singh,

I am pleased to inform you that your manuscript, "mTORC2-NDRG1-CDC42 axis couples fasting to mitochondrial fission", has now been accepted for publication in Nature Cell Biology. Congratulations on this very nice study!

Over the next few weeks, your paper will be copyedited to ensure that it conforms to Nature Cell Biology style. Once your paper is typeset, you will receive an email with a link to choose the

appropriate publishing options for your paper and our Author Services team will be in touch regarding any additional information that may be required.

Please note that *Nature Cell Biology* is a Transformative Journal (TJ). Authors may publish their research with us through the traditional subscription access route or make their paper immediately open access through payment of an article-processing charge (APC). Authors will not be required to make a final decision about access to their article until it has been accepted. Find out more about Transformative Journals

If you have not already done so, we strongly recommend that you upload the step-by-step protocols used in this manuscript to the Protocol Exchange (www.nature.com/protocolexchange), an open online resource established by Nature Protocols that allows researchers to share their detailed experimental know-how. All uploaded protocols are made freely available, assigned DOIs for ease of citation and are fully searchable through nature.com. Protocols and Nature Portfolio journal papers in which they are used can be linked to one another, and this link is clearly and prominently visible in the online versions of both papers. Authors who performed the specific experiments can act as primary authors for the Protocol as they will be best placed to share the methodology details, but the Corresponding Author of the present research paper should be included as one of the authors. By uploading your Protocols to Protocol Exchange, you are enabling researchers to more readily reproduce or adapt the methodology you use, as well as increasing the visibility of your protocols and papers. You can also establish a dedicated page to collect your lab Protocols. Further information can be found at www.nature.com/protocolexchange/about

With kind regards,

Melina

Melina Casadio, PhD
Senior Editor, Nature Cell Biology
ORCID ID: <https://orcid.org/0000-0003-2389-2243>

** Visit the Springer Nature Editorial and Publishing website at www.springernature.com/editorial-and-publishing-jobs for more information about our career opportunities. If you have any questions please click here.**